# RLHF with Inconsistent Multi-Agent Feedback Under General Function Approximation: A Theoretical Perspective

## Abstract

Reinforcement learning from human feedback (RLHF) has been widely studied, as a method for leveraging feedback from human evaluators to guide the learning process. However, existing theoretical analyses typically assume that the human feedback is generated by the ground-truth reward function. This may not be true in practice, because the reward functions in human minds for providing feedback are usually different from the ground-truth reward function, e.g., due to diverse personal experiences and inherent biases. Such inconsistencies could lead to undesirable outcomes when applying existing algorithms, particularly when considering feedback from heterogeneous agents. Therefore, in this paper, we make the first effort to investigate a more practical and general setting of RLHF, where feedback could be generated by multiple agents with reward functions differing from the ground truth. To address this challenge, we develop a new algorithm with novel ideas for handling inconsistent multi-agent feedback, including a Steiner-Point-based confidence set to exploit the benefits of *multi-agent* feedback and a new weighted importance sampling method to manage complexity issues arising from *inconsistency*. Our theoretical analysis develops new methods to demonstrate the optimality of our algorithm. This result is the first of its kind to demonstrate the fundamental impact and potential of inconsistent multi-agent feedback in RLHF.

## 1 Introduction

Reinforcement learning from human feedback (RLHF) (Casper et al., 2023) has been widely studied as a significant advancement in the field of reinforcement learning, where a learner interacts with the environment sequentially to achieve high cumulative reward. Traditional RL (Sutton, 2018; Agarwal et al., 2019; Vamvoudakis et al., 2021) relies on absolute reward values generated by predefined reward functions to guide the learner's behavior. This limits its applicability in complex real-world scenarios, where crafting reward functions is challenging or ambiguous, e.g., in robotics (Jain et al., 2013), large language models (Ouyang et al., 2022), and image generation (Lee et al., 2023).

RLHF addresses this limitation by leveraging feedback from human evaluators to guide the learning process. Various forms of human feedback have been studied. For example, existing works study RL from comparison/ranking feedback or preference-based feedback, which involves (i) presenting a human with two or multiple outcomes, (ii) allowing her to choose the preferred one, and (iii) guiding the learning process towards better policies based on the received human feedback (Wang et al., 2023; Zhu et al., 2023; Chakraborty et al., 2024; Ye et al., 2024; Chen et al., 2022; Chatterji et al., 2021; Kaufmann et al., 2023; Li et al., 2023; Du et al., 2024). In this way, RLHF bridges the gap between pure algorithmic optimization and the nuanced understanding of human judgment.

However, existing theoretical results on RLHF typically rely on the human feedback generated by the ground-truth reward function $R^*(\cdot)$. For example, the commonly used comparison model assumes that: the human feedback is generated according to a Bernoulli distribution based on the value of a link function $\sigma(R^*(\tau_1) - R^*(\tau_0))$, where $R^*(\cdot)$ is assumed to be the ground-truth reward function and $\{\tau_i\}_{i=0,1}$ are two outcomes. If the Bradley-Terry-Luce model (Bradley & Terry, 1952) is considered for the link function $\sigma(\cdot)$, then the human feedback is $\tau_1 \succ \tau_0$ (i.e., outcome $\tau_1$ is preferred to outcome $\tau_0$) with probability equal to $\exp\left(R^*(\tau_1)\right)/\left[\exp\left(R^*(\tau_1)\right) + \exp\left(R^*(\tau_0)\right)\right]$.

In a word, this type of human feedback is generated by the ground-truth reward function $R^*(\cdot)$. Due to page limits, we defer further discussion of related work to Appendix A.

**Inconsistency in the Feedback:** Feedback may not be consistent in practice, due to subjective human judgment, inherent biases, and varying expertise levels (Tjuatja et al., 2024; Yan et al., 2024). That is, human feedback in practice often suffers from *inconsistency* (see the details in Sec. 2.2). For example, instead of being generated by $R^*(\cdot)$, the real-world human feedback is often generated based on $\sigma(R^{\text{human}}(\tau_1) - R^{\text{human}}(\tau_0))$. Here, $R^{\text{human}}(\cdot)$ is the reward function in the human mind, and it is often different from the ground-truth reward function, i.e., $R^{\text{human}}(\cdot) \neq R^*(\cdot)$. Traditional RLHF theories, which often assume a ground-truth reward function $R^*(\cdot)$, may not be applicable in this more uncertain setting. Particularly, if assuming $R^{\text{human}}(\cdot) = R^*(\cdot)$, the resulting policy could overfit to certain subjective signals rather than generalizing effectively. *Therefore, in this paper, we address these unique challenges posed by inconsistent human feedback in the algorithm design and theoretical analysis, and investigate the fundamental impact of this type of inconsistency in RLHF.*

**Multi-Agent Feedback:** Existing theoretical analysis in RLHF leaves untapped potential for richer and more diverse feedback sources. That is, in addition to human evaluators, feedback can be sourced from AI models, data analyzers, and other automated tools (Lee et al., 2024; Guo et al., 2024a). (We call these sources "agents".) Heterogeneity among agents in understanding and interpretation could create a wide spectrum of feedback quality, because of diverse personal experience and varying expertise levels. *Therefore, we investigate the power of feedback from multiple agents.*

Due to multi-agent feedback, the inconsistency issue becomes even more pronounced. On the one hand, discrepancies among agents complicate the learning process, as the policy must navigate and reconcile conflicting signals. This requires us to explore strategies for harmonizing diverse inputs to align more closely with ground-truth objectives. On the other hand, we should intuitively be able to leverage multiple data streams of agent feedback simultaneously, such that individual biases can be reduced. To address these challenges, in this work, we investigate the following open problem:

> *Whether multi-agent feedback with inconsistency in RLHF fundamentally helps the learning process or exacerbates the situation?*

To answer this, we theoretically characterize the fundamental impact and potential of inconsistent multi-agent feedback. Specifically, we study online RLHF with inconsistent multi-agent feedback under general function approximation. In addition to the well-known difficulties in RLHF and in analyzing under general function approximation, the aforementioned properties of *inconsistent multi-agent feedback* introduce significant new challenges in both algorithm design and regret analysis.

*Sharp Regret Under Inconsistency:* We formulate the inconsistency in the multi-agent feedback by the cumulative discrepancy between the human preference model and the ground-truth preference model (see Eq. (2)). Eq. (2) is general and does not require special structures in the inconsistency. Nonetheless, we are able to provide sharp theoretical guarantees. Note that the regret considered in Eq. (3) is essentially the worst-case pseudo-regret, but over all possible human reward functions satisfying the inconsistency model. As a result, our theoretical regret guarantee not only works for the agents providing feedback during the online learning process, but also works for any newly-incoming inconsistent agent, as long as her reward function satisfies the inconsistency model.

*New Algorithm Design and Analytical Ideas:* From a high-level point of view, the steps of our new algorithm include: (i) dynamically searching for the confidence center based on the multi-agent feedback; (ii) constructing a confidence set based on step i and an important subset of inconsistent feedback; (iii) reforming the confidence set in step ii to capture ground-truth comparison with high probability; and (iv) constructing a high confidence policy set to circumvent the absolute reward uncertainty. In this way, the optimal policy can be approximately found with high probability. The new ideas that have been developed are described below.

**New Idea I: Steiner-Point-Based Confidence Center for Leveraging Multi-Agent Feedback.** Since the feedback is inconsistent, a natural idea would be to use the feedback of each agent to estimate their reward models, and then search for the optimal policy jointly. However, this will lose the fundamental power of multi-feedback, i.e., the resulting performance does not *improve* with the number of agents. Thus, we should estimate the confidence center by utilizing multi-feedback simultaneously. However, the traditional complexity analysis in RL does not apply, since the confidence center may be *outside* of the agent reward function space and arbitrarily dynamic

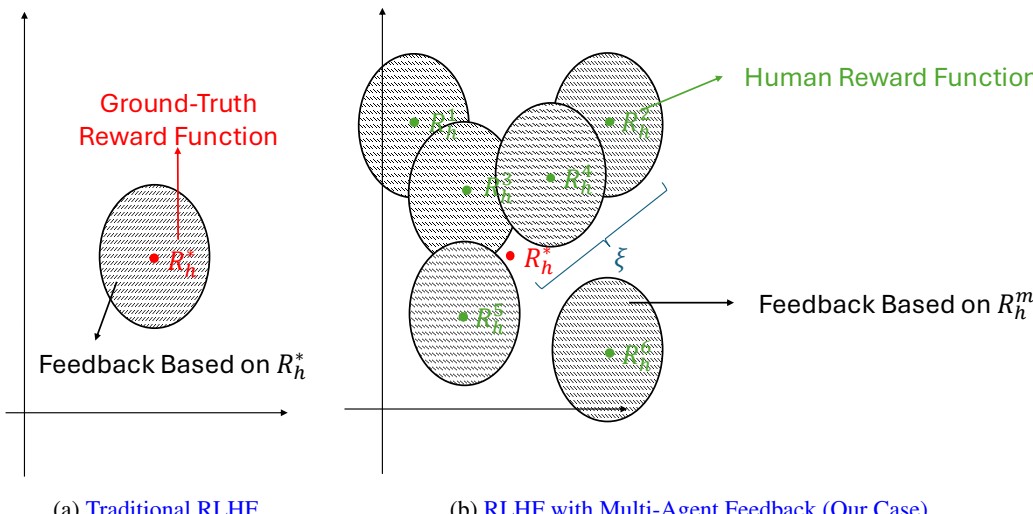

(a) Traditional RLHF          (b) RLHF with Multi-Agent Feedback (Our Case)

Figure 1: Feedback comparison for tradition RLHF case and our case: in our case, the feedback is based on heterogeneous reward functions $R_h^m$, which could be different from the ground truth $R_h^*$

due to inconsistency (see Fig. 1 and Fig. 2). To address this new difficulty, we non-trivially modify Steiner-Point Approximation from theoretical physics and combinatorial geometry (Brazil et al., 2014), which requires fundamentally new analytical methods in RL for a sub-linear regret.

**New Idea II: Sub-Importance Sampling for Reducing Functional Complexity.** Due to the nature of multi-agent feedback and general function approximation, the traditional sample-based complexity would result in a final regret increasing linearly in time horizon $K$. To address this new difficulty, we design a parameterized approximation method for sub-importance sampling under Fermat analysis, such that the functional complexity is reduced as it is based on only a subset of sensitive samples, where the new layer of complexity can be fundamentally reduced and captured in the analysis.

**New Idea III: Scaled Confidence-Based Weights for Reducing Biases and Optimism-in-the-Face-of-Policy-Uncertainty (OFPU).** Existing ideas for addressing biases in the sampling feedback are to add weights to the action selection step. Directly applying this does not work due to the heterogeneous feedback in our case. To resolve this, we design a fundamentally different scaled weight directly on the policy, such that a greedy decision under policy uncertainty in our case still guarantees optimality. Particularly, due to the inconsistent discrepancy, the estimated reward function will always contain a layer of inconsistency. Thus, a $V$-value function is not well-defined. Instead, we construct the policy set directly based on the new bonus terms, i.e., in the face of policy uncertainty.

## 2 PROBLEM FORMULATION

In this section, we introduce the online RLHF setting that we study, especially the inconsistent multi-agent feedback considered in this paper, as well as notions for general function approximation.

### 2.1 REINFORCEMENT LEARNING FROM HUMAN FEEDBACK (RLHF)

We investigate RLHF in episodic Markov decision processes (MDPs), where an online learner interacts with the environment in $K$ episodes. It is typically modelled by $(H, \mathbb{S}, \mathbb{A}, \mathbb{P})$, where $H$ denotes the number of steps in each episode; $\mathbb{S}$ and $\mathbb{A}$ denote the state space and action space, respectively; and $\mathbb{P} : \mathbb{S} \times \mathbb{S} \times \mathbb{A} \to [0, 1]$ denotes the *unknown* transition kernel.[1] At each step $h$ of an episode $k$, based on the current state $s_{k,h}$, the online learner takes an action $a_{k,h}$. Then, the environment transits to the next state $s_{k,h+1}$, which is drawn according to the transition probability $\mathbb{P}(\cdot|s_{k,h}, a_{k,h})$.

---

[1] As typically assumed, we let the initial state in each episode be fixed, i.e., $s_{k,1} = s_1 \in \mathbb{S}$. This can be generalized to the case where $s_{k,1}$ is sampled from a fixed distribution $\Delta_1$ for each episode $k$.

In RLHF, human feedback is typically used to guide the learning process. One conventional human feedback in each episode is a comparison of two trajectories $\tau_k \triangleq (s_{k,1}, a_{k,1}, \ldots, s_{k,H}, a_{k,H})$ and $\tau_0 \triangleq (s_{0,1}, a_{0,1}, \ldots, s_{0,H}, a_{0,H})$ (Wang et al., 2023; Zhu et al., 2023; Du et al., 2024; Zhan et al., 2024). In this case, the feedback is $f_k = 1$, i.e., trajectory $\tau_k$ is preferred to trajectory $\tau_0$ (denoted by $\tau_k \succ \tau_0$), with probability $\sigma(R^*(\tau_k) - R^*(\tau_0))$, where $R^*(\cdot)$ is an unknown ground-truth reward function and $\sigma(\cdot)$ is a link function. Note that this human feedback $f_k$ is generated by a comparison based on the ground-truth reward function $R^*(\cdot)$. This may not be true in practice, due to subjective human judgment, varying expertise levels, diverse personal experience, inherent biases, etc.

## 2.2 INCONSISTENT MULTI-AGENT FEEDBACK

Therefore, in this paper, we extend aforementioned traditional RLHF to a more practical and general online setting, i.e., RLHF with *inconsistent multi-agent feedback*, formalized as follows.

**Multi-Agent Feedback:** We consider feedback that could be generated by multiple agents, e.g., humans (Chakraborty et al., 2024), AI models (Lee et al., 2024), and data analyzers (Guo et al., 2024a). Specifically, at the end of each episode $k$, there are $M$ agents providing comparison feedback $f_k^m$, where $m = 1, \ldots, M$ is the index of the agent. This type of multi-agent feedback has received attention in empirical studies recently. However, to our knowledge, a theoretical understanding of the fundamental impact of (inconsistent) multi-agent feedback is still an open problem.

**Inconsistency in the feedback:** We consider the human feedback that could include inconsistency, i.e., the human feedback is not generated based on the ground-truth reward function $R^*(\cdot)$. Specifically, the feedback $f_k^m$ from each agent $m$ is a Bernoulli random variable with probability[2]

$$\mathcal{P}(f_k^m = 1) \triangleq \mathcal{P}^m(\tau_k \succ \tau_0) = \sigma(R^m(\tau_k) - R^m(\tau_0)), \tag{1}$$

where $R^m(\cdot) : \{\tau\} \to [0,1]$ is the *unknown* reward function of agent $m$, $\{\tau\}$ is the state-action trajectory space with slight abuse of notation, $\tau_0 \triangleq (s_{0,1}, a_{0,1}, \ldots, s_{0,H}, a_{0,H})$ is a fixed trajectory of state-action pairs, and $\tau_k$ is the trajectory visited in episode $k$. We highlight two layers of inconsistency here: (i) the reward function $R^m(\cdot)$ in the mind of each agent $m$ could be **different** from that in the mind of others, and (ii) $\{R^m(\cdot)\}_{m=1}^M$ could be **different** from the ground-truth reward function $R^*(\cdot)$. This is why we call the multi-agent feedback "inconsistent". More specifically, such inconsistency among $R^m$'s and $R^*$ can be formulated by the following inconsistency model:

$$\max_{(\tau_k)_{k=1}^K, \tau_0} \sum_{k=1}^K |\sigma(R^*(\tau_k) - R^*(\tau_0)) - \sigma(R^m(\tau_k) - R^m(\tau_0))| \le \xi, \forall m \in [M]. \tag{2}$$

Note that the inconsistency model in Eq. (2) is general. It only assumes an upper bound on the cumulative worst-case discrepancy between comparisons (i.e., not the absolute values) based on the reward function $R^m(\cdot)$ of each agent and the ground-truth reward function $R^*(\cdot)$. Thus, the discrepancy of each agent could be different, and hence $R^m(\cdot)$ could be different from each other. Moreover, Eq. (2) does not require special structures in the inconsistency. Further, if $\xi = 0$, our setting reduces to the setting without inconsistency, where all agents provide feedback based on the ground-truth reward function. In addition, if $\xi = 0$ and $M = 1$, our setting reduces to the traditional setting, where one human provides feedback generated by the ground-truth function.

**Example 1 (Inconsistent multi-agent feedback in autonomous driving):** When evaluating which maneuver or course is the best during the training of a vehicle, different agents may prioritize different aspects based on her subjective habits, such as safety, timeliness, fuel efficiency, and comfort. This leads to inconsistent opinions on the best course of actions and locations. For instance, assuming course $\tau_1$ is safer and more comfortable, while course $\tau_0$ is faster and more direct. Consider the case of two agents. Agent $m = 1$ might emphasize safety and comfort above all. Thus, she chooses a slower, but more cautious and comfort course (which turns out to be bad), e.g., $R^1(\tau_1) - R^1(\tau_0) = 0.8$. Agent $m = 2$ may prioritize timely arrival. Thus she chooses a faster and more direct path, even if it involves greater risk, e.g., $R^2(\tau_1) - R^2(\tau_0) = -0.2$. However, due to more complicated considerations, such as minimizing traffic disruptions or environmental impact, the ground-truth reward function may suggest that $R^*(\tau_1) - R^*(\tau_0) = 0.4$. Such inconsistency introduces variability in the data, which significantly challenges the RL process.

---

[2]With simple modification, our results can be applied to other settings, e.g., the comparison is based on each state-action pair, preference-based model, and ranking feedback.

## 2.3 PERFORMANCE METRIC - REGRET UNDER INCONSISTENCY

We evaluate the performance of the online RLHF algorithm by the regret under inconsistency, i.e.,

$$\text{Reg}(K) = \textbf{max}_{\textbf{Eq. (2)}} \sum_{k=1}^{K} \left[ V^*(\tau_0) - V^{\pi_k}(\tau_0) \right], \qquad (3)$$

where $V^*(\tau_0) = \max_\pi \mathbb{E} \left[ \sigma(R^*(\tau^\pi) - R^*(\tau_0)) \right]$ is the optimal $V$-value, $V^{\pi_k}(\tau_0) = \mathbb{E}[\sigma(R^*(\tau^{\pi_k}) - R^*(\tau_0))]$, and $\tau^\pi$ denotes the trajectory after implementing policy $\pi$. Note that (i) The regret in Eq. (3) is under the worst-case inconsistent feedback satisfying Eq. (2), i.e., the "max" part in Eq. (3). As a result, our solution works not only for the $M$ agents providing feedback for the online learning process, but also for any newly-coming agent, as long as the reward function $R(\cdot)$ in her mind satisfies Eq. (2). (ii) $V$-value in Eq. (3) is based on the comparison, since we could only learn the reward up to a constant, due to the fact that the agent feedback is only a comparison. (iii) If the regret in Eq. (3) is evaluated based on the unknown $R^m(\cdot)$, our results still hold, with only a constant factor difference. (iii) Achieving a low regret under such inconsistency in RLHF requires novel ideas in both the algorithm design and regret analysis. *To the best of our knowledge, we are the first to study such fundamental impact and potential of inconsistent multi-agent feedback in RLHF from a theoretical perspective.*

## 2.4 GENERAL FUNCTION APPROXIMATION

We consider general function approximation. Below, we provide the definitions of the standard covering number and eluder dimension for capturing the complexity of a function space.

**Definition 1.** *($\epsilon$-covering number) Let $(\mathcal{F}, \|\cdot\|)$ be a metric space, where $\mathcal{F}$ is the function class and $\|\cdot\|$ is the norm used to measure distances between functions. A set of functions $\{f_1, \ldots, f_N\} \subset \mathcal{F}$ is called an $\epsilon$-covering set if for every $f \in \mathcal{F}$, there exists some $f_n$, s.t., the distance $\|f - f_n\| \leq \epsilon$. The $\epsilon$-covering number $\mathcal{N}(\mathcal{F}, \|\cdot\|, \epsilon)$ is the minimum number $N$ of functions in an $\epsilon$-covering set.*

The $\epsilon$-covering number $\mathcal{N}(\mathcal{F}, \|\cdot\|, \epsilon)$ captures how "complex" the function class $\mathcal{F}$ is, i.e., how many different functions are required to approximate any function in the class to within $\epsilon$ accuracy.

**Definition 2.** *(Eluder dimension) Let $\mathcal{F}$ be a class of real-valued functions over a domain $\mathcal{X}$. For a set of previously observed points $\mathcal{X}_N = \{x_1, x_2, \ldots, x_N\} \subset \mathcal{X}$, define the following:*

*- A point $x \in \mathcal{X}$ is said to be $\epsilon$-dependent of $\mathcal{X}_N$ with respect to the function class $\mathcal{F}$ if, for all pairs of functions $f_1, f_2 \in \mathcal{F}$ satisfying $\sqrt{\sum_{n=1}^{N} \left( f_1\left(x_n\right) - f_2\left(x_n\right) \right)^2} \leq \epsilon$, it holds that $|f_1(x) - f_2(x)| \leq \epsilon$. Further, $x$ is $\epsilon$-independent of $\mathcal{X}_N$ with respect to $\mathcal{F}$ if $x$ is not $\epsilon$-dependent on $\mathcal{X}_N$.*

*- The eluder dimension $\dim_E(\mathcal{F}, \epsilon)$ is the largest number of points in set $\mathcal{X}_N$ such that, for some $\epsilon' \geq \epsilon$, each point $x_n$ ($n \in [N]$) is $\epsilon$-independent of its previous points $\{x_1, x_2, \ldots, x_{n-1}\}$.*

The $\epsilon$-dependency shows that the new point $x$ cannot be used to significantly distinguish between functions in $\mathcal{F}$ that agree on the previous data points. The eluder dimension measures how dependent or entangled the predictions of different functions in $\mathcal{F}$ are across the state or state-action space.

# 3 ALGORITHM DESIGN

In this section, we present our new RLHF algorithm for solving the problem defined in Sec. 2. We focus on introducing the three main new ideas for addressing inconsistent multi-agent feedback.

## 3.1 RLHF WITH INCONSISTENT MULTI-AGENT FEEDBACK

The algorithm is formally provided in Algorithm 1. From a high level perspective, in each episode, our algorithm first executes a sub-importance sampling to guarantee the functional complexity not increase linearly with the time horizon (line 3). Next, by applying a Steiner point method, we construct the confidence center that could be outside of the reward space (see Fig. 1 and line 4) and the corresponding confidence set for the reward functions (line 5). Then, based on the trajectories sampled under the Steiner point method, we reform the confidence center and confidence set for the transition kernel (line 7 and line 8). Finally, based on the bonus terms for both reward and transition, we update the policy greedily in each episode (line 10). Below, we focus on introducing these four main new ideas in our algorithm design to enable online RLHF with inconsistent multi-agent feedback. Define $\Gamma_k \triangleq \{\tau_t\}_{t \in [k]}$ and $\sigma(\tau \mid R) \triangleq \sigma(R(\tau) - R(\tau_0))$.

---

**Algorithm 1** RLHF with Inconsistent Multi-Agent Feedback (RLHF-IMAF)

---

1: **Initialization:** Set $\beta_{\mathbb{T}} = \beta_{\mathbb{P}} = 8\log\left(2K\mathcal{N}\left(\mathcal{F}_{\mathbb{T}}, 1/K, \|\cdot\|_{\infty}\right)/\delta\right)$

2: **for** episode $k = 1 : K$ **do**

3:     ▷ ▷ ▷ *New Idea II:*

4:     Add $1/p_{\tau}$ copies of each trajectory $\tau \in \Gamma_{k-1}$ into $\Gamma'_{k-1}$ with probability $p_{\tau}$, where $p_{\tau} = \min\left\{p \in \mathbb{R} \mid p \geq \min\left\{1, \mathcal{T}_{\Gamma,\mathcal{R},\lambda}(\tau) \cdot 72\ln(4\mathcal{N}(\mathcal{R}, \varepsilon/72 \cdot \sqrt{\lambda\delta/(|\Gamma|)})/\delta)/\varepsilon^2\right\}, 1/p \in \mathbb{Z}\right\}$

5:     ▷ ▷ ▷ *New Ideas I and II:*

6:     Update the Steiner-point-based reward confidence center $\hat{R}_k$ according to Eq. (8)

7:     Update the confidence set $\mathcal{R}_k$ for the reward function according to Eq. (12)

8:     Update the bonus term for the reward function exploration as follows,

$$b_k^R(\tau) = \max_{R \in \mathcal{R}_k}\left|\sigma(\tau \mid R) - \sigma\left(\tau \mid \hat{R}_k\right)\right| / \sqrt{\lambda + \sum_{\substack{t \in [k-1], \\ \tau \in \Gamma_{t|t-1}}} \frac{\left(\sigma(\tau|R) - \sigma(\tau|\hat{R}_k)\right)^2}{\max\left\{1, \Lambda_t(\theta)/\left|\sigma(\tau|R) - \sigma(\tau|\hat{R}_t)\right|\right\}}}, \quad (4)$$

9:     ▷ ▷ ▷ *New Ideas I and III:*

10:     Update the Steiner-point-based transition confidence center according to Eq. (14)

11:     Update the confidence set for the transition kernel according to Eq. (15)

12:     Update the bonus term for the transition kernel exploration as follows,

$$b_k^P(\tau) = \sum_{(s,a) \in \tau} \max_{\substack{V \in \mathcal{V} \\ P' \in \mathbb{P}_k}} \frac{\left(P'(\cdot \mid s, a) - \hat{P}_k(\cdot \mid s, a)\right) V(s, a)}{\left(\lambda + \sum_{\substack{t \in [k-1], \\ \tau \in \Gamma_{t|t-1}}} \frac{\left\langle[P' - \hat{P}_k]\left(\cdot|s_{t,h}, a_{t,h}\right), V_{t,h}\right\rangle^2}{\max\left\{1, \Lambda_t^P(\theta)/\left|\left\langle[P' - \hat{P}_t]\left(\cdot|s_{t,h}, a_{t,h}\right), V_{t,h}\right\rangle\right|\right\}}\right)^{1/2}}. \quad (5)$$

13:     ▷ ▷ ▷ *New Idea III:*

14:     Execute the following policy for episode $k$ according to Eq. (18)

15:     Collect the trajectory $\tau_k$ and preference $f_k^m$ from all agents.

16: **end for**

---

As discussed in the introduction, since it is highly unclear whether multi-agent feedback with inconsistency fundamentally helps the learning or exacerbates the situation, the difficulty is how to leverage the potential and circumvent the biased in such feedback.

**New Idea I: Steiner-Point-Based Confidence Center for Leveraging Multi-Agent Feedback (Illustrated in Fig. 2b).** Applying existing ideas for function estimation does not work in our case, due to the inconsistency and heterogeneity in the feedback. This undertanding is fundamentally important for the later algorithm design and theoretical analysis, thus let us elaborate more as follows.

Specifically, to estimate the reward function, a natural but naïve way would be to apply the least-squares method to the feedback from each agent, i.e.,

$$\hat{R}_k^m = \arg\min_{R' \in \mathcal{R}} \sum_{t=1}^{k-1}\left(\sigma\left(R'(\tau_t) - R'(\tau_0)\right) - f_t^m\right)^2, \quad (6)$$

where $\hat{R}_k^m$ denotes the estimated reward function of agent $m$ and $\mathcal{R}$ denotes the agent reward function space. Note that this does not utilize the mutual information $I\left(\mathbf{f}^{m_i}; \mathbf{f}^{m_j}\right) = D_{\mathrm{KL}}\left(P_{(\mathbf{f}^{m_i}, \mathbf{f}^{m_j})} \| P_{\mathbf{f}^{m_i}} \otimes P_{\mathbf{f}^{m_j}}\right)$ among the agents, and thus the resulting regret would not improve when more agents are providing feedback. For example, in Fig. 1, the useful overlaps between feedback generated based on different $R_h^m$'s are not effectively utilized.

To address this, intuitively, if the reward functions in all human minds were identical, we could consider them jointly. Then, according to the chain rule of mutual information, i.e., $I\left(\mathbf{f}^{m_1}, \ldots, \mathbf{f}^{m_{i-1}}; \mathbf{f}^{m_i}\right) = \sum_{j=1}^{i-1} I\left(\mathbf{f}^{m_j}; \mathbf{f}^{m_i} | \mathbf{f}^{m_1}, \ldots, \mathbf{f}^{m_{j-1}}\right)$, by considering the estimate

$$\hat{R}'_k = \arg\min_{R' \in \mathcal{R}} \sum_{t=1}^{k-1} \sum_{m=1}^{M}\left(\sigma\left(R'(\tau_t) - R'(\tau_0)\right) - f_t^m\right)^2, \quad (7)$$

the performance would improve with the number of agents $M$. Compared to the estimate $\hat{R}_k^m$ above, the difference here is to consider the feedback from all $M$ agents jointly, i.e., shown by the sum over all $m$ and the estimate $\hat{R}'_k$ is no longer indexed by (or designed for) each agent $m$ separately.

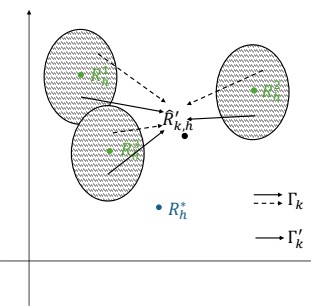 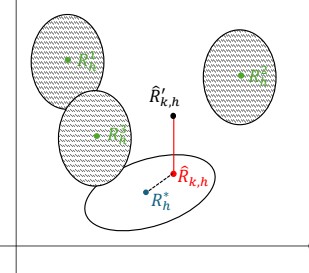 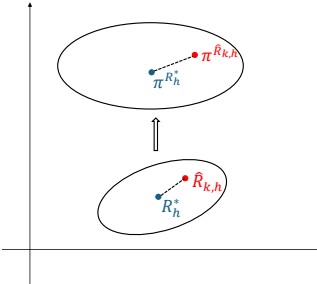

(a) New Idea II: Sub-importance sampling, where solid and dash arrows illustrates the choice of historical trajectories $\hat{\Gamma}_k$ from $\Gamma_k$

(b) New Idea I: Steiner point method, where the mapping from simple average $\hat{R}'_k$ to the Fermat point $\hat{R}_k$ illustrates Steinerization

(c) New Idea III: Scaled weighting and OFPU, where mapping from lower function space to upper policy space illustrates such transfer

Figure 2: Illustration of new ideas in our algorithm design

However, the estimate $\hat{R}'_k$ still does not work, because the reward functions of the agents are actually *not* identical due to the inconsistency in the feedback (see Eq. (2)). Then, one may conclude that when there exists such inconsistency, multi-agent feedback does not help any more. For example, if the agents are highly biased and do not agree with each other, multiple copies of feedback from these agents do not tell us anything about the ground truth. Thus, an open fundamental question remains: whether multi-agent feedback with inconsistency actually helps or exacerbates the situation?

With a deeper thought experiment, we could notice that, since KL divergence $D_{\mathrm{KL}}\left(P_i(f)\|P_j(f)\right) = \sum_f P_i(f) \log \frac{P_i(f)}{P_j(f)}$ is convex in the pair $(P_i, P_j)$, by carefully constructing the confidence center based on the multi-agent feedback, we could still push the estimation of the reward function closer to the ground truth. *Then, the non-trivial question is where such a confidence center is.*

Motivated by theoretical physics and combinatorial geometry, we provide a novel idea to answer this question based on the Steiner point (Gilbert & Pollak, 1968; Brazil et al., 2014). Specifically, the Steiner point is a generalization of the Fermat–Torricelli point. From a geometrical point of view, it is defined to be a point with the minimum total distance to all input points. The effectiveness of Steiner point comes from the fact that it could be a *new* point added to solve a problem, i.e., the solution set could be expanded from the original constrained set based on inputs to a larger set with more flexibility. In our case, when restricting ourselves to the ill-structured agent reward function space, the solution may get stuck due to the inconsistency. After enlarging the space, we could leverage the convexity of KL divergence mentioned above, and hence get closer to the ground truth.

However, the difficulty in applying Steiner point to our problem is that the optimization, i.e., the estimation, for the reward function is based on the randomness of the sampled data, and thus the data covering complexity would be exponential. Despite the worse-case complexity, a polynomial-sized approximate kernelization scheme is still possible. For example, for any $\alpha > 0$, the connected vertex covering algorithm achieves a polynomial-sized kernel with only a $\alpha$ estimation error (Lokshtanov et al., 2017). Therefore, to leverage the potential of multi-agent feedback under inconsistency, we use the heterogeneous feedback joint in an expanded reward function space as follow (Fig. 2b):

$$\hat{R}_k = \arg\min_{\substack{R' \in \mathcal{R}_{k-1} \cap \\ \left\{R' \mid \min_{R \in \mathcal{R}} \|R'-R\|_{\mathrm{RTV}} \leq \alpha\right\}}} \sum_{m=1}^M \sum_{t=1}^{k-1} \sum_{\tau \in \hat{\Gamma}_{t|t-1}} \frac{\left(\sigma(\tau|R') - f_t^m\right)^2}{\max\left\{1, \Lambda_t(\theta) / \left|\sigma(\tau|R) - \sigma(\tau|\hat{R}_t)\right|\right\}},$$

(8)

where the objective function is the Steiner point target function with domain in the expanded space $\bar{\mathcal{R}}_\alpha \triangleq \{R' \mid \min_{R \in \mathcal{R}} \|R' - R\|_{\mathrm{RTV}} \leq \alpha\}$, the reward total variance (RTV, with a slight abuse of notation) is defined to be $\|\cdot\|_{\mathrm{RTV}} \triangleq \max_{(\tau_k)_{k=1}^K, \tau_0} \sum_{k=1}^K \|R'(\tau_k) - R'(\tau_0)| - |R(\tau_k) - R(\tau_0)\|$, $R_k$ is defined in Eq. (12), $\Lambda_t^R(\theta) \triangleq \theta \sqrt{\lambda + \sum_{i=1}^{t-1} \sum_{\tau \in \hat{\Gamma}_{i|i-1}} \left(\sigma(\tau \mid R) - \sigma(\tau \mid \hat{R}_i)\right)^2}$, and $\hat{\Gamma}_{t|t-1} \triangleq \hat{\Gamma}_t - \hat{\Gamma}_{t-1}$. Note that there is a trade-off related to the tunable parameter $\alpha$, e.g., with

a larger $\alpha$, the optimal solution is getting closer to the ground-truth, while the kernel size will be larger, and vice versa.

**New Idea II: Sub-Importance Sampling for Reducing Functional Complexity (Illustrated in Fig. 2a).** Conventionally, in each episode $k$, based on our new Idea I and the replay buffer $\{(\tau_t, f_t^m)\}_{(t,m)\in[k-1]\times[M]}$ that contains all historical data, we can construct the confidence set,

$$\mathcal{R}'_k = \left\{ R' \in \bar{\mathcal{R}}_\alpha \cap \mathcal{R}_{k-1} \mid \sum_{t=1}^{k-1} \left( \sigma\left(R'(\tau_t) - R'(\tau_0)\right) - \sigma\left(\hat{R}_k(\tau_t) - \hat{R}_k(\tau_0)\right)\right)^2 \le \beta^R \right\}, \tag{9}$$

such that the ground-truth reward function $R^*(\cdot)$ is contained with high probability, by choosing the parameter $\beta^R$ correctly. After collecting more and more sampling data by repeating this procedure along all $K$ episodes, the confidence set will be pushed to navigate the ground truth $R^*(\cdot)$, according to the law of large numbers. As a result, a greedy policy based on the $\hat{Q}$-value function constructed on the reward function in the confidence set will be nearly optimal. To encourage such greedy exploration, a bonus term $b_{k,h}$ is usually designed to be the width of the confidence set $\mathcal{R}'_k$, i.e., $b_{k,h} = w\left(\mathcal{R}'_k\right) \triangleq \max_{R_1, R_2 \in \mathcal{R}_k} |\sigma\left(R_1(\tau)\right) - \sigma\left(R_2(\tau)\right)|$, such that $\hat{Q}_{k,h+1}(s,a)$ is guaranteed to be an overestimate of the true $Q$ value $r(\cdot,\cdot) + \sum_{s' \in \mathcal{S}} P\left(s' \mid \cdot, \cdot\right) V_{k,h+1}\left(s'\right)$ with high probability, where the $V$-value function is $V_{k,h+1}(\cdot) = \max_{a \in \mathcal{A}} Q_{k,h+1}(\cdot, a)$.

However, in doing so in our case, two new issues will arise. First, since the confidence set $\mathcal{R}_k$ above relies on all historical data, i.e., represented by the sum over all episodes $1, \dots, k-1$, the bonus term $b_{k,h}$ will also rely on all these data. Then, the complexity could increase linearly with time horizon $T = KH$. One idea to address this is importance sampling, i.e., only include important state-action pairs in the estimation (Langberg & Schulman, 2010; Wang et al., 2020). However, the Steiner-point-based confidence center in Eq. (8) relies on $\mathcal{R}_{k-1}$, and hence will be affected by such sampling. To resolve this new issue, we develop a novel "sub-importance sampling", with the new development mainly on how to determine the importance of the historical data.

Specifically, we first introduce an important notion in such sampling. For a given set of trajectories $\Gamma \subseteq \{\tau\}$ and a function class $\mathcal{R}$, for each $\tau \in \Gamma$, the $\lambda$-sensitivity of $\tau$ with respect to $\Gamma$ and $\mathcal{R}$ is

$$\mathcal{T}_{\Gamma, \mathcal{R}, \lambda}(\tau) \triangleq \max_{R, R' \in \mathcal{R}, \sum_{\tau \in \Gamma}(R(\tau) - R'(\tau))^2 \ge \lambda/(1+\alpha)} \left(R(\tau) - R'(\tau)\right)^2 / \sum_{\tau' \in \Gamma} \left(R(\tau') - R'(\tau')\right)^2. \tag{10}$$

Sensitivity measures the importance of each trajectory $\tau$ in $\Gamma$ with respect to the function pairs $R, R' \in \mathcal{R}$, such that $\tau$ contributes the most to $\sum_{\tau' \in \Gamma} \left(R(\tau') - R'(\tau')\right)^2$. Thus, the trade-off is that, intuitively with larger $\alpha$, Steiner-point-based confidence center is better constructed, but the bonus complexity will be larger. To handle this new trade-off, we filter the historical samples, i.e.,

$$\hat{\Gamma}_k = \{\tau \in \Gamma_k \mid \tau \in \mathcal{C}(\Theta, 1/(8\sqrt{4T/\delta})), \sup_{R, R' \in \bar{\mathcal{R}} \cap \mathcal{R}_{k-1}} |R(\tau) - R'(\tau)| \le 1/(8\sqrt{4T/\delta})\}, \tag{11}$$

where $\bar{R}_k = \{R \in \mathcal{C}(\bar{\mathcal{R}} \cap \mathcal{R}_{k-1}, 1/(8\sqrt{4T/\delta})) \mid \|\bar{R}_k - \hat{R}_k\| \le 1/(8\sqrt{4T/\delta})\}$ is a confidence-center-based shifted covering set. In this way, we only consider the samples from a set guaranteeing sufficient covers (Fig. 2a). Based on this and the constructed confidence center, the confidence set is

$$\mathcal{R}_k = \left\{ R' \in \bar{\mathcal{R}}_\alpha \cap \mathcal{R}_{k-1} \mid \lambda + \sum_{t=1}^{k-1} \sum_{\tau \in \hat{\Gamma}_{t|t-1}} \frac{\left(\sigma(\tau|R') - \sigma(\tau|\hat{R}_k)\right)^2}{\max\left\{1, \Lambda_t(\theta)/\left|\sigma(\tau|R') - \sigma(\tau|\hat{R}_t)\right|\right\}} \le \beta^R \right\}. \tag{12}$$

Second, constructing $\mathcal{R}_k$ in Idea II requires the reward function of each state-action pair, such that the value function at each step $h$ can be calculated. Such a reward value is not available in RLHF settings. Tackling this problem is relatively easier (Ayoub et al., 2020; Ye et al., 2023). We define the loss function as $L_k\left(\mathbb{P}_1, \mathbb{P}_2\right) = \sum_{t=1}^{k-1} \sum_{h=1}^{H} \left(\langle \mathbb{P}_1\left(\cdot \mid s_{t,h}, a_{t,h}\right) - \mathbb{P}_2\left(\cdot \mid s_{t,h}, a_{t,h}\right), V_{t,h}\rangle\right)^2$. Next, we construct the high confidence set for transition $\mathbb{P}$:

$$\mathcal{B}_k^{\mathbb{P}} = \left\{\mathbb{P}' \mid L_k\left(\mathbb{P}', \hat{\mathbb{P}}_k\right) \le \beta^{\mathbb{P}}\right\}. \tag{13}$$

The exploration bonus $b_k^{\mathbb{P}}(s, a, V)$ for the transition estimation then measures the uncertainty of $\mathcal{B}_k^{\mathbb{P}}$, i.e., $b_k^{\mathbb{P}}(s, a, V) = \max_{\mathbb{P}_1, \mathbb{P}_2 \in \mathcal{B}_k^{\mathbb{P}}} \left(\mathbb{P}_1 - \mathbb{P}_2\right) V(s, a)$. Suppose $V_{\max, k, s, a} = \arg\max_{V \in \mathcal{V}} b_k^{\mathbb{P}}(s, a, V)$,

then we use $V_{\max,t,s_{t,h,i},a_{t,h,i}}$ as the online target for the history sample $(s_{t,h}, a_{t,h}, s_{t,h+1})$. With a slight abuse of notation, we use $b_k^{\mathbb{P}}(s,a) = \max_{V \in \mathcal{V}} b_k^{\mathbb{P}}(s,a,V)$ to denote the maximum uncertainty for a given state-action pair $(s,a)$. Define the bonus term $b_k^{\mathbb{P}}(\tau) = \sum_{(s,a) \in \tau} b_{\mathbb{P},k}(s,a)$.

**New Idea III: Scaled Confidence-Based Weights for Reducing Biases and Optimism-in-the-Face-of-Policy-Uncertainty (Illustrated in Fig. 2c).** Based on Idea I and Idea II, we are ready to construct optimistic $\hat{Q}$-value function. However, when the ground-truth reward function is not in the candidate set, an additional non-negligible regret will be incurred, e.g., simply applying online ridge regression over all collected samples could result in a regret that grows linearly in a constant error times $O(\sqrt{T})$ (He et al., 2022). One existing solution is to assign a weight $w_k$ to each selected action. The key idea is to assign a small weight to it to avoid the potentially large sub-regret, e.g.,

$$\hat{P}_k = \arg\min_{P' \in \mathbb{P}_{k-1}} \sum_{t=1}^{k-1} \sum_{\substack{\tau \in \hat{\Gamma}_{t|t-1}, \\ h \in [H]}} \frac{\left(\langle P'(\cdot|s_{t,h},a_{t,h}),V_{t,h}\rangle - V_{k,h}(s_{t,h+1})\right)^2}{\max\left\{1, \Lambda_t^P(\theta)/\left|\langle P'(\cdot|s_{t,h},a_{t,h}) - \hat{P}_t(\cdot|s_{t,h},a_{t,h}),V_{t,h}\rangle\right|\right\}}, \quad (14)$$

where $\Lambda_t^P(\theta) \triangleq \theta\sqrt{\lambda + \sum_{i=1}^{t-1}\sum_{\tau \in \Gamma_{i|i-1}}\left(\left\langle P'(\cdot \mid s_{i,h},a_{i,h}) - \hat{P}_i(\cdot \mid s_{i,h},a_{i,h}),V_{i,h}\right\rangle\right)^2}$ is the weight to normalize the traditional regression error for stability. Then, the confidence set will be

$$\mathbb{P}_k = \left\{P' \in \mathbb{P}_{k-1} \mid \lambda + \sum_{t\in[k-1]}\sum_{\substack{\tau \in \Gamma_{t|t-1}, \\ h\in[H]}} \frac{\left(\langle P'(\cdot|s_{t,h},a_{t,h})-\hat{P}_k(\cdot|s_{t,h},a_{t,h}),V_{t,h}\rangle\right)^2}{\max\left\{1,\Lambda_t^P(\theta)/\left|\langle[P'-\hat{P}_t](\cdot|s_{t,h},a_{t,h}),V_{t,h}\rangle\right|\right\}} \leq \beta^P\right\}. \quad (15)$$

However, this idea is not directly applicable in our case with inconsistent multi-agent feedback, because simply adding weights to the action does not help to explore the ground truth that is an outlier. To address this new issue, we choose the weight as a scaled inverse exploration confidence,

$$w_k = \max\left\{1, \theta\sqrt{\lambda + \sum_{t=1}^{k-1}\left(R(\tau_t) - \hat{R}_k(\tau_t)\right)^2} \Big/ \left|R(\tau_k) - \hat{R}_k(\tau_k)\right|\right\}, \quad (16)$$

where $\theta > 0$ is a tunable parameter. Moreover, since the absolute reward for each state-action pair is not available in RLHF, we cannot get an optimistic $\hat{Q}$-value function. Instead, we construct the optimistic policy set. With the confidence set and bonus terms, we construct the following set $\mathcal{S}_k$:

$$\mathcal{S}_k = \left\{\pi \mid \mathbb{E}_{\tau\sim(\hat{\mathbb{P}}_k,\pi)}\left[\sigma\left(\tau,\tau_0 \mid \hat{\mathcal{R}}_k\right) + b_k^{\mathcal{R}}\left(\tau,\tau_0\right) + b_k^{\mathbb{P}}(\tau)\right] \geq 0, \forall\pi_0 \in \Pi\right\}, \quad (17)$$

where $\Pi$ is a set containing all history-dependent policies. Intuitively, $\mathcal{S}_k$ consists of policies such that no other policy outperforms it. Finally, we choose a policy that maximizes uncertainty,

$$\pi_k = \underset{\left\{\pi\mid\mathbb{E}_{\tau\sim(\hat{P}_k,\pi)}[\sigma(\tau|\hat{R}_k)+b_k^R(\tau)+b_k^P(\tau)]\geq 0\right\}}{\arg\max} \mathbb{E}_{\tau\sim(\hat{P}_k,\pi)}\left(\sqrt{\beta^R}b_k^R(\tau) + \sqrt{\beta^P}b_k^P(\tau)\right). \quad (18)$$

## 4 THEORETICAL ANALYSIS

In this section, we focus on discussing about new difficulties in the regret analysis of our setting with inconsistent multi-agent feedback. Due to page limits, please see Appendix B for details.

**Theorem 1.** *Let* $\alpha \in (0,\xi)$, $C_1(k,\xi) = 2\left(\xi^2 + 2k + 3\ln(2/\delta)\right)$, $\beta_k^R \geq \tilde{O}\left(\left(\ln\left(H\mathcal{N}_K(\epsilon,\alpha)/\delta\right) + \xi\sup_{t<k}\beta_t^R + \left(\sup_t \beta_t^R\right)^2 K + \sup_t \beta_t^R\sqrt{KC_1(k,\xi)}\right)^{1/2}\right)$, *and* $\beta_k^P \geq \tilde{O}\left(\ln\left(H\mathcal{N}_K(\epsilon,\alpha)/\delta\right) + \xi\sup_{t<k}\beta_t^R + \left(\sup_t\beta_t^R\right)^2 K + \sup_t\beta_t^R\sqrt{KC_1(k,\xi)}\right)^{1/2}$ *for all* $k \in [K]$, *then with probability* $1 - 2\delta$, *the regret of RLHF-IMAF is upper-bounded as follows,*

$$\text{Reg}^{RLHF\text{-}IMAF}(K) \leq \tilde{\mathcal{O}}\left(\sqrt{\frac{HK}{M}\ln\left(\mathcal{N}_K(\epsilon,\alpha)\right)\dim_E(\mathcal{R},\epsilon/K)} + \xi\left(\dim_E(\mathcal{R},\epsilon/K)\right)\right). \quad (19)$$

Our regret analysis reveals the following: (i) The regret decreases with $M$, indicating that having more feedback sources is generally beneficial, even in the presence of inconsistency. This highlights

the utility of multi-agent feedback in improving performance. (ii) However, the regret also includes a term dependent on $\xi$ that does not decrease with $M$. This indicates that while increasing $M$ can mitigate some effects of inconsistency, if the feedback quality is consistently poor (i.e., high $\xi$), part of the overall regret remains significant regardless of $M$. Thus, the benefit of additional feedback is *limited* by its quality. (iii) The regret depends on $\alpha$, i.e., the Steiner point constant. Thus, there is a fundamental trade-off between complexity and the regret performance.

*Proof Sketch:* Due to the three new ideas in our algorithm design, there are three main steps.

First, we need to show the impact of inconsistency resolved by constructing a Steiner-point-based confidence center. Specifically, the bonus parameter $\beta_k^R$ depends on $\mathcal{N}_K(\epsilon, \alpha) \triangleq \mathcal{N}(\mathcal{R}, \epsilon, \|\cdot\|_\infty) \cdot \mathcal{N}(\mathcal{S}_\alpha \times \mathcal{A}_\alpha, \epsilon, \|\cdot\|_\infty)$, which captures the covering over the new function space with regard to the $\alpha$-Steiner points. Thus, with high probability at least $1 - \delta$, where $\delta \in (0, 1)$, we have $R^*(\cdot) \in \mathcal{R}_k$. Note that $\mathcal{S}_\alpha$ and $\mathcal{A}_\alpha$ represents the Steiner-point-based state space and action space, respectively, and they are constructed based on the aforementioned construction for the Steiner-point-based confidence set, as well as the transition kernel. See Appendix B.1 for details.

Second, we need to derive the sub-regret based on the gap incurred by sub-importance sampling and the resulting bonus terms. To capture this, we extend the idea in Wang et al. (2020) (see discussions in Appendix D) to capture our new sub-importance sampling method, i.e., we show that $-\xi_k \leq V_{k,1}(\tau_0 \mid P) - V_{k,1}(\tau_0 \mid P^*) \leq 2\sqrt{\beta^P} b_k^P(s, a) + \xi_k$. This captures the gap due to the error in sub-importance sampling for the comparison feedback. See Appendix B.2 for details.

Third, since we design scaled confidence-based weights for reducing biased in each agent feedback, we need to derive the final regret based on a deforming indicator function and the threshold-based bonus values (see discussions in Appendix E). Specifically, we decompose the regret as follows, $(\sigma(\tau \mid R) \triangleq \sigma(R(\tau) - R(\tau_0))$ with slight abuse of notation)

$$
\begin{aligned}
\text{Reg}^{\text{RLHF-IMAF}}(K) = \sum_{k=1}^K &\left( \mathbb{E}_{\tau^* \sim (\hat{\mathbb{P}}_k, \pi^*)} \sigma(\tau^* \mid R) - \mathbb{E}_{\tau_k \sim (\hat{\mathbb{P}}_k, \pi_k)} \sigma(\tau_k \mid R) \right) \\
&+ \sum_{k=1}^K \left( \mathbb{E}_{\tau^* \sim (\mathbb{P}^*, \pi^*)} \sigma(\tau^* \mid R^*) - \mathbb{E}_{\tau_k \sim (\mathbb{P}^*, \pi_k)} \sigma(\tau_k \mid R^*) \right) \\
&- \sum_{k=1}^K \left( \mathbb{E}_{\tau^* \sim (\hat{\mathbb{P}}_k, \pi^*)} \sigma(\tau^* \mid R^*) - \mathbb{E}_{\tau_k \sim (\hat{\mathbb{P}}_k, \pi_k)} \sigma(\tau_k \mid R^*) \right) \\
&+ \sum_{k=1}^K \left( \mathbb{E}_{\tau^* \sim (\hat{\mathbb{P}}_k, \pi^*)} \sigma(\tau^* \mid R^*) - \mathbb{E}_{\tau_k \sim (\hat{\mathbb{P}}_k, \pi_k)} \sigma(\tau_k \mid R^*) \right) \\
&- \sum_{k=1}^K \left( \mathbb{E}_{\tau^* \sim (\hat{\mathbb{P}}_k, \pi^*)} \sigma(\tau^* \mid R) - \mathbb{E}_{\tau_k \sim (\hat{\mathbb{P}}_k, \pi_k)} \sigma(\tau_k \mid R) \right).
\end{aligned} \tag{20}
$$

Then, we bound the first term, second and third terms, fourth and fifth terms on the right-hand side one-by-one. The first term captures the gap due to the Steiner point in estimating the confidence center. The second and third terms capture the gap due to scaled confidence-based weights for optimistic exploration. The fourth and fifth terms capture the gap due to sub-importance sampling of the trajectories. See Appendix B.3 for details. After bounding these terms by the corresponding bonus terms and eluder analysis, the final regret will then follow. $\qquad\square$

## 5 CONCLUSION

This paper studies RLHF with inconsistent multi-agent feedback under general function approximation from a theoretical point of view. In summary, the inconsistency in agent/human feedback can result in suboptimal outcomes, especially when feedback comes from diverse agents. To address this gap, this paper presents the first effort to explore a more realistic setting of RLHF, where feedback is provided by multiple agents with differing reward functions. We propose a novel algorithm designed to manage inconsistent multi-agent feedback, introducing a Steiner-Point-based confidence set to harness the advantages of multiple sources of feedback and a weighted importance sampling technique to handle the complexity of inconsistency. Our theoretical contributions demonstrate the optimality of this approach and highlight, for the first time, the significant implications and potential of inconsistent multi-agent feedback in RLHF. Since this work only study the case with one single ground-truth reward function, it would be interesting to extend our results to the case with multiple (personalized) ground-truth to handle the preference of users. It would also be important to consider more general form of inconsistency.

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

## A  MORE RELATED WORK

Reinforcement Learning with Human Feedback (RLHF) has gained substantial attention as an approach to align machine learning models with human values and preferences. Early works, such as Knox & Stone (2011)'s exploration of incorporating human feedback into reinforcement learning agents, established foundational methods for improving learning efficiency through interactive feedback mechanisms. A significant breakthrough was achieved by (Christiano et al., 2017), who introduced techniques for scaling human feedback to deep reinforcement learning, enabling the training of more complex models through reward learning. Further developments included studies by Stiennon et al. (2020), who demonstrated how RLHF could be applied to tasks like summarization, optimizing model outputs through iterative human feedback loops.

In recent years, advancements have focused on the robustness and scalability of RLHF systems. For instance,Hwang et al. (2023) proposed sequential preference ranking to enhance feedback efficiency in complex tasks. Concurrently,Casper et al. (2023) identified open challenges in RLHF, such as balancing the trade-offs between automation and human involvement, and ensuring scalability to real-world applications. Additionally,Kaufmann et al. (2023) surveyed approaches for learning reward models from human feedback, emphasizing the shift towards robust policy training over direct reward optimization. Emerging research also explores AI-assisted feedback mechanisms to augment human inputs.Lee et al. (2023) demonstrated that integrating AI feedback with RLHF could maintain model alignment with human values while improving efficiency. Liu (2023)'s work on transforming human interactions via RLHF highlighted the potential for this methodology in ethical AI and social robotics. More recently, RLHF has also been extensively studied, e.g., in Wang et al. (2023); Zhu et al. (2023); Chakraborty et al. (2024); Ye et al. (2024); Chen et al. (2022); Chatterji et al. (2021); Kaufmann et al. (2023); Li et al. (2023); Du et al. (2024), and references therein.

Overall, RLHF continues to evolve as a pivotal framework for creating systems that reflect human intent, fostering advancements in areas such as robotics, natural language processing, and ethical AI. Further research into scalable architectures, enhanced feedback modalities, and cross-domain applications promises to extend its impact across AI-driven industries.

Research has also explored broader preference structures beyond the reward-based paradigm, e.g., in Munos et al. (2023); Rosset et al. (2024); Swamy et al. (2024); Ye et al. (2024), and techniques for post-processing models (Lin et al., 2023; Zheng et al., 2024). Direct preference learning has notably advanced RLHF, particularly in the post-training of open-source models. Following these advancements, recent studies, e.g., (Guo et al., 2024b; Liu et al., 2024; Meng et al., 2024; Tajwar et al., 2024; Xie et al., 2024), have demonstrated the effectiveness of on-policy sampling and online exploration in improving direct preference learning. In particular, online iterative DPO (Xiong et al., 2024; Xu et al., 2023) and its variants, e.g., Chen et al. (2024); Rosset et al. (2024), have achieved state-of-the-art results. Moreover, robust learning is also one related direction studying the corruption/imperfection in the feedback, e.g., in He et al. (2022); Ye et al. (2023); Wei et al. (2022); Wang et al. (2020); Kong et al. (2021); Yan et al. (2024), and references therein.

## B  PROOF FOR THEOREM 1

Our regret proof involves three important steps, which are related to the three new ideas in our algorithm design, detailed as follows. First, since in our new Idea I in the algorithm design we construct the confidence set based on the Steiner point technique, in Step I below (Appendix B.1), we derive the confidence radius and construct the high-probability event that are related to the impact of Steiner point in historical sample sets and bonus term values. Second, since in our new Idea II in the algorithm design we design a sub-importance sampling method for reducing the complexity in the function space, in Step II below (Appendix B.2), we derive the sub-regret based on the gap incurred by such sub-importance sampling and the resulting bonus terms. Third, since in our new Idea III in the algorithm design we design scaled confidence-based weights for reducing biased in each agent feedback, in Step III below (Appendix B.3), based on Step I and Step II, we derive the final regret based on a deforming indicator function and the threshold-based bonus values.

### B.1 STEP I: STEINER-POINT-BASED HIGH PROBABILITY EVENTS

In Step I, we first derive the confidence radius for both the reward confidence set $\mathcal{R}_k$ and the transition confidence set $\mathbb{P}_k$ in Algorithm 1. Because the absolute reward value is unavailable, we cannot construct high probability events for the $V$-value function any more. However, based on these, we can still construct a high probability event directly for the uncertain policies.

#### B.1.1 HIGH PROBABILITY EVENT FOR THE REWARD FUNCTION

**Lemma 1.** *For all* $(k) \in [K]$*, if for all* $k > 0$*, we let* $\beta_{k,H+1} = 0$ *and from* $h = H$ *to* $h = 1$*,*

$$
\beta_k^R \geq \left( 12\lambda + 12\ln\left(2H\mathcal{N}_K(\epsilon,\alpha)/\delta\right) + 12\gamma\xi \sup_{t<k} \beta_t^R + 12\left(5 \sup_t \beta_t^R \gamma\right)^2 K + 60 \sup_t \beta_t^R \gamma \sqrt{KC_1(k,\xi)} \right)^{1/2},
$$
(21)

*where* $\mathcal{N}_K(\epsilon,\alpha) = \mathcal{N}\left(\mathcal{R},\epsilon,\|\cdot\|_\infty\right) \cdot \mathcal{N}\left(\mathcal{S}_\alpha \times \mathcal{A}_\alpha,\epsilon,\|\cdot\|_\infty\right)$ *and* $C_1(k,\xi) = 2\left(\xi^2 + 2k + 3\ln(2/\delta)\right)$*, then with high probability at least* $1 - \delta$*, where* $\delta \in (0,1)$*, we have* $R^*(\cdot) \in \mathcal{R}_k$*.*

*Proof.* To prove $R^*(\cdot) \in \mathcal{R}_k$ with probability at least $1 - \delta$, we prove that with probability at least $1 - \delta$, we have for all $k \in [K]$,

$$
\lambda + \sum_{t=1}^{k-1} \sum_{\tau \in \hat{\Gamma}_{t|t-1}} \frac{\left(\sigma\left(\tau \mid R^*\right) - \sigma\left(\tau \mid \hat{R}_k\right)\right)^2}{\max\left\{1, \Lambda_t(\theta)/\left|\sigma(\tau \mid R^*) - \sigma(\tau \mid \hat{R}_t)\right|\right\}} \leq \beta^R,
$$
(22)

by mathematical induction.

Base case: First, we have that Eq. (22) trivially holds for episode $k = 1$.

Hypothesis: Then, for episode $k > 1$, we assume that Eq. (22) holds for all episode $t \leq k-1$, which means that for all episodes $t \in [k-1]$,

$$
\lambda + \sum_{i=1}^{t-1} \sum_{\tau \in \hat{\Gamma}_{i|i-1}} \frac{\left(\sigma\left(\tau \mid R^*\right) - \sigma\left(\tau \mid \hat{R}_t\right)\right)^2}{\max\left\{1, \Lambda_i(\theta)/\left|\sigma(\tau \mid R^*) - \sigma(\tau \mid \hat{R}_i)\right|\right\}} \leq \beta^R.
$$
(23)

Induction: Thus, for episode $k$, we let $\mathcal{R}_k^{\epsilon,\sigma}$ be a $\epsilon$-covering set of $\mathcal{R}_k$ under the $\|\cdot\|_\infty$ norm. Then, we construct $\overline{\mathcal{R}}_k^{\epsilon,\sigma} = \mathcal{R}_k^{\epsilon,\sigma} \oplus \beta^R \mathcal{B}_k^{\epsilon,\sigma}$ as a $\left(1 + \beta^R\right) \epsilon$-covering set of $\mathcal{R}_k^{\epsilon,\sigma}$ under the $\|\cdot\|_\infty$ norm, where $\mathcal{B}_k^{\epsilon,\sigma}$ is the bonus function space which can be relaxed under our sub-importance sampling idea (i.e., represented by the sum over $\tau \in \Gamma_{i|i-1}$), and note that the covering set depends on the link function $\sigma$. Thus, to compare with $R^*$, let $\bar{R}_k \in \overline{\mathcal{R}}_k^{\epsilon,\sigma}$ so that $\left\|\sigma(\bar{R}_k\left(\cdot\right) - \bar{R}_k\left(\tau_0\right)) - \sigma(R_t^*\left(\cdot\right) - R_t^*\left(\tau_0\right))\right\|_\infty \leq \bar{\epsilon} = \left(1 + \beta^R\right)\epsilon$. Then, by letting

$$
\tilde{R}_k = \underset{R \in \mathcal{R}_{k-1}}{\arg\min} \sum_{t=1}^{k-1} \sum_{\tau \in \hat{\Gamma}_{t|t-1}} \left(\sigma(R\left(\tau\right) - R\left(\tau_0\right)) - \sigma(\bar{R}_t(\tau) - \bar{R}_t(\tau_0))\right)^2.
$$
(24)

we have that

$$\left(\sum_{t=1}^{k-1} \sum_{\tau \in \hat{\Gamma}_{t|t-1}} \left(\sigma(\hat{R}_k\left(\tau\right) - \hat{R}_k\left(\tau_0\right)) - \sigma(\bar{R}_t(\tau) - \bar{R}_t(\tau_0))\right)^2\right)^{1/2}$$

$$\leq \left(\sum_{t=1}^{k-1} \sum_{\tau \in \hat{\Gamma}_{t|t-1}} \left(\sigma(\hat{R}_k\left(\tau\right) - \hat{R}_k\left(\tau_0\right)) - \sigma(R_t^*(\tau) - R_t^*(\tau_0))\right)^2\right)^{1/2} + \sqrt{k}\bar{\epsilon}$$

$$\leq \left(\sum_{t=1}^{k-1} \sum_{\tau \in \hat{\Gamma}_{t|t-1}} \left(\sigma(\tilde{R}_k\left(\tau\right) - \tilde{R}_k\left(\tau_0\right)) - \sigma(R_t^*(\tau) - R_t^*(\tau_0))\right)^2\right)^{1/2} + \sqrt{k}\bar{\epsilon}$$

$$\leq \left(\sum_{t=1}^{k-1} \sum_{\tau \in \hat{\Gamma}_{t|t-1}} \left(\sigma(\tilde{R}_k\left(\tau\right) - \tilde{R}_k\left(\tau_0\right)) - \sigma(\bar{R}_t(\tau) - \bar{R}_t(\tau_0))\right)^2\right)^{1/2} + 2\sqrt{k}\bar{\epsilon}, \quad (25)$$

where the first and third inequality is obtained by applying $\left\|\sigma(\bar{R}_k\left(\cdot\right) - \bar{R}_k\left(\tau_0\right)) - \sigma(R_t^*(\cdot) - R_t^*(\tau_0))\right\|_\infty \leq \bar{\epsilon} = \left(1 + \beta^R\right)\epsilon$.

Finally, we leverage the relation between $\sum_{t=1}^{k-1} \sum_{\tau \in \hat{\Gamma}_{t|t-1}} \left(\sigma(\hat{R}_k\left(\tau_t\right) - \hat{R}_k\left(\tau_0\right)) - \sigma(\bar{R}_t(\tau_t) - \bar{R}_t(\tau_0))\right)^2$ and $\sum_{t=1}^{k-1} \sum_{\tau \in \hat{\Gamma}_{t|t-1}} \left(\sigma(\tilde{R}_k\left(\tau_t\right) - \tilde{R}_k\left(\tau_0\right)) - \sigma(\bar{R}_t(\tau_t) - \bar{R}_t(\tau_0))\right)^2$ above to complete the induction step. Specifically, consider a function space $\overline{\mathcal{R}}_k^{\epsilon,\sigma} : \hat{\Gamma} \to \mathbb{R}$ and filtered sequence $\{\tau_k, \eta_k\}$ in $\hat{\Gamma} \times \mathbb{R}$, such that, $\eta_k$ is conditionally zero-mean $G$-sub-Gaussian noise. For $R^*(\cdot) : \hat{\Gamma} \to \mathbb{R}$, suppose that $f_k = \sigma(R^*\left(\tau_k\right) - R^*\left(\tau_0\right)) + \eta_k$ and there exists a function $\bar{R}_t \in \overline{\mathcal{R}}_k^{\epsilon,\sigma}$, such that, for any $k \in [K], \sum_{t=1}^{k} |\sigma(R^*\left(\tau_t\right) - R^*\left(\tau_0\right)) - \sigma(R_t\left(\tau_t\right) - R_t\left(\tau_0\right))| \leq \zeta$. If $\hat{R}_k$ is an approximate empirical risk minimization solution up to some $\epsilon' \geq 0$, i.e.,

$$\left(\sum_{t=1}^{k} \sum_{\tau \in \hat{\Gamma}_{t|t-1}} \frac{\left(\sigma(\hat{R}_k\left(\tau\right) - \hat{R}_k\left(\tau_0\right)) - f_t\right)^2}{\max\left\{1, \Lambda_t(\theta)/\left|\sigma(\tau \mid \hat{R}_t) - f_t\right|\right\}}\right)^{1/2}$$

$$\leq \min_{R \in \mathcal{R}_{k-1}} \left(\sum_{t=1}^{k} \sum_{\tau \in \hat{\Gamma}_{t|t-1}} \frac{\left(\sigma(R\left(\tau\right) - R\left(\tau_0\right)) - f_t\right)^2}{\max\left\{1, \Lambda_t(\theta)/\left|\sigma(\tau \mid \hat{R}_t) - f_t\right|\right\}}\right)^{1/2} + \sqrt{k}\epsilon', \quad (26)$$

with probability at least $1 - \delta$, then we have for all episodes $k \in [K]$,

$$\left(\sum_{t=1}^{k} \sum_{\tau \in \hat{\Gamma}_{t|t-1}} \frac{\left(\sigma(\hat{R}_k\left(\tau_t\right) - \hat{R}_k\left(\tau_0\right)) - \sigma(\bar{R}_t(\tau_t) - \bar{R}_t(\tau_0))\right)^2}{\max\left\{1, \Lambda_t(\theta)/\left|\sigma(\tau \mid \bar{R}) - \sigma(\tau \mid \hat{R}_t)\right|\right\}}\right)^{1/2}$$

$$\leq 10\eta^2 \ln\left(2\mathcal{N}\left(\overline{\mathcal{R}}_k^{\epsilon,\sigma}, \epsilon, \|\cdot\|_\infty\right)/\delta\right)$$

$$+ 5\sum_{t=1}^{k} \sum_{\tau \in \hat{\Gamma}_{t|t-1}} \frac{\left|\sigma(\hat{R}_t\left(\tau_t\right) - \hat{R}_t\left(\tau_0\right)) - \sigma(\bar{R}\left(\tau_t\right) - \bar{R}\left(\tau_0\right))\right| \xi_t}{\max\left\{1, \Lambda_t(\theta)/\left|\sigma(\tau \mid \bar{R}) - \sigma(\tau \mid \hat{R}_t)\right|\right\}}$$

$$+ 10\left(\gamma + \epsilon'\right)\left(\left(\gamma + \epsilon'\right)k + \sqrt{kC_1(k,\xi)}\right), \quad (27)$$

where $C_1(k,\xi) = 2\left(\xi^2 + 2kG^2 + 3G^2\ln(2/\delta)\right)$. The reason is as follows. For $R \in \overline{\mathcal{R}}_k^{\epsilon,\sigma}$, we define $\phi\left(R, \tau_k\right) = -a\left[\left(\sigma\left(\tau_k \mid R\right) - f_k\right)^2 - \left(\sigma\left(\tau_k \mid \bar{R}\right) - f_k\right)^2\right]/\max\left\{1, \Lambda_k(\theta)/\left|\sigma(\tau \mid \bar{R}) - \sigma(\tau \mid \hat{R}_k)\right|\right\}$, where $a = \frac{G^{-2}}{4}$. Let $\mathcal{R}^\epsilon$ be an $\epsilon$-cover of $\mathcal{R}$ under the $\|\cdot\|_\infty$ norm. Denote the cardinality of $\mathcal{R}^\epsilon$ by

$\mathcal{N} = \mathcal{N}(\mathcal{R}, \epsilon, \|\cdot\|_\infty)$. Since $\epsilon_k$ is conditional $G$-sub-Gaussian and $\phi(R, \tau_k)$ can be written as

$$\phi(R, \tau_k) = 2a\left[\sigma(\tau_k \mid R) - \sigma(\tau_k \mid \bar{R})\right] / \max\left\{1, \Lambda_k(\theta)/\left|\sigma(\tau \mid \bar{R}) - \sigma(\tau \mid \hat{R}_k)\right|\right\} \cdot \epsilon_k$$

$$- a\left[\sigma(\tau_k \mid R) - \sigma(\tau_k \mid \bar{R})\right]^2 / \max\left\{1, \Lambda_k(\theta)/\left|\sigma(\tau \mid \bar{R}) - \sigma(\tau \mid \hat{R}_k)\right|\right\}$$

$$+ 2a\left[\sigma(\tau_k \mid R) - \sigma(\tau_k \mid \bar{R})\right] / \max\left\{1, \Lambda_k(\theta)/\left|\sigma(\tau \mid \bar{R}) - \sigma(\tau \mid \hat{R}_k)\right|\right\} \xi, \tag{28}$$

and $\phi(R, \tau_k)$ is conditional $2aG\left[\sigma(\tau_k \mid R) - \sigma(\tau_k \mid \bar{R})\right] / \max\left\{1, \Lambda_k(\theta)/\left|\sigma(\tau \mid \bar{R}) - \sigma(\tau \mid \hat{R}_k)\right|\right\}$-sub-Gaussian with mean

$$\mu = -a\left[\sigma(\tau_k \mid R) - \sigma(\tau_k \mid \bar{R})\right]^2 / \max\left\{1, \Lambda_k(\theta)/\left|\sigma(\tau \mid \bar{R}) - \sigma(\tau \mid \hat{R}_k)\right|\right\}$$

$$+ 2a\xi\left[\sigma(\tau_k \mid R) - \sigma(\tau_k \mid \bar{R})\right] / \max\left\{1, \Lambda_k(\theta)/\left|\sigma(\tau \mid \bar{R}) - \sigma(\tau \mid \hat{R}_k)\right|\right\}, \tag{29}$$

where $a = \frac{G^{-2}}{4}$. According to Lemma 5, if a variable $X$ is $\sigma$-sub-Gaussian with mean $\mu$ conditional on $\mathcal{S}$, the property of sub-Gaussianity implies that

$$\ln \mathbb{E}[\exp(s(X - \mu)) \mid \mathcal{S}] \le \frac{\sigma^2 s^2}{2}. \tag{30}$$

By taking $s = 1$ in the inequality above, we get

$$\ln \mathbb{E}_{f_k}\left[\exp\left(\phi(R, \tau_k) - \mu\right) \mid \tau_k, \Gamma_{k-1}\right] \le \frac{4a^2 G^2 \left[\sigma(\tau_k \mid R) - \sigma(\tau_k \mid \bar{R})\right]^2}{2\max\left\{1, \Lambda_k(\theta)/\left|\sigma(\tau \mid \bar{R}) - \sigma(\tau \mid \hat{R}_k)\right|\right\}^2}$$

$$= \frac{\left[\sigma(\tau_k \mid R) - \sigma(\tau_k \mid \bar{R})\right]^2}{8G^2 \max\left\{1, \Lambda_k(\theta)/\left|\sigma(\tau \mid \bar{R}) - \sigma(\tau \mid \hat{R}_k)\right|\right\}^2}. \tag{31}$$

It follows that

$$\ln \mathbb{E}_{f_k}\left[\exp\left(\phi(R, \tau_k)\right) \mid \tau_k, \Gamma_{t-1}\right]$$

$$\le \frac{\left[\sigma(\tau_k \mid R) - \sigma(\tau_k \mid \bar{R})\right]^2}{8G^2 \max\left\{1, \Lambda_k(\theta)/\left|\sigma(\tau \mid \bar{R}) - \sigma(\tau \mid \hat{R}_k)\right|\right\}^2} - \frac{\left[\sigma(\tau_k \mid R) - \sigma(\tau_k \mid \bar{R})\right]^2}{4G^2 \max\left\{1, \Lambda_k(\theta)/\left|\sigma(\tau \mid \bar{R}) - \sigma(\tau \mid \hat{R}_k)\right|\right\}}$$

$$+ \frac{\xi_k \left[\sigma(\tau_k \mid R) - \sigma(\tau_k \mid \bar{R})\right]^2}{2G^2 \max\left\{1, \Lambda_k(\theta)/\left|\sigma(\tau \mid \bar{R}) - \sigma(\tau \mid \hat{R}_k)\right|\right\}}$$

$$\le -\frac{\left[\sigma(\tau_k \mid R) - \sigma(\tau_k \mid \bar{R})\right]^2}{8G^2 \max\left\{1, \Lambda_k(\theta)/\left|\sigma(\tau \mid \bar{R}) - \sigma(\tau \mid \hat{R}_k)\right|\right\}}$$

$$+ \frac{\xi_k \left[\sigma(\tau_k \mid R) - \sigma(\tau_k \mid \bar{R})\right]^2}{2G^2 \max\left\{1, \Lambda_k(\theta)/\left|\sigma(\tau \mid \bar{R}) - \sigma(\tau \mid \hat{R}_k)\right|\right\}}, \tag{32}$$

where the second inequality is because $\max\left\{1, \Lambda_k(\theta)/\left|\sigma(\tau \mid \bar{R}) - \sigma(\tau \mid \hat{R}_k)\right|\right\} \ge 1$. According to Lemma 4 with $\lambda = 1$, we have for all $R \in \mathcal{R}^\epsilon$ and $k \in [K]$, with probability at least $1 - \delta/2$,

$$\sum_{t=1}^k \phi(R, \tau_t) \le -\sum_{t=1}^k \frac{\left[\sigma(\tau_k \mid R) - \sigma(\tau_k \mid \bar{R})\right]^2}{8G^2 \max\left\{1, \Lambda_k(\theta)/\left|\sigma(\tau \mid \bar{R}) - \sigma(\tau \mid \hat{R}_k)\right|\right\}}$$

$$+ \sum_{t=1}^k \frac{\left[\sigma(\tau_k \mid R) - \sigma(\tau_k \mid \bar{R})\right]^2 \xi}{2G^2 \max\left\{1, \Lambda_k(\theta)/\left|\sigma(\tau \mid \bar{R}) - \sigma(\tau \mid \hat{R}_k)\right|\right\}} + \ln(2\mathcal{N}/\delta). \tag{33}$$

Additionally, for all episode $k \in [K]$, we have with probability at least $1 - \delta/2$,

$$
\begin{aligned}
\sum_{t=1}^{k} \left( \sigma \left( \tau_t \mid \bar{R} \right) - f_t \right)^2 &\leq \sum_{t=1}^{k} \left( \sigma \left( \tau_t \mid \bar{R} \right) - \sigma \left( \tau_t \mid R^* \right) + \sigma \left( \tau_t \mid R^* \right) - f_t \right)^2 \\
&\leq 2 \sum_{t=1}^{k-1} \left( \left( \sigma \left( \tau_t \mid \bar{R} \right) - \sigma \left( \tau_t \mid R^* \right) \right)^2 + \left( \sigma \left( \tau_t \mid R^* \right) - f_t \right)^2 \right) \\
&\leq 2 \left( \sum_{t=1}^{k-1} \xi_t^2 + \sum_{t=1}^{k-1} \epsilon_t^2 \right) \\
&\leq 2 \left( \xi^2 + 2kG^2 + 3G^2 \ln(2/\delta) \right),
\end{aligned}
\tag{34}
$$

where the first inequality is obtained since Cauchy-Schwarz inequality and the last inequality is due to Lemma 4. Now, given $\hat{R}_k$, there exists $R \in \overline{\mathcal{R}}_k^{\epsilon, \sigma}$, such that $\left\| \hat{R}_k - R \right\|_\infty \leq \epsilon$. With probability at least $1 - \delta/2$,

$$
\begin{aligned}
&\sum_{t=1}^{k} \left[ \left( \sigma \left( \tau_t \mid R \right) - f_t \right)^2 - \left( \sigma \left( \tau_t \mid \bar{R} \right) - f_t \right)^2 \right] / \max \left\{ 1, \Lambda_t(\theta) / \left| \sigma(\tau \mid \bar{R}) - \sigma(\tau \mid \hat{R}_t) \right| \right\} \\
&\leq \left( \sqrt{\sum_{t=1}^{k} \left( \sigma \left( \tau_t \mid \hat{R}_t \right) - f_t \right)^2 / \max \left\{ 1, \Lambda_t(\theta) / \left| \sigma(\tau \mid \bar{R}) - \sigma(\tau \mid \hat{R}_t) \right| \right\}} + \sqrt{k}\epsilon \right)^2 \\
&\quad - \sum_{t=1}^{k} \left( \sigma \left( \tau_t \mid \bar{R} \right) - f_t \right)^2 / \max \left\{ 1, \Lambda_t(\theta) / \left| \sigma(\tau \mid \bar{R}) - \sigma(\tau \mid \hat{R}_t) \right| \right\} \\
&\leq \left( \sqrt{\sum_{t=1}^{k} \left( \sigma \left( \tau_t \mid \bar{R}_t \right) - f_t \right)^2 / \max \left\{ 1, \Lambda_t(\theta) / \left| \sigma(\tau \mid \bar{R}) - \sigma(\tau \mid \hat{R}_t) \right| \right\}} + \sqrt{k}(\epsilon + \epsilon') \right)^2 \\
&\quad - \sum_{t=1}^{k} \left( \sigma \left( \tau_t \mid \bar{R} \right) - f_t \right)^2 / \max \left\{ 1, \Lambda_t(\theta) / \left| \sigma(\tau \mid \bar{R}) - \sigma(\tau \mid \hat{R}_t) \right| \right\} \\
&\leq \left( \epsilon + \epsilon' \right)^2 k + 2 \left( \epsilon + \epsilon' \right) \sqrt{k C_1(k, \xi)},
\end{aligned}
\tag{35}
$$

where the first inequality uses $\left|\sigma\left(\tau_t \mid R\right) - \sigma\left(\tau_t \mid \hat{R}\right)\right| \le \epsilon$ and triangle inequality for all $t$. Finally, with probability at least $1 - \delta$, we have

$$\left(\sum_{t=1}^{k}\left(\sigma\left(\tau_t \mid \hat{R}_t\right) - \sigma\left(\tau_t \mid \bar{R}\right)\right)^2 / \max\left\{1, \Lambda_t(\theta)/\left|\sigma(\tau \mid \bar{R}) - \sigma(\tau \mid \hat{R}_t)\right|\right\}\right)^{1/2}$$

$$\le \sqrt{\epsilon^2 k} + \left(\sum_{t=1}^{k}\left(\sigma\left(\tau_t \mid R\right) - \sigma\left(\tau_t \mid \bar{R}\right)\right)^2 / \max\left\{1, \Lambda_t(\theta)/\left|\sigma(\tau \mid \bar{R}) - \sigma(\tau \mid \hat{R}_t)\right|\right\}\right)^{1/2}$$

$$\le \sqrt{\epsilon^2 k} + \left(4\sum_{t=1}^{k}\left(\sigma\left(\tau_t \mid R\right) - \sigma\left(\tau_t \mid \bar{R}\right)\right)\xi_t / \max\left\{1, \Lambda_t(\theta)/\left|\sigma(\tau \mid \bar{R}) - \sigma(\tau \mid \hat{R}_t)\right|\right\}\right.$$

$$\left. + 8G^2\ln(2\mathcal{N}/\delta) - 8G^2\sum_{t=1}^{k}\phi\left(R, \tau_t\right)\right)^{1/2}$$

$$\le \sqrt{\epsilon^2 k} + \left(4\sum_{t=1}^{k}\left|\sigma\left(\tau_t \mid \hat{R}_t\right) - \sigma\left(\tau_t \mid \bar{R}\right)\right|\xi_t / \max\left\{1, \Lambda_t(\theta)/\left|\sigma(\tau \mid \bar{R}) - \sigma(\tau \mid \hat{R}_t)\right|\right\}\right.$$

$$\left. + 4\epsilon\xi + 8G^2\ln(2\mathcal{N}/\delta) + 2\left(\epsilon + \epsilon'\right)^2 t + 4\left(\epsilon + \epsilon'\right)\sqrt{k}C_1'(k, \xi)\right)^{1/2}$$

$$\le \left(10G^2\ln(2\mathcal{N}/\delta) + 5\sum_{t=1}^{k}\left|\sigma\left(\tau_t \mid \hat{R}_t\right) - \sigma\left(\tau_t \mid \bar{R}\right)\right|\xi_t / \max\left\{1, \Lambda_t(\theta)/\left|\sigma(\tau \mid \bar{R}) - \sigma(\tau \mid \hat{R}_t)\right|\right\}\right.$$

$$\left. + 5\epsilon\xi + 8\left(\epsilon + \epsilon'\right)^2 k + 5\left(\epsilon + \epsilon'\right)\sqrt{tC_1(k, \xi)}\right)^{1/2}, \tag{36}$$

where the second inequality is deduced from Eq. (33) and the last inequality uses Cauchy-Schwarz inequality.

Up to here, by letting $\epsilon' = 2\bar{\epsilon}, G = 1$ and adding the sum over only sub-sampling feedback $\Gamma_{t|t-1}$, and taking a union bound over $\bar{R}_\kappa \in \overline{\mathcal{R}}_k^{\epsilon,\sigma}$, we can have that with probability at least $1 - \delta$, the following inequality holds for all episodes $k \in [K]$ :

$$\sum_{t=1}^{k-1}\sum_{\tau \in \Gamma_{t|t-1}}\frac{\left(\sigma\left(\tau_t \mid \hat{R}_k\right) - \bar{R}_\kappa\left(\tau_t\right)\right)^2}{\max\left\{1, \Lambda_t(\theta)/\left|\sigma(\tau \mid \bar{R}) - \sigma(\tau \mid \hat{R}_t)\right|\right\}}$$

$$\le 10\ln\left(2H\mathcal{N}_K(\epsilon)/\delta\right) + 5\sum_{t=1}^{k-1}\sum_{\tau \in \Gamma_{t|t-1}}\frac{\left|\sigma\left(\tau_t \mid \hat{R}_k\right) - \sigma\left(\tau_t \mid \bar{R}_\kappa\right)\right| \cdot \xi_t}{\max\left\{1, \Lambda_t(\theta)/\left|\sigma(\tau \mid \bar{R}) - \sigma(\tau \mid \hat{R}_t)\right|\right\}}$$

$$+ 10(\epsilon + 2\bar{\epsilon}) \cdot \left(\left(\epsilon + 2\bar{\epsilon}\right)k + \sqrt{2k\left(\xi^2 + 2k + 3\ln(2/\delta)\right)}\right), \tag{37}$$

Further, for all episodes $t \le k - 1$, we have that

$$\left|\sigma\left(\tau_t \mid \hat{R}_k\right) - \sigma\left(\tau_t \mid \bar{R}_\kappa\right)\right| / \max\left\{1, \Lambda_t(\theta)/\left|\sigma(\tau \mid \bar{R}) - \sigma(\tau \mid \hat{R}_t)\right|\right\}$$

$$\le \left|\sigma\left(\tau_t \mid \hat{R}_k\right) - \sigma\left(\tau_t \mid \bar{R}_\kappa\right)\right| / \max\left\{1, \Lambda_t(\theta)/\left|\sigma(\tau \mid \bar{R}) - \sigma(\tau \mid \hat{R}_t)\right|\right\} + \epsilon$$

$$\le \frac{\left|\sigma\left(\tau_t \mid \hat{R}_k\right) - \sigma\left(\tau_t \mid \hat{R}_t\right)\right|}{\max\left\{1, \Lambda_t(\theta)/\left|\sigma(\tau \mid \bar{R}) - \sigma(\tau \mid \hat{R}_t)\right|\right\}} + \frac{\left|\sigma\left(\tau_t \mid \bar{R}_\kappa\right) - \sigma\left(\tau_t \mid \hat{R}_t\right)\right|}{\max\left\{1, \Lambda_t(\theta)/\left|\sigma(\tau \mid \bar{R}) - \sigma(\tau \mid \hat{R}_t)\right|\right\}} + \epsilon$$

$$\le 2\alpha\beta^R + \epsilon, \tag{38}$$

where the last inequality is due to $\hat{R}_k \in \mathcal{R}_{k-1} \subset \mathcal{R}_t$ and the induction hypothesis that $\bar{R}_\kappa \in \mathcal{R}_t$ for $\kappa \geq t$. Therefore, we have that, with probability at least $1 - \delta$,

$$
\left( \lambda + \sum_{t=1}^{k-1} \sum_{\tau \in \Gamma_{t|t-1}} \frac{\left( \sigma\left(\tau_t \mid R_k\right) - \sigma\left(\tau_t \mid \bar{R}_\kappa\right) \right)^2}{\max\left\{1, \Lambda_t(\theta)/\left|\sigma(\tau \mid \bar{R}) - \sigma(\tau \mid R_t)\right|\right\}} \right)^{1/2}
$$

$$
\leq \left( \sum_{t=1}^{k-1} \sum_{\tau \in \Gamma_{t|t-1}} \frac{\left( \sigma\left(\tau_t \mid \hat{R}_k\right) - \sigma\left(\tau_t \mid \bar{R}_\kappa\right) \right)^2}{\max\left\{1, \Lambda_t(\theta)/\left|\sigma(\tau \mid \bar{R}) - \sigma(\tau \mid \hat{R}_t)\right|\right\}} \right)^{1/2} + \sqrt{t}\bar{\epsilon} + \sqrt{\lambda}
$$

$$
\leq \left( 10\ln\left(2H\mathcal{N}_K(\epsilon)/\delta\right) + 10\alpha\xi \sup_{s<t} \beta_s^R + 5\epsilon\xi + 10\left(2\beta_\kappa^R + 3\right)^2 \epsilon^2 K + 10\left(2\beta_\kappa^R + 3\right)\gamma\sqrt{KC_1(k,\xi)} \right)^{1/2}
$$

$$
+ \left(\beta_\kappa^R + 1\right)\epsilon\sqrt{K} + \sqrt{\lambda}
$$

$$
\leq \left( 12\lambda + 12\ln\left(2H\mathcal{N}_K(\epsilon)/\delta\right) + 12\gamma\xi \sup_{t<k} \beta_s^R + 12\left(5\sup_s \beta_s^R \gamma\right)^2 K + 60\sup_s \beta_s^R \gamma\sqrt{KC_1(k,\xi)} \right)^{1/2}
$$

$$
\leq \beta_k^R, \tag{39}
$$

where the first inequality uses the triangle inequality and the second last inequality uses Cauchy-Schwarz inequality. Therefore, we validate the statement in Eq. (22). For all $k \in [K]$, by taking $\kappa = k$ in Eq. (22), we finally complete the proof.

$$\square$$

By Lemma 1, we know that the comparison based on ground-truth reward function $R^*(\cdot) \in \mathcal{R}_k$ with high probability.

### B.1.2 HIGH PROBABILITY EVENT FOR THE TRANSITION KERNEL

**Lemma 2.** *For all $(k) \in [K]$, if for all $k > 0$, we let $\beta_{k,H+1} = 0$ and from $h = H$ to $h = 1$,*

$$
\beta_k^{\mathbb{P}} \geq \left( 12\lambda + 12\ln\left(2H\mathcal{N}_K(\epsilon, \alpha)/\delta\right) + 12\gamma\xi \sup_{t<k} \beta_t^{\mathbb{P}} + 12\left(5\sup_t \beta_t^{\mathbb{P}} \gamma\right)^2 K + 60\sup_t \beta_t^{\mathbb{P}} \gamma\sqrt{KC_1(k,\xi)} \right)^{1/2},
$$
$$\tag{40}$$

*where $\mathcal{N}_K(\epsilon, \alpha) = \mathcal{N}\left(\mathcal{P}, \epsilon, \|\cdot\|_\infty\right) \cdot \mathcal{N}\left(\mathcal{S}_\alpha \times \mathcal{A}_\alpha, \epsilon, \|\cdot\|_\infty\right)$ and $C_1(k,\xi) = 2\left(\xi^2 + 2k + 3\ln(2/\delta)\right)$, then with high probability at least $1 - \delta$, where $\delta \in (0,1)$, we have $\mathbb{P}^*(\cdot) \in \mathcal{P}_k$.*

*Proof.* To prove $\mathbb{P}^*(\cdot) \in \mathcal{P}_k$ with probability at least $1 - \delta$, we prove that with probability at least $1 - \delta$, we have for all $k \in [K]$,

$$
\lambda + \sum_{t=1}^{k-1} \sum_{\tau \in \hat{\Gamma}_{t|t-1}} \frac{\left( \sigma\left(\tau \mid \mathbb{P}^*\right) - \sigma\left(\tau \mid \hat{\mathbb{P}}_k\right) \right)^2}{\max\left\{1, \Lambda_t(\theta)/\left|\sigma(\tau \mid \mathbb{P}^*) - \sigma(\tau \mid \hat{\mathbb{P}}_t)\right|\right\}} \leq \beta^{\mathbb{P}}, \tag{41}
$$

by mathematical induction.

Base case: First, we have that Eq. (41) trivially holds for episode $k = 1$.

Hypothesis: Then, for episode $k > 1$, we assume that Eq. (41) holds for all episode $t \leq k-1$, which means that for all episodes $t \in [k-1]$,

$$
\lambda + \sum_{i=1}^{t-1} \sum_{\tau \in \hat{\Gamma}_{i|i-1}} \frac{\left( \sigma\left(\tau \mid \mathbb{P}^*\right) - \sigma\left(\tau \mid \hat{\mathbb{P}}_t\right) \right)^2}{\max\left\{1, \Lambda_i(\theta)/\left|\sigma(\tau \mid \mathbb{P}^*) - \sigma(\tau \mid \hat{\mathbb{P}}_i)\right|\right\}} \leq \beta^{\mathbb{P}}. \tag{42}
$$

Induction: Thus, for episode $k$, we let $\mathcal{P}_k^{\epsilon,\sigma}$ be a $\epsilon$-covering set of $\mathcal{P}_k$ under the $\|\cdot\|_\infty$ norm. Then, we construct $\overline{\mathcal{P}}_k^{\epsilon,\sigma} = \mathcal{P}_k^{\epsilon,\sigma} \oplus \beta^{\mathbb{P}} \mathcal{B}_k^{\epsilon,\sigma}$ as a $\left(1+\beta^{\mathbb{P}}\right)\epsilon$-covering set of $\mathcal{P}_k^{\epsilon,\sigma}$ under the $\|\cdot\|_\infty$ norm, where $\mathcal{B}_k^{\epsilon,\sigma}$ is the bonus function space which can be relaxed under our sub-importance sampling idea (i.e., represented by the sum over $\tau \in \hat{\Gamma}_{i|i-1}$), and note that the covering set depends on the link function $\sigma$. Thus, to compare with $\mathbb{P}^*$, let $\bar{\mathbb{P}}_k \in \overline{\mathcal{P}}_k^{\epsilon,\sigma}$ so that $\left\|\sigma(\bar{\mathbb{P}}_k\left(\cdot\right) - \bar{\mathbb{P}}_k\left(\tau_0\right)) - \sigma(\mathbb{P}_t^*(\cdot) - \mathbb{P}_t^*(\tau_0))\right\|_\infty \leq \bar{\epsilon} = \left(1+\beta^{\mathbb{P}}\right)\epsilon$. Then, by letting

$$\tilde{\mathbb{P}}_k = \arg\min_{\mathbb{P}\in\mathcal{P}_{k-1}} \sum_{t=1}^{k-1} \sum_{\tau\in\hat{\Gamma}_{t|t-1}} \left(\sigma(\mathbb{P}\left(\tau\right) - \mathbb{P}\left(\tau_0\right)) - \sigma(\bar{\mathbb{P}}_t(\tau) - \bar{\mathbb{P}}_t(\tau_0))\right)^2. \tag{43}$$

we have that

$$\left(\sum_{t=1}^{k-1} \sum_{\tau\in\hat{\Gamma}_{t|t-1}} \left(\sigma(\hat{\mathbb{P}}_k\left(\tau\right) - \hat{\mathbb{P}}_k\left(\tau_0\right)) - \sigma(\bar{\mathbb{P}}_t(\tau) - \bar{\mathbb{P}}_t(\tau_0))\right)^2\right)^{1/2}$$

$$\leq \left(\sum_{t=1}^{k-1} \sum_{\tau\in\hat{\Gamma}_{t|t-1}} \left(\sigma(\hat{\mathbb{P}}_k\left(\tau\right) - \hat{\mathbb{P}}_k\left(\tau_0\right)) - \sigma(\mathbb{P}_t^*(\tau) - \mathbb{P}_t^*(\tau_0))\right)^2\right)^{1/2} + \sqrt{k}\bar{\epsilon}$$

$$\leq \left(\sum_{t=1}^{k-1} \sum_{\tau\in\hat{\Gamma}_{t|t-1}} \left(\sigma(\tilde{\mathbb{P}}_k\left(\tau\right) - \tilde{\mathbb{P}}_k\left(\tau_0\right)) - \sigma(\mathbb{P}_t^*(\tau) - \mathbb{P}_t^*(\tau_0))\right)^2\right)^{1/2} + \sqrt{k}\bar{\epsilon}$$

$$\leq \left(\sum_{t=1}^{k-1} \sum_{\tau\in\hat{\Gamma}_{t|t-1}} \left(\sigma(\tilde{\mathbb{P}}_k\left(\tau\right) - \tilde{\mathbb{P}}_k\left(\tau_0\right)) - \sigma(\bar{\mathbb{P}}_t(\tau) - \bar{\mathbb{P}}_t(\tau_0))\right)^2\right)^{1/2} + 2\sqrt{k}\bar{\epsilon}, \tag{44}$$

where the first and third inequality is obtained by applying $\left\|\sigma(\bar{\mathbb{P}}_k\left(\cdot\right) - \bar{\mathbb{P}}_k\left(\tau_0\right)) - \sigma(\mathbb{P}_t^*(\cdot) - \mathbb{P}_t^*(\tau_0))\right\|_\infty \leq \bar{\epsilon} = \left(1+\beta^{\mathbb{P}}\right)\epsilon$.

Finally, we leverage the relation between $\sum_{t=1}^{k-1} \sum_{\tau\in\hat{\Gamma}_{t|t-1}} \left(\sigma(\hat{\mathbb{P}}_k\left(\tau_t\right) - \hat{\mathbb{P}}_k\left(\tau_0\right)) - \sigma(\bar{\mathbb{P}}_t(\tau_t) - \bar{\mathbb{P}}_t(\tau_0))\right)^2$ and $\sum_{t=1}^{k-1} \sum_{\tau\in\hat{\Gamma}_{t|t-1}} \left(\sigma(\tilde{\mathbb{P}}_k\left(\tau_t\right) - \tilde{\mathbb{P}}_k\left(\tau_0\right)) - \sigma(\bar{\mathbb{P}}_t(\tau_t) - \bar{\mathbb{P}}_t(\tau_0))\right)^2$ above to complete the induction step. Specifically, consider a function space $\overline{\mathcal{P}}_k^{\epsilon,\sigma} : \hat{\Gamma} \to \mathbb{R}$ and filtered sequence $\{\tau_k, \eta_k\}$ in $\Gamma \times \mathbb{R}$, such that, $\eta_k$ is conditionally zero-mean $G$-sub-Gaussian noise. For $\mathbb{P}^*(\cdot) : \hat{\Gamma} \to \mathbb{R}$, suppose that $f_k = \sigma(\mathbb{P}^*\left(\tau_k\right) - \mathbb{P}^*\left(\tau_0\right)) + \eta_k$ and there exists a function $\bar{\mathbb{P}}_t \in \overline{\mathcal{P}}_k^{\epsilon,\sigma}$, such that, for any $k \in [K], \sum_{t=1}^k |\sigma(\mathbb{P}^*\left(\tau_t\right) - \mathbb{P}^*\left(\tau_0\right)) - \sigma(R_t\left(\tau_t\right) - R_t\left(\tau_0\right))| \leq \zeta$. If $\hat{\mathbb{P}}_k$ is an approximate empirical risk minimization solution up to some $\epsilon' \geq 0$, i.e.,

$$\left(\sum_{t=1}^k \sum_{\tau\in\hat{\Gamma}_{t|t-1}} \frac{\left(\sigma(\hat{\mathbb{P}}_k\left(\tau\right) - \hat{\mathbb{P}}_k\left(\tau_0\right)) - f_t\right)^2}{\max\left\{1, \Lambda_t(\theta)/\left|\sigma(\tau\mid\hat{\mathbb{P}}_t) - f_t\right|\right\}}\right)^{1/2}$$

$$\leq \min_{\mathbb{P}\in\mathcal{P}_{k-1}} \left(\sum_{t=1}^k \sum_{\tau\in\hat{\Gamma}_{t|t-1}} \frac{(\sigma(\mathbb{P}\left(\tau\right) - \mathbb{P}\left(\tau_0\right)) - f_t)^2}{\max\left\{1, \Lambda_t(\theta)/\left|\sigma(\tau\mid\hat{\mathbb{P}}_t) - f_t\right|\right\}}\right)^{1/2} + \sqrt{k}\epsilon', \tag{45}$$

with probability at least $1 - \delta$, then we have for all episodes $k \in [K]$,

$$
\left( \sum_{t=1}^{k} \sum_{\tau \in \hat{\Gamma}_{t|t-1}} \frac{\left( \sigma(\hat{\mathbb{P}}_k\left(\tau_t\right) - \hat{\mathbb{P}}_k\left(\tau_0\right)) - \sigma(\bar{\mathbb{P}}_t(\tau_t) - \bar{\mathbb{P}}_t(\tau_0)) \right)^2}{\max\left\{ 1, \Lambda_t(\theta) / \left| \sigma(\tau \mid \bar{\mathbb{P}}) - \sigma(\tau \mid \hat{\mathbb{P}}_t) \right| \right\}} \right)^{1/2}
$$

$$
\leq 10\eta^2 \ln\left( 2\mathcal{N}\left( \overline{\mathcal{P}}_k^{\epsilon,\sigma}, \epsilon, \|\cdot\|_\infty \right) / \delta \right)
$$

$$
+ 5 \sum_{t=1}^{k} \sum_{\tau \in \hat{\Gamma}_{t|t-1}} \frac{\left| \sigma(\hat{\mathbb{P}}_t\left(\tau_t\right) - \hat{\mathbb{P}}_t\left(\tau_0\right)) - \sigma(\bar{\mathbb{P}}\left(\tau_t\right) - \bar{\mathbb{P}}\left(\tau_0\right)) \right| \xi_t}{\max\left\{ 1, \Lambda_t(\theta) / \left| \sigma(\tau \mid \bar{\mathbb{P}}) - \sigma(\tau \mid \hat{\mathbb{P}}_t) \right| \right\}}
$$

$$
+ 10\left(\gamma + \epsilon'\right)\left( \left(\gamma + \epsilon'\right) k + \sqrt{kC_1(k,\xi)} \right), \tag{46}
$$

where $C_1(k,\xi) = 2\left( \xi^2 + 2kG^2 + 3G^2 \ln(2/\delta) \right)$. The reason is as follows. For $\mathbb{P} \in \overline{\mathcal{P}}_k^{\epsilon,\sigma}$, we define $\phi\left(\mathbb{P}, \tau_k\right) = -a\left[ \left(\sigma\left(\tau_k \mid \mathbb{P}\right) - f_k\right)^2 - \left(\sigma\left(\tau_k \mid \bar{\mathbb{P}}\right) - f_k\right)^2 \right] / \max\left\{ 1, \Lambda_k(\theta) / \left| \sigma(\tau \mid \bar{\mathbb{P}}) - \sigma(\tau \mid \hat{\mathbb{P}}_k) \right| \right\}$, where $a = \frac{G^{-2}}{4}$. Let $\mathcal{P}^\epsilon$ be an $\epsilon$-cover of $\mathcal{P}$ under the $\|\cdot\|_\infty$ norm. Denote the cardinality of $\mathcal{P}^\epsilon$ by $\mathcal{N} = \mathcal{N}\left(\mathcal{P}, \epsilon, \|\cdot\|_\infty\right)$. Since $\epsilon_k$ is conditional $G$-sub-Gaussian and $\phi\left(\mathbb{P}, \tau_k\right)$ can be written as

$$
\phi\left(\mathbb{P}, \tau_k\right) = 2a\left[ \sigma\left(\tau_k \mid \mathbb{P}\right) - \sigma\left(\tau_k \mid \bar{\mathbb{P}}\right) \right] / \max\left\{ 1, \Lambda_k(\theta) / \left| \sigma(\tau \mid \bar{\mathbb{P}}) - \sigma(\tau \mid \hat{\mathbb{P}}_k) \right| \right\} \cdot \epsilon_k
$$

$$
- a\left[ \sigma\left(\tau_k \mid \mathbb{P}\right) - \sigma\left(\tau_k \mid \bar{\mathbb{P}}\right) \right]^2 / \max\left\{ 1, \Lambda_k(\theta) / \left| \sigma(\tau \mid \bar{\mathbb{P}}) - \sigma(\tau \mid \hat{\mathbb{P}}_k) \right| \right\}
$$

$$
+ 2a\left[ \sigma\left(\tau_k \mid \mathbb{P}\right) - \sigma\left(\tau_k \mid \bar{\mathbb{P}}\right) \right] / \max\left\{ 1, \Lambda_k(\theta) / \left| \sigma(\tau \mid \bar{\mathbb{P}}) - \sigma(\tau \mid \hat{\mathbb{P}}_k) \right| \right\} \xi, \tag{47}
$$

and $\phi\left(\mathbb{P}, \tau_k\right)$ is conditional $2aG\left[ \sigma\left(\tau_k \mid \mathbb{P}\right) - \sigma\left(\tau_k \mid \bar{\mathbb{P}}\right) \right] / \max\left\{ 1, \Lambda_k(\theta) / \left| \sigma(\tau \mid \bar{\mathbb{P}}) - \sigma(\tau \mid \hat{\mathbb{P}}_k) \right| \right\}$-sub-Gaussian with mean

$$
\mu = -a\left[ \sigma\left(\tau_k \mid \mathbb{P}\right) - \sigma\left(\tau_k \mid \bar{\mathbb{P}}\right) \right]^2 / \max\left\{ 1, \Lambda_k(\theta) / \left| \sigma(\tau \mid \bar{\mathbb{P}}) - \sigma(\tau \mid \hat{\mathbb{P}}_k) \right| \right\}
$$

$$
+ 2a\xi\left[ \sigma\left(\tau_k \mid \mathbb{P}\right) - \sigma\left(\tau_k \mid \bar{\mathbb{P}}\right) \right] / \max\left\{ 1, \Lambda_k(\theta) / \left| \sigma(\tau \mid \bar{\mathbb{P}}) - \sigma(\tau \mid \hat{\mathbb{P}}_k) \right| \right\}, \tag{48}
$$

where $a = \frac{G^{-2}}{4}$. According to Lemma 5, if a variable $X$ is $\sigma$-sub-Gaussian with mean $\mu$ conditional on $\mathcal{S}$, the property of sub-Gaussianity implies that

$$
\ln \mathbb{E}[\exp(s(X - \mu)) \mid \mathcal{S}] \leq \frac{\sigma^2 s^2}{2}. \tag{49}
$$

By taking $s = 1$ in the inequality above, we get

$$
\ln \mathbb{E}_{f_k}\left[ \exp\left(\phi\left(\mathbb{P}, \tau_k\right) - \mu\right) \mid \tau_k, \hat{\Gamma}_{k-1} \right] \leq \frac{4a^2 G^2 \left[ \sigma\left(\tau_k \mid \mathbb{P}\right) - \sigma\left(\tau_k \mid \bar{\mathbb{P}}\right) \right]^2}{2\max\left\{ 1, \Lambda_k(\theta) / \left| \sigma(\tau \mid \bar{\mathbb{P}}) - \sigma(\tau \mid \hat{\mathbb{P}}_k) \right| \right\}^2}
$$

$$
= \frac{\left[ \sigma\left(\tau_k \mid \mathbb{P}\right) - \sigma\left(\tau_k \mid \bar{\mathbb{P}}\right) \right]^2}{8G^2 \max\left\{ 1, \Lambda_k(\theta) / \left| \sigma(\tau \mid \bar{\mathbb{P}}) - \sigma(\tau \mid \hat{\mathbb{P}}_k) \right| \right\}^2}. \tag{50}
$$

It follows that

$$\ln \mathbb{E}_{f_k}\left[\exp\left(\phi\left(\mathbb{P}, \tau_k\right)\right) \mid \tau_k, \hat{\Gamma}_{t-1}\right]$$

$$\leq \frac{\left[\sigma\left(\tau_k \mid \mathbb{P}\right) - \sigma\left(\tau_k \mid \bar{\mathbb{P}}\right)\right]^2}{8G^2 \max\left\{1, \Lambda_k(\theta)/\left|\sigma(\tau \mid \bar{\mathbb{P}}) - \sigma(\tau \mid \hat{\mathbb{P}}_k)\right|\right\}^2} - \frac{\left[\sigma\left(\tau_k \mid \mathbb{P}\right) - \sigma\left(\tau_k \mid \bar{\mathbb{P}}\right)\right]^2}{4G^2 \max\left\{1, \Lambda_k(\theta)/\left|\sigma(\tau \mid \bar{\mathbb{P}}) - \sigma(\tau \mid \hat{\mathbb{P}}_k)\right|\right\}}$$

$$+ \frac{\xi_k \left[\sigma\left(\tau_k \mid \mathbb{P}\right) - \sigma\left(\tau_k \mid \bar{\mathbb{P}}\right)\right]^2}{2G^2 \max\left\{1, \Lambda_k(\theta)/\left|\sigma(\tau \mid \bar{\mathbb{P}}) - \sigma(\tau \mid \hat{\mathbb{P}}_k)\right|\right\}}$$

$$\leq -\frac{\left[\sigma\left(\tau_k \mid \mathbb{P}\right) - \sigma\left(\tau_k \mid \bar{\mathbb{P}}\right)\right]^2}{8G^2 \max\left\{1, \Lambda_k(\theta)/\left|\sigma(\tau \mid \bar{\mathbb{P}}) - \sigma(\tau \mid \hat{\mathbb{P}}_k)\right|\right\}}$$

$$+ \frac{\xi_k \left[\sigma\left(\tau_k \mid \mathbb{P}\right) - \sigma\left(\tau_k \mid \bar{\mathbb{P}}\right)\right]^2}{2G^2 \max\left\{1, \Lambda_k(\theta)/\left|\sigma(\tau \mid \bar{\mathbb{P}}) - \sigma(\tau \mid \hat{\mathbb{P}}_k)\right|\right\}}, \tag{51}$$

where the second inequality is because $\max\left\{1, \Lambda_k(\theta)/\left|\sigma(\tau \mid \bar{\mathbb{P}}) - \sigma(\tau \mid \hat{\mathbb{P}}_k)\right|\right\} \geq 1$. According to Lemma 4 with $\lambda = 1$, we have for all $\mathbb{P} \in \mathcal{P}^\epsilon$ and $k \in [K]$, with probability at least $1 - \delta/2$,

$$\sum_{t=1}^{k} \phi\left(\mathbb{P}, \tau_t\right) \leq -\sum_{t=1}^{k} \frac{\left[\sigma\left(\tau_k \mid \mathbb{P}\right) - \sigma\left(\tau_k \mid \bar{\mathbb{P}}\right)\right]^2}{8G^2 \max\left\{1, \Lambda_k(\theta)/\left|\sigma(\tau \mid \bar{\mathbb{P}}) - \sigma(\tau \mid \hat{\mathbb{P}}_k)\right|\right\}}$$

$$+ \sum_{t=1}^{k} \frac{\left[\sigma\left(\tau_k \mid \mathbb{P}\right) - \sigma\left(\tau_k \mid \bar{\mathbb{P}}\right)\right]^2 \xi}{2G^2 \max\left\{1, \Lambda_k(\theta)/\left|\sigma(\tau \mid \bar{\mathbb{P}}) - \sigma(\tau \mid \hat{\mathbb{P}}_k)\right|\right\}} + \ln(2\mathcal{N}/\delta). \tag{52}$$

Additionally, for all episode $k \in [K]$, we have with probability at least $1 - \delta/2$,

$$\sum_{t=1}^{k} \left(\sigma\left(\tau_t \mid \bar{\mathbb{P}}\right) - f_t\right)^2 \leq \sum_{t=1}^{k} \left(\sigma\left(\tau_t \mid \bar{\mathbb{P}}\right) - \sigma\left(\tau_t \mid \mathbb{P}^*\right) + \sigma\left(\tau_t \mid \mathbb{P}^*\right) - f_t\right)^2$$

$$\leq 2\sum_{t=1}^{k-1} \left(\left(\sigma\left(\tau_t \mid \bar{\mathbb{P}}\right) - \sigma\left(\tau_t \mid \mathbb{P}^*\right)\right)^2 + \left(\sigma\left(\tau_t \mid \mathbb{P}^*\right) - f_t\right)^2\right)$$

$$\leq 2\left(\sum_{t=1}^{k-1} \xi_t^2 + \sum_{t=1}^{k-1} \epsilon_t^2\right)$$

$$\leq 2\left(\xi^2 + 2kG^2 + 3G^2 \ln(2/\delta)\right), \tag{53}$$

where the first inequality is obtained since Cauchy-Schwarz inequality and the last inequality is due to Lemma 4. Now, given $\hat{\mathbb{P}}_k$, there exists $\mathbb{P} \in \overline{\mathcal{P}}_k^{\epsilon,\sigma}$, such that $\left\|\hat{\mathbb{P}}_k - \mathbb{P}\right\|_\infty \leq \epsilon$. With probability at

least $1 - \delta/2$,

$$
\sum_{t=1}^{k} \left[ \left( \sigma\left(\tau_t \mid \mathbb{P}\right) - f_t \right)^2 - \left( \sigma\left(\tau_t \mid \bar{\mathbb{P}}\right) - f_t \right)^2 \right] / \max\left\{ 1, \Lambda_t(\theta) / \left| \sigma(\tau \mid \bar{\mathbb{P}}) - \sigma(\tau \mid \hat{\mathbb{P}}_t) \right| \right\}
$$

$$
\leq \left( \sqrt{ \sum_{t=1}^{k} \left( \sigma\left(\tau_t \mid \hat{\mathbb{P}}_t\right) - f_t \right)^2 / \max\left\{ 1, \Lambda_t(\theta) / \left| \sigma(\tau \mid \bar{\mathbb{P}}) - \sigma(\tau \mid \hat{\mathbb{P}}_t) \right| \right\} } + \sqrt{k}\epsilon \right)^2
$$

$$
- \sum_{t=1}^{k} \left( \sigma\left(\tau_t \mid \bar{\mathbb{P}}\right) - f_t \right)^2 / \max\left\{ 1, \Lambda_t(\theta) / \left| \sigma(\tau \mid \bar{\mathbb{P}}) - \sigma(\tau \mid \hat{\mathbb{P}}_t) \right| \right\}
$$

$$
\leq \left( \sqrt{ \sum_{t=1}^{k} \left( \sigma\left(\tau_t \mid \bar{\mathbb{P}}_t\right) - f_t \right)^2 / \max\left\{ 1, \Lambda_t(\theta) / \left| \sigma(\tau \mid \bar{\mathbb{P}}) - \sigma(\tau \mid \hat{\mathbb{P}}_t) \right| \right\} } + \sqrt{k}(\epsilon + \epsilon') \right)^2
$$

$$
- \sum_{t=1}^{k} \left( \sigma\left(\tau_t \mid \bar{\mathbb{P}}\right) - f_t \right)^2 / \max\left\{ 1, \Lambda_t(\theta) / \left| \sigma(\tau \mid \bar{\mathbb{P}}) - \sigma(\tau \mid \hat{\mathbb{P}}_t) \right| \right\}
$$

$$
\leq (\epsilon + \epsilon')^2 k + 2(\epsilon + \epsilon') \sqrt{k C_1(k, \xi)}, \tag{54}
$$

where the first inequality uses $\left| \sigma\left(\tau_t \mid \mathbb{P}\right) - \sigma\left(\tau_t \mid \hat{\mathbb{P}}\right) \right| \leq \epsilon$ and triangle inequality for all $t$. Finally, with probability at least $1 - \delta$, we have

$$
\left( \sum_{t=1}^{k} \left( \sigma\left(\tau_t \mid \hat{\mathbb{P}}_t\right) - \sigma\left(\tau_t \mid \bar{\mathbb{P}}\right) \right)^2 / \max\left\{ 1, \Lambda_t(\theta) / \left| \sigma(\tau \mid \bar{\mathbb{P}}) - \sigma(\tau \mid \hat{\mathbb{P}}_t) \right| \right\} \right)^{1/2}
$$

$$
\leq \sqrt{\epsilon^2 k} + \left( \sum_{t=1}^{k} \left( \sigma\left(\tau_t \mid \mathbb{P}\right) - \sigma\left(\tau_t \mid \bar{\mathbb{P}}\right) \right)^2 / \max\left\{ 1, \Lambda_t(\theta) / \left| \sigma(\tau \mid \bar{\mathbb{P}}) - \sigma(\tau \mid \hat{\mathbb{P}}_t) \right| \right\} \right)^{1/2}
$$

$$
\leq \sqrt{\epsilon^2 k} + \left( 4 \sum_{t=1}^{k} \left( \sigma\left(\tau_t \mid \mathbb{P}\right) - \sigma\left(\tau_t \mid \bar{\mathbb{P}}\right) \right) \xi_t / \max\left\{ 1, \Lambda_t(\theta) / \left| \sigma(\tau \mid \bar{\mathbb{P}}) - \sigma(\tau \mid \hat{\mathbb{P}}_t) \right| \right\} \right.
$$

$$
+ 8G^2 \ln(2\mathcal{N}/\delta) - 8G^2 \sum_{t=1}^{k} \phi\left(\mathbb{P}, \tau_t\right) \bigg)^{1/2}
$$

$$
\leq \sqrt{\epsilon^2 k} + \left( 4 \sum_{t=1}^{k} \left| \sigma\left(\tau_t \mid \hat{\mathbb{P}}_t\right) - \sigma\left(\tau_t \mid \bar{\mathbb{P}}\right) \right| \xi_t / \max\left\{ 1, \Lambda_t(\theta) / \left| \sigma(\tau \mid \bar{\mathbb{P}}) - \sigma(\tau \mid \hat{\mathbb{P}}_t) \right| \right\} \right.
$$

$$
+ 4\epsilon\xi + 8G^2 \ln(2\mathcal{N}/\delta) + 2(\epsilon + \epsilon')^2 t + 4(\epsilon + \epsilon') \sqrt{k} C_1'(k, \xi) \bigg)^{1/2}
$$

$$
\leq \left( 10G^2 \ln(2\mathcal{N}/\delta) + 5 \sum_{t=1}^{k} \left| \sigma\left(\tau_t \mid \hat{\mathbb{P}}_t\right) - \sigma\left(\tau_t \mid \bar{\mathbb{P}}\right) \right| \xi_t / \max\left\{ 1, \Lambda_t(\theta) / \left| \sigma(\tau \mid \bar{\mathbb{P}}) - \sigma(\tau \mid \hat{\mathbb{P}}_t) \right| \right\} \right.
$$

$$
+ 5\epsilon\xi + 8(\epsilon + \epsilon')^2 k + 5(\epsilon + \epsilon') \sqrt{t C_1(k, \xi)} \bigg)^{1/2}, \tag{55}
$$

where the second inequality is deduced from Eq. (52) and the last inequality uses Cauchy-Schwarz inequality.

Up to here, by letting $\epsilon' = 2\bar{\epsilon}$, $G = 1$ and adding the sum over only sub-sampling feedback $\Gamma_{t|t-1}$, and taking a union bound over $\bar{\mathbb{P}}_\kappa \in \overline{\mathcal{P}}_k^{\epsilon, \sigma}$, we can have that with probability at least $1 - \delta$, the

following inequality holds for all episodes $k \in [K]$ :

$$\sum_{t=1}^{k-1} \sum_{\tau \in \hat{\Gamma}_{t|t-1}} \frac{\left(\sigma\left(\tau_t \mid \hat{\mathbb{P}}_k\right) - \bar{\mathbb{P}}_\kappa\left(\tau_t\right)\right)^2}{\max\left\{1, \Lambda_t(\theta)/\left|\sigma(\tau \mid \bar{\mathbb{P}}) - \sigma(\tau \mid \hat{\mathbb{P}}_t)\right|\right\}}$$

$$\leq 10 \ln\left(2H\mathcal{N}_K(\epsilon)/\delta\right) + 5\sum_{t=1}^{k-1} \sum_{\tau \in \Gamma_{t|t-1}} \frac{\left|\sigma\left(\tau_t \mid \hat{\mathbb{P}}_k\right) - \sigma\left(\tau_t \mid \bar{\mathbb{P}}_\kappa\right)\right| \cdot \xi_t}{\max\left\{1, \Lambda_t(\theta)/\left|\sigma(\tau \mid \bar{\mathbb{P}}) - \sigma(\tau \mid \hat{\mathbb{P}}_t)\right|\right\}}$$

$$+ 10(\epsilon + 2\bar{\epsilon}) \cdot \left((\epsilon + 2\bar{\epsilon})k + \sqrt{2k\left(\xi^2 + 2k + 3\ln(2/\delta)\right)}\right), \tag{56}$$

Further, for all episodes $t \leq k - 1$, we have that

$$\left|\sigma\left(\tau_t \mid \hat{\mathbb{P}}_k\right) - \sigma\left(\tau_t \mid \bar{\mathbb{P}}_\kappa\right)\right| / \max\left\{1, \Lambda_t(\theta)/\left|\sigma(\tau \mid \bar{\mathbb{P}}) - \sigma(\tau \mid \hat{\mathbb{P}}_t)\right|\right\}$$

$$\leq \left|\sigma\left(\tau_t \mid \hat{\mathbb{P}}_k\right) - \sigma\left(\tau_t \mid \bar{\mathbb{P}}_\kappa\right)\right| / \max\left\{1, \Lambda_t(\theta)/\left|\sigma(\tau \mid \bar{\mathbb{P}}) - \sigma(\tau \mid \hat{\mathbb{P}}_t)\right|\right\} + \epsilon$$

$$\leq \frac{\left|\sigma\left(\tau_t \mid \hat{\mathbb{P}}_k\right) - \sigma\left(\tau_t \mid \hat{\mathbb{P}}_t\right)\right|}{\max\left\{1, \Lambda_t(\theta)/\left|\sigma(\tau \mid \bar{\mathbb{P}}) - \sigma(\tau \mid \hat{\mathbb{P}}_t)\right|\right\}} + \frac{\left|\sigma\left(\tau_t \mid \bar{\mathbb{P}}_\kappa\right) - \sigma\left(\tau_t \mid \hat{\mathbb{P}}_t\right)\right|}{\max\left\{1, \Lambda_t(\theta)/\left|\sigma(\tau \mid \bar{\mathbb{P}}) - \sigma(\tau \mid \hat{\mathbb{P}}_t)\right|\right\}} + \epsilon$$

$$\leq 2\alpha\beta^{\mathbb{P}} + \epsilon, \tag{57}$$

where the last inequality is due to $\hat{\mathbb{P}}_k \in \mathcal{P}_{k-1} \subset \mathcal{P}_t$ and the induction hypothesis that $\bar{\mathbb{P}}_\kappa \in \mathcal{P}_t$ for $\kappa \geq t$. Therefore, we have that, with probability at least $1 - \delta$,

$$\left(\lambda + \sum_{t=1}^{k-1} \sum_{\tau \in \hat{\Gamma}_{t|t-1}} \frac{\left(\sigma\left(\tau_t \mid R_k\right) - \sigma\left(\tau_t \mid \bar{\mathbb{P}}_\kappa\right)\right)^2}{\max\left\{1, \Lambda_t(\theta)/\left|\sigma(\tau \mid \bar{\mathbb{P}}) - \sigma(\tau \mid R_t)\right|\right\}}\right)^{1/2}$$

$$\leq \left(\sum_{t=1}^{k-1} \sum_{\tau \in \hat{\Gamma}_{t|t-1}} \frac{\left(\sigma\left(\tau_t \mid \hat{\mathbb{P}}_k\right) - \sigma\left(\tau_t \mid \bar{\mathbb{P}}_\kappa\right)\right)^2}{\max\left\{1, \Lambda_t(\theta)/\left|\sigma(\tau \mid \bar{\mathbb{P}}) - \sigma(\tau \mid \hat{\mathbb{P}}_t)\right|\right\}}\right)^{1/2} + \sqrt{t\bar{\epsilon}} + \sqrt{\lambda}$$

$$\leq \left(10 \ln\left(2H\mathcal{N}_K(\epsilon, \alpha)/\delta\right) + 10\alpha\xi \sup_{s<t} \beta_s^{\mathbb{P}} + 5\epsilon\xi + 10\left(2\beta_\kappa^{\mathbb{P}} + 3\right)^2 \epsilon^2 K + 10\left(2\beta_\kappa^{\mathbb{P}} + 3\right)\gamma\sqrt{KC_1(k,\xi)}\right)^{1/2}$$

$$+ \left(\beta_\kappa^{\mathbb{P}} + 1\right)\epsilon\sqrt{K} + \sqrt{\lambda}$$

$$\leq \left(12\lambda + 12 \ln\left(2H\mathcal{N}_K(\epsilon, \alpha)/\delta\right) + 12\gamma\xi \sup_{t<k} \beta_s^{\mathbb{P}} + 12\left(5\sup_s \beta_s^{\mathbb{P}}\gamma\right)^2 K + 60\sup_s \beta_s^{\mathbb{P}}\gamma\sqrt{KC_1(k,\xi)}\right)^{1/2}$$

$$\leq \beta_k^{\mathbb{P}}, \tag{58}$$

where the first inequality uses the triangle inequality and the second last inequality uses Cauchy-Schwarz inequality. Therefore, we validate the statement in Eq. (41). For all $k \in [K]$, by taking $\kappa = k$ in Eq. (41), we finally complete the proof.

$\square$

By Lemma 2, we know that the comparison based on ground-truth reward function $\mathbb{P}^*(\cdot) \in \mathbb{P}_k$ with high probability.

### B.1.3 HIGH PROBABILITY EVENT FOR THE POLICY

**Lemma 3.** *Under the high probability events for reward function $R$ and transition kernel $\mathbb{P}$, we have $\pi^* \in \Omega_k$ for all episodes $k$.*

*Proof.* First, we know that $\mathbb{E}_{\tau^* \sim (\mathbb{P}, \pi^*)} \sigma(\tau^* \mid R^*) \geq 0$. We decompose the Left-Hand-Side (LHS) of the above inequality into the following three terms:

$$
\begin{aligned}
& \mathbb{E}_{\tau^* \sim (\mathbb{P}, \pi^*)} \sigma(\tau^* \mid R^*) \\
& = \mathbb{E}_{\tau^* \sim (\mathbb{P}, \pi^*)} \sigma(\tau^* \mid R^*) - \mathbb{E}_{\tau^* \sim (\hat{\mathbb{P}}_k, \pi^*)} \sigma(\tau^* \mid R^*) \\
& \quad + \mathbb{E}_{\tau^* \sim (\hat{\mathbb{P}}_k, \pi^*)} \sigma(\tau^* \mid R^*) - \mathbb{E}_{\tau^* \sim (\hat{\mathbb{P}}_k, \pi^*)} \sigma(\tau^* \mid R) \\
& \quad + \mathbb{E}_{\tau^* \sim (\hat{\mathbb{P}}_k, \pi^*)} \sigma(\tau^* \mid R).
\end{aligned}
\tag{59}
$$

We can upper bound the first term in the following way:

$$
\mathbb{E}_{\tau^* \sim (\mathbb{P}, \pi^*)} \sigma(\tau^* \mid R^*) - \mathbb{E}_{\tau^* \sim (\hat{\mathbb{P}}_k, \pi^*)} \sigma(\tau^* \mid R^*) \leq \mathbb{E}_{\tau^* \sim (\hat{\mathbb{P}}_k, \pi^*)} \left[ b_k^{\mathbb{P}}(\tau^*) \right].
\tag{60}
$$

By Lemma 2, we have that,

$$
\begin{aligned}
& \mathbb{E}_{\tau^* \sim (\hat{\mathbb{P}}_k, \pi^*)} \sigma(\tau^* \mid R^*) - \mathbb{E}_{\tau^* \sim (\hat{\mathbb{P}}_k, \pi^*)} \sigma(\tau^* \mid R) \\
& \leq \mathbb{E}_{\tau^* \sim (\hat{\mathbb{P}}_k, \pi^*)} \max_{f_1, f_2 \in \mathcal{B}_{\mathbb{T}, k}} |\sigma(\tau^* \mid R_1) - \sigma(\tau^* \mid R_2)| \\
& = \mathbb{E}_{\tau^* \sim (\hat{\mathbb{P}}_k, \pi^*), \tau_0 \sim (\hat{\mathbb{P}}_k, \pi_0)} b_k^R.
\end{aligned}
\tag{61}
$$

Therefore, we have

$$
\mathbb{E}_{\tau^* \sim (\hat{\mathbb{P}}_k, \pi^*)} \left( \hat{\mathbb{T}}_k(\tau^*) + b_{R_k, k}(\tau^*) + b_k^{\mathbb{P}}(\tau^*) \right) \geq 0, \forall \pi_0,
\tag{62}
$$

which indicates that $\pi^* \in \Omega_k$.

$\square$

## B.2 STEP II: SUB-REGRET UNDER SUM-IMPORTANT STEINER POINTS

According to the confidence set Eq. (15) in Algorithm 1, for all $P' \in \mathbb{P}_k$, we have

$$
\lambda + \sum_{t \in [k-1]} \sum_{\tau \in \Gamma_{t|t-1}, h \in [H]} \frac{\left( \left\langle P'(\cdot \mid s_{t,h}, a_{t,h}) - \hat{P}_k(\cdot \mid s_{t,h}, a_{t,h}), V_{t,h} \right\rangle \right)^2}{\min \left\{ 1, \Lambda_t^P(\theta) / \left| \left\langle [P' - \hat{P}_t](\cdot \mid s_{t,h}, a_{t,h}), V_{t,h} \right\rangle \right| \right\}} \leq \beta^P.
\tag{63}
$$

Let

$$
b_k^P(s, a) \triangleq \max_{\substack{V \in \mathcal{V}, \\ P' \in \mathbb{P}_k}} \frac{\left( P'(\cdot \mid s, a) - \hat{P}_k(\cdot \mid s, a) \right) V(s, a)}{\left( \lambda + \sum_{t=1}^{k-1} \sum_{\tau \in \Gamma_{t|t-1}} \frac{\left\langle [P' - \hat{P}_k](\cdot \mid s_{t,h}, a_{t,h}), V_{t,h} \right\rangle^2}{\max \left\{ 1, \Lambda_t^P(\theta) / \left| \left\langle [P' - \hat{P}_t](\cdot \mid s_{t,h}, a_{t,h}), V_{t,h} \right\rangle \right| \right\}} \right)^{1/2}}.
\tag{64}
$$

According to Eq. (63) and Eq. (64), we have $\left\langle P'(\cdot \mid s_{k,h}, a_{k,h}) - \hat{P}_k(\cdot \mid s_{k,h}, a_{k,h}), V_{k,h} \right\rangle \leq \sqrt{\beta^P} b_k^P(s, a)$, and thus

$$
\left| V_{k,1}(\tau_0 \mid P') - V_{k,1}(\tau_0 \mid \hat{P}_k) \right| \leq \sqrt{\beta^P} b_k^P(\tau_k),
\tag{65}
$$

where $V_{k,1}(\tau_0 \mid P)$ is equal to $V_{k,1}(\tau_0)$ under transition $P$. Then, according to Eq. (2) and the triangle inequality, we have

$$
-\xi_k \leq V_{k,1}(\tau_0 \mid P) - V_{k,1}(\tau_0 \mid P^*) \leq 2\sqrt{\beta^P} b_k^P(s, a) + \xi_k,
\tag{66}
$$

under the high probability event $P^* \in \mathbb{P}_k$.

Then, we can show the sub-regret due to the inconsistency in the agent feedback as follows,

$$
\text{Reg}(K) = \sum_{k=1}^K \left[ V_1^*(\tau_0) - V_1^{\pi_k}(\tau_0) \right] \leq H\zeta + \sum_{k=1}^K \left[ V_{k,1}(\tau_0) - V_1^{\pi_k}(\tau_0) \right],
\tag{67}
$$

where the inequality uses Eq. (66). Next, we focus on bounding the second term on the right-hand side of Eq. (67). A thought experiment: if the reward value for each state action pair $r_h$ is available, then given any policy $\pi : \mathcal{S} \to \mathcal{A}$ and a function $f : \mathcal{S} \times \mathcal{A} \to [0,1]$, at step $h$, the average Bellman error of $f$ under the roll-in policy $\pi$, $\mathbb{E}(f, \pi, h) = \mathbb{E}\left[ f\left(s_h, a_h\right) - r_h - f\left(s_{h+1}, a_{h+1}\right) \mid a_{1:h-1} \sim \pi, a_{h:h+1} \sim \pi_f \right]$ can be used to bound the overestimation gap: $V_f - V^{\pi_f} = \sum_{h=1}^{H} \mathcal{E}\left(f, \pi_f, h\right)$, where $V_f = \mathbb{E}\left[f\left(s_1, \pi_f\left(s_1\right)\right)\right]$. This is because

$$\sum_{h=1}^{H} \mathbb{E}\left[ f\left(s_h, a_h\right) - r_h - f\left(s_{h+1}, a_{h+1}\right) \mid a_{1:h-1} \sim \pi_f, a_{h:h+1} \sim \pi_f \right]$$

$$= \sum_{h=1}^{H} \mathbb{E}\left[ f\left(s_h, a_h\right) - r_h - f\left(s_{h+1}, a_{h+1}\right) \mid a_{1:H} \sim \pi_f \right]$$

$$= \mathbb{E}\left[ \sum_{h=1}^{H} \left( f\left(s_h, a_h\right) - r_h - f\left(s_{h+1}, a_{h+1}\right) \right) \mid a_{1:H} \sim \pi_f \right]$$

$$= \mathbb{E}\left[ f\left(s_1, \pi_f\left(s_1\right)\right) \right] - \mathbb{E}\left[ \sum_{h=1}^{H} r_h \mid a_{1:H} \sim \pi_f \right]$$

$$= V_f - V^{\pi_f}, \tag{68}$$

where the first equality is because all $H$ expected values share the same distribution over trajectories, which is the one induced by $a_{1:H} \sim \pi_f$. Inspired by this idea, we can develop the upper bound of the the second term on the right-hand side of Eq. (67), when the reward value for each state action pair $r_h$ is unavailable, i.e., only a trajectory-wide comparison is available.

### B.3 STEP III: BOUND THE SUM OF POLICY UNCERTAIN BONUSES

Now, we can upper bound the cumulative regret in Theorem 1 as follows. Since $-\xi_k \leq V_{k,1}(\tau_0 \mid P) - V_{k,1}(\tau_0 \mid P^*) \leq 2\sqrt{\beta^P} b_k^P(s, a) + \xi_k$, for all episodes $k \in [K]$, we have that

$$\operatorname{Reg}(K) = \sum_{k=1}^{K} \mathbb{E}_{\tau^* \sim (\mathbb{P}^*, \pi^*)}\left[ \sigma\left(\tau^* \mid R^*\right) \right] - \mathbb{E}_{\tau_k \sim (\mathbb{P}^*, \pi^k)}\left[ \sigma\left(\tau_k \mid R^*\right) \right]$$

$$= \sum_{k=1}^{K} \left( \mathbb{E}_{\tau^* \sim (\hat{\mathbb{P}}_k, \pi^*)} \sigma\left(\tau^* \mid R\right) - \mathbb{E}_{\tau_k \sim (\hat{\mathbb{P}}_k, \pi_k)} \sigma\left(\tau_k \mid R\right) \right)$$

$$+ \sum_{k=1}^{K} \left( \mathbb{E}_{\tau^* \sim (\mathbb{P}^*, \pi^*)} \sigma\left(\tau^* \mid R^*\right) - \mathbb{E}_{\tau_k \sim (\mathbb{P}^*, \pi_k)} \sigma\left(\tau_k \mid R^*\right) \right)$$

$$- \sum_{k=1}^{K} \left( \mathbb{E}_{\tau^* \sim (\hat{\mathbb{P}}_k, \pi^*)} \sigma\left(\tau^* \mid R^*\right) - \mathbb{E}_{\tau_k \sim (\hat{\mathbb{P}}_k, \pi_k)} \sigma\left(\tau_k \mid R^*\right) \right)$$

$$+ \sum_{k=1}^{K} \left( \mathbb{E}_{\tau^* \sim (\hat{\mathbb{P}}_k, \pi^*)} \sigma\left(\tau^* \mid R^*\right) - \mathbb{E}_{\tau_k \sim (\hat{\mathbb{P}}_k, \pi_k)} \sigma\left(\tau_k \mid R^*\right) \right)$$

$$- \sum_{k=1}^{K} \left( \mathbb{E}_{\tau^* \sim (\hat{\mathbb{P}}_k, \pi^*)} \sigma\left(\tau^* \mid R\right) - \mathbb{E}_{\tau_k \sim (\hat{\mathbb{P}}_k, \pi_k)} \sigma\left(\tau_k \mid R\right) \right). \tag{69}$$

We can bound the first term, second and third terms, fourth and fifth terms one-by-one. By definition, we have $0 \leq b_k^R(\tau) \leq 1$ and $0 \leq b_k^P(\tau) \leq 1$. By Azuma's inequality, the following inequality holds with probability at least $1 - \delta/2$,

$$\operatorname{Reg}(K) \leq \xi + \mathbb{E}\left[ \sum_{k=1}^{K} \sum_{\tau \in \Gamma_{t|t-1}} b_k^R + b_k^P\left(\tau_k\right) + 4\sqrt{K \log(4/\delta)} \right]. \tag{70}$$

Thus,

$$
\begin{aligned}
\mathrm{Reg}(K) \leq &\xi + \sum_{k=1}^{K}\sum_{h=1}^{H} \mathbb{E}_{\pi_k}\left[\mathcal{E}_h\left(f_k, s_{k,h}, a_{k,h}\right)\right] \\
\leq &2H\zeta + 2 \underbrace{\sum_{(k,h):\sigma_{k,h}=1} \mathbb{E}_{\pi_k}\left[\min\left(1, \beta_{k,h}^R b_{k,h}^R\left(s_{k,h}, a_{k,h}\right)\right)\right]}_{p_1} \\
&+ 2 \underbrace{\sum_{(k,h):\sigma_{k,h}>1} \mathbb{E}_{\pi_k}\left[\min\left(1, \beta_{k,h}^P b_{k,h}^P\left(s_{k,h}, a_{k,h}\right)\right)\right]}_{p_2},
\end{aligned}
\tag{71}
$$

Therefore, it follows that

$$
\mathrm{Reg}(K) = \tilde{\mathcal{O}}\left(\sqrt{KH\ln\left(\mathcal{N}_K(\gamma)\right)\dim_E(\mathcal{F}, \lambda/K)} + \zeta\left(H + \dim_E(\mathcal{F}, \lambda/K)\right)\right).
\tag{72}
$$

## C  SUPPORTING RESULTS

For completeness, we provide some preliminary results.

### C.1  PRELIMINARY RESULTS IN ZHANG (2023)

**Lemma 4.** *Let $\{\epsilon_s\}$ be a sequence of zero-mean conditional $\sigma$-sub-Gaussian random variables: $\ln \mathbb{E}\left[e^{\lambda\epsilon_i} \mid \mathcal{S}_{i-1}\right] \leq \lambda^2\sigma^2/2$, where $\mathcal{S}_{i-1}$ represents the history data. We have for $t \geq 1$, with probability at least $1 - \delta$,*

$$
\sum_{s=1}^{t} \epsilon_i^2 \leq 2t\sigma^2 + 3\sigma^2\ln(1/\delta).
\tag{73}
$$

*Proof.* By invoking the logarithmic moment generating function estimate in Theorem 2.29 from Zhang (2023), we know that for $\lambda \geq 0$,

$$
\ln \mathbb{E}\left[\exp\left(\lambda\epsilon_i^2\right) \mid \mathcal{S}_{i-1}\right] \leq \lambda\sigma^2 + \frac{\left(\lambda\sigma^2\right)^2}{1 - 2\lambda\sigma^2}.
\tag{74}
$$

Then, by using iterated expectations due to the tower property of conditional expectation, we get

$$
\begin{aligned}
\mathbb{E}\left[\exp\left(\lambda\sum_{i=1}^{t}\epsilon_i^2\right)\right] &= \mathbb{E}\left\{\mathbb{E}\left[\exp\left(\lambda\sum_{i=1}^{t-1}\epsilon_i^2 + \epsilon_t^2\right) \mid \mathcal{S}_{t-1}\right]\right\} \\
&= \mathbb{E}\left\{\exp\left(\lambda\sum_{i=1}^{t-1}\epsilon_i^2\right) \cdot \mathbb{E}\left[\exp\left(\epsilon_t^2\right) \mid \mathcal{S}_{t-1}\right]\right\} \\
&\leq \exp\left(\lambda\sigma^2 + \frac{\left(\lambda\sigma^2\right)^2}{1 - 2\lambda\sigma^2}\right) \cdot \mathbb{E}\left\{\exp\left(\lambda\sum_{i=1}^{t-1}\epsilon_i^2\right)\right\} \\
&\ldots \leq \exp\left(\lambda t\sigma^2 + \frac{\left(\lambda t\sigma^2\right)^2}{1 - 2\lambda\sigma^2}\right),
\end{aligned}
\tag{75}
$$

where the first ineqaulity uses Eq. (74). Now, we can apply the second ineqaulity of Lemma 2.9 from (Zhang, 2023) with $\mu = t\sigma^2, \alpha = 2t\sigma^4, \beta = 2\sigma^2$ and $\epsilon = 2\sigma^2\sqrt{ut}$ to obtain

$$
\inf_{\lambda \geq 0}\left\{-\lambda\left(t\sigma^2 + 2\sqrt{ut\sigma^4} + 2u\sigma^2\right) + \ln \mathbb{E}\left[\exp\left(\lambda\sum_{i=1}^{t}\epsilon_i^2\right)\right]\right\} \leq -u.
\tag{76}
$$

Thus, it follows that

$$\mathbb{P}\left(\sum_{s=1}^{t} \epsilon_i^2 \leq t\sigma^2 + 2\sqrt{ut\sigma^4} + 2u\sigma^2\right)$$

$$\leq \inf_{\lambda \geq 0} \frac{\mathbb{E}\left[\exp\left(\lambda \sum_{i=1}^{t} \epsilon_i^2\right)\right]}{\exp\left(\lambda\left(t\sigma^2 + 2\sqrt{ut\sigma^4} + 2u\sigma^2\right)\right)}$$

$$= \inf_{\lambda \geq 0} \exp\left(-\lambda\left(t\sigma^2 + 2\sqrt{ut\sigma^4} + 2u\sigma^2\right) + \ln\mathbb{E}\left[\exp\left(\lambda\sum_{i=1}^{t}\epsilon_i^2\right)\right]\right)$$

$$\leq e^{-u}, \tag{77}$$

where the first inequality applies Markov's Inequality, and the second inequality uses Eq. (76) and the monotonicity of the exponential function. Taking $u = \ln(1/\delta)$ for $\delta > 0$, we obtain that with probability at least $1 - \delta$

$$\sum_{s=1}^{t} \epsilon_i^2 \leq t\sigma^2 + 2\sqrt{t\ln(1/\delta)\sigma^4} + 2\ln(1/\delta)\sigma^2 \tag{78}$$

$$\leq 2t\sigma^2 + 3\sigma^2 \ln(1/\delta), \tag{79}$$

where the second inequality is deduced since $2\sqrt{t\ln(1/\delta)\sigma^4} \leq t\sigma^2 + \ln(1/\delta)\sigma^2$.

$\square$

**Lemma 5.** *Let $\{X_i\}_{i=1}^{n}$ be independent zero-mean sub-Gaussian random variables that satisfies*

$$\ln\mathbb{E}_{X_i}\left[\exp\left(\lambda X_i\right)\right] \leq \frac{\lambda^2 b_i}{2}, \tag{80}$$

*then for $\lambda < 0.5 b_i$, we have*

$$\ln\mathbb{E}_{X_i}\left[\exp\left(\lambda X_i^2\right)\right] \leq -\frac{1}{2}\ln\left(1 - 2\lambda b_i\right). \tag{81}$$

*Let $Z = \sum_{i=1}^{n} X_i^2$, then*

$$\Pr\left[Z \geq \sum_{i=1}^{n} b_i + 2\sqrt{t\sum_{i=1}^{n} b_i^2} + 2t\left(\max_i b_i\right)\right] \leq e^{-t}, \tag{82}$$

*and*

$$\Pr\left[Z \leq \sum_{i=1}^{n} b_i - 2\sqrt{t\sum_{i=1}^{n} b_i^2}\right] \leq e^{-t}. \tag{83}$$

*Proof.* Let $\xi \sim N(0, 1)$ which is independent of $X_i$. Then for all $\lambda b_i < 0.5$, we have

$$\Lambda_{X_i^2}(\lambda) = \ln\mathbb{E}_{X_i}\left[\exp\left(\lambda X_i^2\right)\right]$$

$$= \ln\mathbb{E}_{X_i}\left[\mathbb{E}_\xi\left[\exp\left(\sqrt{2\lambda}\xi X_i\right)\right]\right]$$

$$= \ln\mathbb{E}_\xi\left[\mathbb{E}_{X_i}\left[\exp\left(\sqrt{2\lambda}\xi X_i\right)\right]\right]$$

$$\leq \ln\mathbb{E}_\xi\left[\exp\left(\lambda\xi^2 b_i\right)\right]$$

$$= -\frac{1}{2}\ln\left(1 - 2\lambda b_i\right), \tag{84}$$

where the inequality used the sub-Gaussian assumption. The second and the last equalities can be obtained using Gaussian integration. This proves the first bound of the lemma.

For $\lambda \geq 0$, we obtain

$$\Lambda_{X_i^2}(\lambda) \leq -0.5 \ln\left(1 - 2\lambda b_i\right)$$

$$= 0.5 \sum_{k=1}^{\infty} \frac{(2\lambda b_i)^k}{k}$$

$$\leq \lambda b_i + (\lambda b_i)^2 \sum_{k \geq 0} (2\lambda b_i)^k$$

$$= \lambda b_i + \frac{(\lambda b_i)^2}{1 - 2\lambda b_i}. \tag{85}$$

The first probability inequality of the lemma follows from Theorem 2.10 with $\mu = n^{-1} \sum_{i=1}^n b_i, \alpha = (2/n) \sum_{i=1}^n b_i^2$ and $\beta = 2 \max_i b_i$.

If $\lambda \leq 0$, then

$$\Lambda_{X_i^2}(\lambda) \leq -0.5 \ln\left(1 - 2\lambda b_i\right) \leq \lambda b_i + \lambda^2 b_i^2. \tag{86}$$

The second probability inequality of the theorem follows from the sub-Gaussian tail inequality of Theorem 2.12 with $\mu = n^{-1} \sum_{i=1}^n b_i$ and $b = (2/n) \sum_{i=1}^n b_i^2$.

$\square$

From Lemma 5, we can obtain the following expressions for $\chi_n^2$ tail bound by taking $b_i = 1$. With probability at least $1 - \delta$:

$$Z \leq n + 2\sqrt{n \ln(1/\delta)} + 2 \ln(1/\delta). \tag{87}$$

and with probability at least $1 - \delta$ :

$$Z \geq n - 2\sqrt{n \ln(1/\delta)}. \tag{88}$$

**Definition 3.** *Given a random variable $X$, we may define its logarithmic moment generating function as*

$$\Lambda_X(\lambda) = \ln \mathbb{E}\left[e^{\lambda X}\right]. \tag{89}$$

*Moreover, given $z \in \mathbb{R}$, the rate function $I_X(z)$ is defined as*

$$I_X(z) = \begin{cases} \sup_{\lambda > 0} \left[\lambda z - \Lambda_X(\lambda)\right] & z > \mu \\ 0 & z = \mu \\ \sup_{\lambda < 0} \left[\lambda z - \Lambda_X(\lambda)\right] & z < \mu \end{cases} \tag{90}$$

*where $\mu = \mathbb{E}[X]$.*

The above definition can be used to obtain exponential tail bounds for sums of independent variables as follows.

**Lemma 6.** *For any $n$ and $\epsilon > 0$ :*

$$\frac{1}{n} \ln \Pr\left(\bar{X}_n \geq \mu + \epsilon\right) \leq -I_{X_1}(\mu + \epsilon) = \inf_{\lambda > 0} \left[-\lambda(\mu + \epsilon) + \ln \mathbb{E}e^{\lambda X_1}\right] \tag{91}$$

$$\frac{1}{n} \ln \Pr\left(\bar{X}_n \leq \mu - \epsilon\right) \leq -I_{X_1}(\mu - \epsilon) = \inf_{\lambda < 0} \left[-\lambda(\mu - \epsilon) + \ln \mathbb{E}e^{\lambda X_1}\right] \tag{92}$$

*Proof.* We choose $h(z) = e^{\lambda n z}$ in Theorem 2.2 with $S = \left\{\bar{X}_n - \mu \geq \epsilon\right\}$. For $\lambda > 0$, we have

$$\Pr\left(\bar{X}_n \geq \mu + \epsilon\right) \leq \frac{\mathbb{E}e^{\lambda n \bar{X}_n}}{e^{\lambda n(\mu+\epsilon)}} = \frac{\mathbb{E}e^{\lambda \sum_{i=1}^n X_i}}{e^{\lambda n(\mu+\epsilon)}}$$

$$= \frac{\mathbb{E} \prod_{i=1}^n e^{\lambda X_i}}{e^{\lambda n(\mu+\epsilon)}} = e^{-\lambda n(\mu+\epsilon)} \left[\mathbb{E}e^{\lambda X_1}\right]^n. \tag{93}$$

The last equation used the independence of $X_i$ as well as they are identically distributed. Therefore by taking logarithm, we obtain

$$\ln \Pr\left(\bar{X}_n \geq \mu + \epsilon\right) \leq n \left[-\lambda(\mu + \epsilon) + \ln \mathbb{E}e^{\lambda X_1}\right]. \tag{94}$$

Taking inf over $\lambda > 0$ on the right hand side, we obtain the first desired bound. Similarly, we can obtain the second bound.

$\square$

**Proposition 1.** *Fix any sequence $\{\mathcal{F}_t : t \in \mathbb{N}\}$, where $\mathcal{F}_t \subset \mathcal{F}$ is measurable with respect to $\sigma(H_t)$. Then for any $T \in \mathbb{N}$, with probability 1,*

$$\operatorname{Reg}\left(T, \pi^{\mathcal{F}_{1:\infty}}\right) \leq \sum_{t=1}^{T} \left[w_{\mathcal{F}_t}(A_t) + C\mathbf{1}\left(f_\theta \notin \mathcal{F}_t\right)\right] \tag{95}$$

$$\mathbb{E}\left[\operatorname{Reg}\left(T, \pi^{\mathrm{TS}}\right)\right] \leq \mathbb{E}\left[\sum_{t=1}^{T} \left[w_{\mathcal{F}_t}(A_t) + C\mathbf{1}\left(f_\theta \notin \mathcal{F}_t\right)\right]\right]. \tag{96}$$

*Proof.* To reduce notation, define the upper and lower bounds $U_t(a) = \sup\{f(a) : f \in \mathcal{F}_t\}$ and $L_t(a) = \inf\{f(a) : f \in \mathcal{F}_t\}$. Whenever $f_\theta \in \mathcal{F}_t$, the bounds $L_t(a) \leq f_\theta(a) \leq U_t(a)$ hold for all actions. This implies

$$\begin{aligned}
f_\theta(A_t^*) - f_\theta(A_t) &\leq U_t(A_t^*) - L_t(A_t) + C\mathbf{1}\left(f_\theta \notin \mathcal{F}_t\right) \\
&= w_{\mathcal{F}_t}(A_t) + C\mathbf{1}\left(f_\theta \notin \mathcal{F}_t\right) + \left[U_t(A_t^*) - U_t(A_t)\right].
\end{aligned} \tag{97}$$

Eq. (95) follows almost immediately, since the policy $\pi^{\mathcal{F}_{1:\infty}}$ chooses an action $A_t$ that maximizes $U_t(a)$. This implies $U_t(A_t) \geq U_t(A_t^*)$ by definition, and the last term in Eq. (97) is negative. The result Eq. (95) follows by summing over $t$.

Now consider Eq. (96). Summing equation Eq. (97) over $t$ shows,

$$\operatorname{Reg}\left(T, \pi^{\mathrm{TS}}\right) \leq \sum_{t=1}^{T} \left[w_{\mathcal{F}_t}(A_t) + C\mathbf{1}\left(f_\theta \notin \mathcal{F}_t\right)\right] + M_T, \tag{98}$$

where $M_T := \sum_{t=1}^{T} \left[U_t(A_t^*) - U_t(A_t)\right]$. Now, by the definition of Thompson sampling $\mathbb{P}\left(A_t \in \cdot \mid H_t\right) = \mathbb{P}\left(A_t^* \in \cdot \mid H_t\right)$. That is $A_t$ and $A_t^*$ are identically distributed under the posterior. In addition, since the confidence set $\mathcal{F}_t$ is $\sigma(H_t)$-measurable, so is the induced upper confidence bound $U_t(\cdot)$. This implies $\mathbb{E}\left[U_t(A_t) \mid H_t\right] = \mathbb{E}\left[U_t(A_t^*) \mid H_t\right]$, and therefore that $\mathbb{E}\left[M_T\right] = 0$.

$\square$

### C.2.1 PRELIMINARIES: MARTINGALE EXPONENTIAL INEQUALITIES

Consider random variables $(Z_n \mid n \in \mathbb{N})$ adapted to the filtration $(\mathcal{H}_n : n = 0, 1, \ldots)$. Assume $\mathbb{E}\left[\exp\{\lambda Z_i\}\right]$ is finite for all $\lambda$. Define the conditional mean $\mu_i = \mathbb{E}\left[Z_i \mid \mathcal{H}_{i-1}\right]$. We define the conditional cumulant generating function of the centered random variable $[Z_i - \mu_i]$ by $\psi_i(\lambda) = \log \mathbb{E}\left[\exp\left(\lambda[Z_i - \mu_i]\right) \mid \mathcal{H}_{i-1}\right]$. Let

$$M_n(\lambda) = \exp\left\{\sum_{i=1}^{n} \lambda[Z_i - \mu_i] - \psi_i(\lambda)\right\}. \tag{99}$$

**Lemma 7.** $(M_n(\lambda) \mid n \in \mathbb{N})$ *is a Martinagale, and* $\mathbb{E}\left[M_n(\lambda)\right] = 1$.

*Proof.* By definition

$$\begin{aligned}
&\mathbb{E}\left[M_1(\lambda) \mid \mathcal{H}_0\right] \\
&= \mathbb{E}\left[\exp\{\lambda[Z_1 - \mu_1] - \psi_1(\lambda) \mid \mathcal{H}_0\}\right] \\
&= \mathbb{E}\left[\exp\{\lambda[Z_1 - \mu_1]\} \mid \mathcal{H}_0\right] / \exp\{\psi_1(\lambda)\} \\
&= 1.
\end{aligned} \tag{100}$$

Then, for any $n \geq 2$,

$$\mathbb{E}\left[M_n(\lambda) \mid \mathcal{H}_{n-1}\right]$$

$$= \mathbb{E}\left[\exp\left\{\sum_{i=1}^{n-1} \lambda\left[Z_i - \mu_i\right] - \psi_i(\lambda)\right\} \exp\left\{\lambda\left[Z_n - \mu_n\right] - \psi_n(\lambda)\right\} \mid \mathcal{H}_{n-1}\right]$$

$$= \exp\left\{\sum_{i=1}^{n-1} \lambda\left[Z_i - \mu_i\right] - \psi_i(\lambda)\right\} \mathbb{E}\left[\exp\left\{\lambda\left[Z_n - \mu_n\right] - \psi_n(\lambda)\right\} \mid \mathcal{H}_{n-1}\right]$$

$$= \exp\left\{\sum_{i=1}^{n-1} \lambda\left[Z_i - \mu_i\right] - \psi_i(\lambda)\right\}$$

$$= M_{n-1}(\lambda). \tag{101}$$

$\square$

**Lemma 8.** *For all $x \geq 0$ and $\lambda \geq 0$, $\mathbb{P}\left(\sum_1^n \lambda Z_i \leq x + \sum_1^n \left[\lambda\mu_i + \psi_i(\lambda)\right], \forall n \in \mathbb{N}\right) \geq 1 - e^{-x}$.*

*Proof.* For any $\lambda$, $M_n(\lambda)$ is a martingale with $\mathbb{E}\left[M_n(\lambda)\right] = 1$. Therefore, for any stopping time $\tau$, $\mathbb{E}\left[M_{\tau \wedge n}(\lambda)\right] = 1$. For arbitrary $x \geq 0$, define $\tau_x = \inf\left\{n \geq 0 \mid M_n(\lambda) \geq x\right\}$ and note that $\tau_x$ is a stopping time corresponding to the first time $M_n$ crosses the boundary at $x$. Then, $\mathbb{E}\left[M_{\tau_x \wedge n}(\lambda)\right] = 1$ and by Markov's inequality:

$$x\mathbb{P}\left(M_{\tau_x \wedge n}(\lambda) \geq x\right) \leq \mathbb{E}M_{\tau_x \wedge n}(\lambda) = 1. \tag{102}$$

We note that the event $\left\{M_{\tau_x \wedge n}(\lambda) \geq x\right\} = \bigcup_{k=1}^n \left\{M_k(\lambda) \geq x\right\}$. So we have shown that for all $x \geq 0$ and $n \geq 1$,

$$\mathbb{P}\left(\bigcup_{k=1}^n \left\{M_k(\lambda) \geq x\right\}\right) \leq \frac{1}{x}. \tag{103}$$

Taking the limit as $n \to \infty$, and applying the monotone convergence theorem shows $\mathbb{P}\left(\bigcup_{k=1}^\infty \left\{M_k(\lambda) \geq x\right\}\right) \leq \frac{1}{x}$, or, $\mathbb{P}\left(\bigcup_{k=1}^\infty \left\{M_k(\lambda) \geq e^x\right\}\right) \leq e^{-x}$. This then shows, using the definition of $M_k(\lambda)$, that

$$\mathbb{P}\left(\bigcup_{n=1}^\infty \left\{\sum_{i=1}^n \lambda\left[Z_i - \mu_i\right] - \psi_i(\lambda) \geq x\right\}\right) \leq e^{-x}. \tag{104}$$

$\square$

### C.2.2 PROOF OF LEMMA 9

**Lemma 9.** *For any $\delta > 0$ and $f : \mathcal{A} \mapsto \mathbb{R}$,*

$$\mathbb{P}\left(L_{2,t}(f) \geq L_{2,t}(f_\theta) + \frac{1}{2}\|f - f_\theta\|_{2,E_t}^2 - 4\eta^2 \log(1/\delta), \forall t \in \mathbb{N} \,\middle|\, \theta\right) \geq 1 - \delta. \tag{105}$$

We will transform our problem in order to apply the general exponential martingale result shown above. Since we work conditionally on $\theta$, to reduce notation we denote the conditional probability and expectation operators $\mathbb{P}_\theta(\cdot) = \mathbb{P}(\cdot \mid \theta)$ and $\mathbb{E}_\theta[\cdot] = \mathbb{E}[\cdot \mid \theta]$. We set $\mathcal{H}_{t-1}$ to be the $\sigma$-algebra generated by $(H_t, A_t)$ and set $\mathcal{H}_0 = \sigma(\emptyset, \Omega)$. By previous assumptions, $\epsilon_t := R_t - f_\theta(A_t)$ satisfies $\mathbb{E}_\theta\left[\epsilon_t \mid \mathcal{H}_{t-1}\right] = 0$ and $\mathbb{E}_\theta\left[\exp\left\{\lambda\epsilon_t\right\} \mid \mathcal{H}_{t-1}\right] \leq \exp\left\{\frac{\lambda^2\eta^2}{2}\right\}$ a.s. for all $\lambda$. Define $Z_t = \left(f_\theta(A_t) - R_t\right)^2 - \left(f(A_t) - R_t\right)^2$.

*Proof.* By definition $\sum_1^T Z_t = L_{2,T+1}(f_\theta) - L_{2,T+1}(f)$. Some calculation shows that $Z_t = -\left(f(A_t) - f_\theta(A_t)\right)^2 + 2\left(f(A_t) - f_\theta(A_t)\right)\epsilon_t$. Therefore, the conditional mean and conditional

cumulant generating function satisfy:

$$\mu_t = \mathbb{E}_\theta \left[ Z_t \mid \mathcal{H}_{t-1} \right] = - \left( f(A_t) - f_\theta(A_t) \right)^2 \tag{106}$$

$$\psi_t(\lambda) = \log \mathbb{E}_\theta \left[ \exp \left( \lambda \left[ Z_t - \mu_t \right] \right) \mid \mathcal{H}_{t-1} \right]$$

$$= \log \mathbb{E}_\theta \left[ \exp \left( 2\lambda \left( f(A_t) - f_\theta(A_t) \right) \epsilon_t \right) \mid \mathcal{H}_{t-1} \right] \leq \frac{\left( 2\lambda \left[ f(A_t) - f_\theta(A_t) \right] \right)^2 \eta^2}{2}. \tag{107}$$

Applying Lemma 8 shows that for all $x \geq 0, \lambda \geq 0$,

$$\mathbb{P}_\theta \left( \sum_{k=1}^t \lambda Z_k \leq x - \lambda \sum_{k=1}^t \left( f(A_k) - f_\theta(A_k) \right)^2 + \frac{\lambda^2}{2} \left( 2f(A_k) - 2f_\theta(A_k) \right)^2 \eta^2, \forall t \in \mathbb{N} \right)$$

$$\geq 1 - e^{-x}. \tag{108}$$

Rearranging terms, we have

$$\mathbb{P}_\theta \left( \sum_{k=1}^t Z_k \leq \frac{x}{\lambda} + \sum_{k=1}^t \left( f(A_k) - f_\theta(A_k) \right)^2 \left( 2\lambda\eta^2 - 1 \right), \forall t \in \mathbb{N} \right) \geq 1 - e^{-x}. \tag{109}$$

Choosing $\lambda = \frac{1}{4\eta^2}, x = \log \frac{1}{\delta}$, and using the definition of $\sum_1^t Z_k$ implies

$$\mathbb{P}_\theta \left( L_{2,t}(f) \geq L_{2,t}(f_\theta) + \frac{1}{2} \|f - f_\theta\|_{2,E_t}^2 - 4\eta^2 \log(1/\delta), \forall t \in \mathbb{N} \right) \geq 1 - \delta. \tag{110}$$

$\square$

### C.2.3 LEAST SQUARES BOUND - PROOF OF PROPOSITION 2

**Proposition 2.** *For all $\delta > 0$ and $\alpha > 0$, if $\mathcal{F}_t = \left\{ f \in \mathcal{F} : \left\| f - \hat{f}_t^{LS} \right\|_{2,E_t} \leq \sqrt{\beta_t^*(\mathcal{F}, \delta, \alpha)} \right\}$ for all $t \in \mathbb{N}$, then*

$$\mathbb{P}_\theta \left( f_\theta \in \bigcap_{t=1}^\infty \mathcal{F}_t \right) \geq 1 - 2\delta. \tag{111}$$

*Proof.* Let $\mathcal{F}^\alpha \subset \mathcal{F}$ be an $\alpha$-cover of $\mathcal{F}$ in the sup-norm in the sense that for any $f \in \mathcal{F}$ there is an $f^\alpha \in \mathcal{F}^\alpha$ such that $\|f^\alpha - f\|_\infty \leq \epsilon$. By a union bound, conditional on $\theta$, with probability at least $1 - \delta$,

$$L_{2,t}(f^\alpha) - L_{2,t}(f_\theta) \geq \frac{1}{2} \|f^\alpha - f_\theta\|_{2,E_t} - 4\eta^2 \log \left( |\mathcal{F}^\alpha| / \delta \right), \forall t \in \mathbb{N}, f \in \mathcal{F}^\alpha. \tag{112}$$

Therefore, with probability at least $1 - \delta$, for all $t \in \mathbb{N}$ and $f \in \mathcal{F}$, we have

$$L_{2,t}(f) - L_{2,t}(f_\theta) \geq \frac{1}{2} \|f - f_\theta\|_{2,E_t}^2 - 4\eta^2 \log \left( |\mathcal{F}^\alpha| / \delta \right)$$

$$+ \underbrace{\min_{f^\alpha \in \mathcal{F}^\alpha} \left\{ \frac{1}{2} \|f^\alpha - f_\theta\|_{2,E_t}^2 - \frac{1}{2} \|f - f_\theta\|_{2,E_t}^2 + L_{2,t}(f) - L_{2,t}(f^\alpha) \right\}}_{\text{Discretization Eror}}. \tag{113}$$

Lemma 10, which we establish in the next section, asserts that with probability at least $1 - \delta$, the discretization error is bounded for all $t$ by $\alpha\eta_t$ where $\eta_t := t \left[ 8C + \sqrt{8\eta^2 \ln (4t^2/\delta)} \right]$. Since the least squares estimate $\hat{f}_t^{LS}$ has lower squared error than $f_\theta$ by definition, we find with probability at least $1 - 2\delta$,

$$\frac{1}{2} \left\| \hat{f}_t^{\text{LS}} - f_\theta \right\|_{2,E_t}^2 \leq 4\eta^2 \log \left( |\mathcal{F}^\alpha| / \delta \right) + \alpha\eta_t. \tag{114}$$

Taking the infimum over the size of $\alpha$ covers implies:

$$\left\| \hat{f}_t^{LS} - f_\theta \right\|_{2,E_t} \leq \sqrt{8\eta^2 \log \left( N(\mathcal{F}, \alpha, \|\cdot\|_\infty) / \delta \right) + 2\alpha\eta_t} \overset{\text{def}}{=} \sqrt{\beta_t^*(\mathcal{F}, \delta, \alpha)}. \tag{115}$$

$\square$

### C.2.4 DISCRETIZATION ERROR

**Lemma 10.** *If $f^\alpha$ satisfies $\|f - f^\alpha\|_\infty \le \alpha$, then, conditional on $\theta$, with probability at least $1 - \delta$,*

$$\left| \frac{1}{2} \|f^\alpha - f_\theta\|_{2,E_t}^2 - \frac{1}{2} \|f - f_\theta\|_{2,E_t}^2 + L_{2,t}(f) - L_{2,t}(f^\alpha) \right|$$

$$\le \alpha t \left[ 8C + \sqrt{8\eta^2 \ln(4t^2/\delta)} \right], \forall t \in \mathbb{N}. \tag{116}$$

*Proof.* Since any two functions in $f, f^\alpha \in \mathcal{F}$ satisfy $\|f - f^\alpha\|_\infty \le C$, it is enough to consider $\alpha \le C$. We find

$$\left| (f^\alpha)^2(a) - (f)^2(a) \right| \le \max_{-\alpha \le y \le \alpha} \left| (f(a) + y)^2 - f(a)^2 \right| = 2f(a)\alpha + \alpha^2 \le 2C\alpha + \alpha^2, \tag{117}$$

which implies

$$\left| (f^\alpha(a) - f_\theta(a))^2 - (f(a) - f_\theta(a))^2 \right|$$

$$= \left| \left[ (f^\alpha)(a)^2 - f(a)^2 \right] + 2f_\theta(a)(f(a) - f^\alpha(a)) \right|$$

$$\le 4C\alpha + \alpha^2, \tag{118}$$

and

$$\left| (R_t - f(a))^2 - (R_t - f^\alpha(a))^2 \right|$$

$$= \left| 2R_t(f^\alpha(a) - f(a)) + f(a)^2 - f^\alpha(a)^2 \right|$$

$$\le 2\alpha |R_t| + 2C\alpha + \alpha^2. \tag{119}$$

Summing over $t$, we find that the left hand side of Eq. (116) is bounded by

$$\sum_{k=1}^{t-1} \left( \frac{1}{2} \left[ 4C\alpha + \alpha^2 \right] + \left[ 2\alpha |R_k| + 2C\alpha + \alpha^2 \right] \right) \le \alpha \sum_{k=1}^{t-1} \left( 6C + 2|R_k| \right). \tag{120}$$

Because $\epsilon_k$ is sub-Gaussian, $\mathbb{P}_\theta \left( |\epsilon_k| > \sqrt{2\eta^2 \ln(2/\delta)} \right) \le \delta$. By a union bound, $\mathbb{P}_\theta \left( \exists k, s.t., |\epsilon_k| > \sqrt{2\eta^2 \ln(4t^2/\delta)} \right) \le \frac{\delta}{2} \sum_{k=1}^\infty \frac{1}{k^2} \le \delta$. Since $|R_k| \le C + |\epsilon_k|$, this shows that with probability at least $1 - \delta$, the discretization error is bounded for all $t$ by $\alpha\eta_t$, where $\eta_t \triangleq t \left[ 8C + 2\sqrt{2\eta^2 \ln(4t^2/\delta)} \right]$.

$\square$

### C.2.5 BOUNDING THE SUM OF WIDTHS

**Proposition 3.** *If $(\beta_t \ge 0 \mid t \in \mathbb{N})$ is a nondecreasing sequence and $\mathcal{F}_t := \left\{ f \in \mathcal{F} : \left\| f - \hat{f}_t^{LS} \right\|_{2,E_t} \le \sqrt{\beta_t} \right\}$ then*

$$\sum_{t=1}^T \mathbf{1}\left( w_{\mathcal{F}_t}(A_t) > \epsilon \right) \le \left( \frac{4\beta_T}{\epsilon^2} + 1 \right) \dim_E(\mathcal{F}, \epsilon), \tag{121}$$

*for all $T \in \mathbb{N}$ and $\epsilon > 0$.*

*Proof.* (i) We begin by showing that if $w_t(A_t) > \epsilon$ then $A_t$ is $\epsilon$-dependent on fewer than $4\beta_T/\epsilon^2$ disjoint subsequences of $(A_1, .., A_{t-1})$, for $T > t$.

To see this, note that if $w_{\mathcal{F}_t}(A_t) > \epsilon$ there are $\underline{f}, \bar{f} \in \mathcal{F}_t$ such that $\bar{f}(A_t) - \underline{f}(A_t) > \epsilon$. By definition, since $\bar{f}(A_t) - \underline{f}(A_t) > \epsilon$, if $A_t$ is $\epsilon$-dependent on a subsequence $(A_{i_1}, \ldots, A_{i_k})$ of $(A_1, .., A_{t-1})$, then $\sum_{j=1}^k \left( \bar{f}(A_{i_j}) - \underline{f}(\bar{A}_{i_j}) \right)^2 > \epsilon^2$. It follows that, if $A_t$ is $\epsilon$-dependent on $K$

disjoint subsequences of $(A_1, .., A_{t-1})$, then $\|\bar{f} - \underline{f}\|_{2,E_t}^2 > K\epsilon^2$. By the triangle inequality, we have

$$\|\bar{f} - \underline{f}\|_{2,E_t} \leq \left\|\bar{f} - \hat{f}_t^{LS}\right\|_{2,E_t} + \left\|\underline{f} - \hat{f}_t^{LS}\right\|_{2,E_t} \leq 2\sqrt{\beta_t} \leq 2\sqrt{\beta_T}, \tag{122}$$

and it follows that $K < 4\beta_T/\epsilon^2$.

(ii) Next, we show that in any action sequence $(a_1, .., a_\tau)$, there is some element $a_j$ that is $\epsilon$-dependent on at least $\tau/d - 1$ disjoint subsequences of $(a_1, .., a_{j-1})$, where $d \triangleq \dim_E(\mathcal{F}, \epsilon)$.

To show this, for an integer $K$ satisfying $Kd + 1 \leq \tau \leq Kd + d$, we will construct $K$ disjoint subsequences $B_1, \ldots, B_K$. First let $B_i = (a_i)$ for $i = 1, .., K$. If $a_{K+1}$ is $\epsilon$-dependent on each subsequence $B_1, .., B_K$, our claim is established. Otherwise, select a subsequence $B_i$ such that $a_{K+1}$ is $\epsilon$-independent and append $a_{K+1}$ to $B_i$. Repeat this process for elements with indices $j > K+1$ until $a_j$ is $\epsilon$-dependent on each subsequence or $j = \tau$. In the latter scenario $\sum |B_i| \geq Kd$, and since each element of a subsequence $B_i$ is $\epsilon$-independent of its predecessors, $|B_i| = d$. In this case, $a_\tau$ must be $\epsilon$-dependent on each subsequence, by the definition of $\dim_E(\mathcal{F}, \epsilon)$.

Now consider taking $(a_1, \ldots, a_\tau)$ to be the subsequence $(A_{t_1}, \ldots, A_{t_\tau})$ of $(A_1, \ldots, A_T)$ consisting of elements $A_t$ for which $w_{\mathcal{F}_t}(A_t) > \epsilon$. As we have established, each $A_{t_j}$ is $\epsilon$-dependent on fewer than $4\beta_T/\epsilon^2$ disjoint subsequences of $(A_1, .., A_{t_j-1})$. It follows that each $a_j$ is $\epsilon$-dependent on fewer than $4\beta_T/\epsilon^2$ disjoint subsequences of $(a_1, .., a_{j-1})$. Combining this with the fact we have established that there is some $a_j$ that is $\epsilon$-dependent on at least $\tau/d - 1$ disjoint subsequences of $(a_1, .., a_{j-1})$, we have $\tau/d - 1 \leq 4\beta_T/\epsilon^2$. It follows that $\tau \leq (4\beta_T/\epsilon^2 + 1) d$, which is our desired result.

$\square$

**Lemma 11.** *If* $(\beta_t \geq 0 \mid t \in \mathbb{N})$ *is a nondecreasing sequence and* $\mathcal{F}_t := \left\{f \in \mathcal{F} : \left\|f - \hat{f}_t^{LS}\right\|_{2,E_t} \leq \sqrt{\beta_t}\right\}$ *then with probability 1,*

$$\sum_{t=1}^T w_{\mathcal{F}_t}(A_t) \leq \frac{1}{T} + \min\left\{\dim_E\left(\mathcal{F}, \alpha_T^{\mathcal{F}}\right), T\right\} C + 4\sqrt{\dim_E\left(\mathcal{F}, \alpha_T^{\mathcal{F}}\right)\beta_T T}, \tag{123}$$

*for all* $T \in \mathbb{N}$.

*Proof.* To reduce notation, write $d = \dim_E\left(\mathcal{F}, \alpha_T^{\mathcal{F}}\right)$ and $w_t = w_t(A_t)$. Reorder the sequence $(w_1, \ldots, w_T) \to (w_{i_1}, \ldots, w_{i_T})$ where $w_{i_1} \geq w_{i_2} \geq \ldots \geq w_{i_T}$. We have

$$\sum_{t=1}^T w_{\mathcal{F}_t}(A_t) = \sum_{t=1}^T w_{i_t}$$

$$= \sum_{t=1}^T w_{i_t} \mathbf{1}\left\{w_{i_t} \leq \alpha_T^{\mathcal{F}}\right\} + \sum_{t=1}^T w_{i_t} \mathbf{1}\left\{w_{i_t} > \alpha_T^{\mathcal{F}}\right\}$$

$$\leq \frac{1}{T} + \sum_{t=1}^T w_{i_t} \mathbf{1}\left\{w_{i_t} > \alpha_T^{\mathcal{F}}\right\}. \tag{124}$$

The final step in the above inequality uses that either $\alpha_T^{\mathcal{F}} = T^{-2}$ and $\sum_{t=1}^T \alpha_T^{\mathcal{F}} = T^{-1}$ or $\alpha_T^{\mathcal{F}}$ is set below the smallest possible width and hence $\mathbf{1}\left\{w_{i_t} \leq \alpha_T^{\mathcal{F}}\right\}$ never occurs.

Now, we know $w_{i_t} \leq C$. In addition, $w_{i_t} > \epsilon \Longleftrightarrow \sum_{k=1}^T \mathbf{1}(w_{\mathcal{F}_k}(A_k) > \epsilon) \geq t$. By Proposition 3, this can only occur if $t < \left(\frac{4\beta_T}{\epsilon^2} + 1\right) \dim_E(\mathcal{F}, \epsilon)$. For $\epsilon \geq \alpha_T^{\mathcal{F}}$, $\dim_E(\mathcal{F}, \epsilon) \leq \dim_E\left(\mathcal{F}, \alpha_T^{\mathcal{F}}\right) = d$, since $\dim_E(\mathcal{F}, \epsilon')$ is nonincreasing in $\epsilon'$. Therefore, when $w_{i_t} > \epsilon \geq \alpha_T^{\mathcal{F}}, t \leq \left(\frac{4\beta_T}{\epsilon^2} + 1\right) d$, which

implies $\epsilon \leq \sqrt{\frac{4\beta_T d}{t-d}}$. This shows that if $w_{i_t} > \alpha_T^{\mathcal{F}}$, then $w_{i_t} \leq \min\left\{C, \sqrt{\frac{4\beta_T d}{t-d}}\right\}$. Therefore,

$$\sum_{t=1}^{T} w_{i_t} \mathbf{1}\left\{w_{i_t} > \alpha_T^{\mathcal{F}}\right\} \leq dC + \sum_{t=d+1}^{T} \sqrt{\frac{4d\beta_T}{t-d}}$$

$$\leq dC + 2\sqrt{d\beta_T} \int_{t=0}^{T} \frac{1}{\sqrt{t}} dt$$

$$= dC + 4\sqrt{d\beta_T T}. \tag{125}$$

$\square$

**Lemma 12.** *(Optimism drives exploration, analog of Lemma 2). If the estimates $\hat{V}_f$ and $\tilde{\mathcal{E}}(f_t, \pi_t, h)$ in Line 3 and 8 of Algorithm 3 always satisfy*

$$\left|\hat{V}_f - V_f\right| \leq \epsilon'/8, \quad \left|\tilde{\mathcal{E}}(f_t, \pi_t, h) - \mathcal{E}(f_t, \pi_t, h)\right| \leq \frac{\epsilon'}{8H}, \tag{126}$$

*throughout the execution of the algorithm (recall that $\epsilon'$ is defined on Line 1), and $f_\theta^\star$ is never eliminated, then in any iteration $t$, either the algorithm does not terminate and*

$$\mathcal{E}(f_t, \pi_t, h_t) \geq \frac{\epsilon'}{2H} \tag{127}$$

*or the algorithm terminates and the output policy $\pi_t$ satisfies $V^{\pi_t} \geq V_{\mathcal{F},\theta}^\star - \epsilon' - H\theta$.*

Then, we bound the two terms above respectively. For the first term, we deduce that

$$p_1 \leq \sum_{(k,h):\sigma_{k,h}=1} \mathbb{E}_{\pi_k}\left[\max\left(1, \beta_{k,h}\right) \cdot \min\left(1, b_{k,h}(s_{k,h}, a_{k,h})\right)\right]$$

$$\leq \sqrt{\sum_{k=1}^{K}\sum_{h=1}^{H} \max\left(1, (\beta_{k,h})^2\right)} \cdot \mathbb{E}_{\pi_k}\left[\sqrt{\sum_{(k,h):\sigma_{k,h}=1} \min\left(1, (b_{k,h}(s_{k,h}, a_{k,h}))^2\right)}\right]$$

$$\leq \sqrt{KH}(1+\beta)\sqrt{\sum_{h=1}^{H} \sup_{Z_{K,h}} \sum_{k=1}^{K} \left(D_{\lambda,\sigma_h,\mathcal{F}_{k,h}}(Z_{k,h})\right)^2}, \tag{128}$$

where the first inequality is due to the fact that $\min(a_1 a_2, b_1 b_2) \leq \max(a_1, b_1) \cdot \min(a_2, b_2)$, the second inequality is obtained by using Cauchy-Schwarz inequality, and the last inequality utilizes the definition of $D_{\lambda,\sigma_h,\mathcal{F}_{k,h}}(Z_{k,h})$ in (13) and the selection of confidence radius: $\beta_{k,h} = \beta$.

Then, for $\sigma_{k,h} > 1$, according to the definition of $\sigma_{k,h}$ in (14), we have $(\sigma_{k,h})^2 = 1/\alpha \cdot b_{k,h}(s_{k,h}, a_{k,h})$. Thus, we can bound the second term as

$$p_2 \leq \sum_{(k,h):\sigma_{k,h}>1} \mathbb{E}_{\pi_k}\left[\min\left(1, \beta_{k,h}(\sigma_{k,h})^2 \cdot b_{k,h}(s_{k,h}, a_{k,h})/(\sigma_{k,h})^2\right)\right]$$

$$\leq \sum_{(k,h):\sigma_{k,h}>1} \mathbb{E}_{\pi_k}\left[\min\left(1, \beta_{k,h}/\alpha \cdot (b_{k,h}(s_{k,h}, a_{k,h}))^2/(\sigma_{k,h})^2\right)\right]$$

$$\leq \beta/\alpha \cdot \sum_{k=1}^{K}\sum_{h=1}^{H} \mathbb{E}_{\pi_k}\left[\min\left(1, (b_{k,h}(s_{k,h}, a_{k,h}))^2/(\sigma_{k,h})^2\right)\right]$$

$$\leq \beta/\alpha \cdot \sum_{h=1}^{H}\sum_{k=1}^{K} \mathbb{E}_{\pi_k}\left[\left(D_{\lambda,\sigma_h,\mathcal{F}_{k,h}}(Z_{k,h})\right)^2\right]$$

$$\leq \beta/\alpha \cdot \sum_{h=1}^{H} \sup_{Z_{K,h}} \sum_{k=1}^{K} \left(D_{\lambda,\sigma_h,\mathcal{F}_{k,h}}(Z_{k,h})\right)^2, \tag{129}$$

where the $D_{\lambda,\sigma_h,\mathcal{F}_{k,h}}(Z_{k,h})$ is formulated in Definition 13 . Combining these results, we get

$$
\begin{aligned}
\mathrm{Reg}(K) &\leq 2H\zeta + \sqrt{KH}(1+\beta)\sqrt{\sum_{h=1}^{H} \sup_{Z_{K,h}} \sum_{k=1}^{K} \left(D_{\lambda,\sigma_h,\mathcal{F}_{k,h}}(Z_{k,h})\right)^2} \\
&\qquad + \beta/\alpha \cdot \sum_{h=1}^{H} \sup_{Z_{K,h}} \sum_{k=1}^{K} \left(D_{\lambda,\sigma_h,\mathcal{F}_{k,h}}(Z_{k,h})\right)^2 \\
&= \tilde{\mathcal{O}}\left(\left(H + \sum_{h=1}^{H} \sup_{Z_{K,h}} \sum_{k=1}^{K} \left(D_{\lambda,\sigma_h,\mathcal{F}_{k,h}}(Z_{k,h})\right)^2\right)\zeta\right. \\
&\qquad + \sqrt{KH\ln\left(\mathcal{N}_K(\gamma)\right)\sum_{h=1}^{H} \sup_{Z_{K,h}} \sum_{k=1}^{K} \left(D_{\lambda,\sigma_h,\mathcal{F}_{k,h}}(Z_{k,h})\right)^2} \\
&\qquad + \alpha\zeta\sqrt{KH\sum_{h=1}^{H} \sup_{Z_{K,h}} \sum_{k=1}^{K} \left(D_{\lambda,\sigma_h,\mathcal{F}_{k,h}}(Z_{k,h})\right)^2} \\
&\qquad \left.+ \sqrt{\ln\left(\mathcal{N}_K(\gamma)\right)}\sum_{h=1}^{H} \sup_{Z_{K,h}} \sum_{k=1}^{K} \left(D_{\lambda,\sigma_h,\mathcal{F}_{k,h}}(Z_{k,h})\right)^2/\alpha\right) \\
&= \tilde{\mathcal{O}}\left(\sqrt{KH\ln\left(\mathcal{N}_K(\gamma)\right)\sum_{h=1}^{H} \sup_{Z_{K,h}} \sum_{k=1}^{K} \left(D_{\lambda,\sigma_h,\mathcal{F}_{k,h}}(Z_{k,h})\right)^2}\right. \\
&\qquad \left.+ \zeta\sum_{h=1}^{H} \sup_{Z_{K,h}} \sum_{k=1}^{K} \left(D_{\lambda,\sigma_h,\mathcal{F}_{k,h}}(Z_{k,h})\right)^2\right),
\end{aligned}
\tag{130}
$$

where the first inequality is deduced by taking the bounds of terms $p_1$ and $p_2$ back into Eq. (71), the first equality uses the choice of $\beta = \mathcal{O}\left(\alpha\zeta + \sqrt{\ln\left(H\ln\left(\mathcal{N}_K(\gamma)\right)/\delta\right)}\right)$, and the last equation is obtained by setting $\alpha = \sqrt{\ln\left(\mathcal{N}_K(\gamma)\right)}/\zeta$.

Then, it suffices to replace weighted eluder dimension $\sup_{Z_{K,h}} \sum_{k=1}^{K} \left(D_{\lambda,\sigma_h,\mathcal{F}_{k,h}}(Z_{k,h})\right)^2$ with the eluder dimension $\dim_E(\mathcal{F},\epsilon)$ in Definition 2.7. Because $\mathcal{F}$ is factorized as $\prod_{h=1}^{H} \mathcal{F}_h$, we get

$$
\dim_E(\mathcal{F},\epsilon) = \sum_{h=1}^{H} \dim_E(\mathcal{F}_h,\epsilon).
\tag{131}
$$

By invoking Lemma 5.1 for each function space $\mathcal{F}_h$, we obtain

$$
\sup_{Z_{K,h}} \sum_{k=1}^{K} \left(D_{\lambda,\sigma_h,\mathcal{F}_{k,h}}(Z_{k,h})\right)^2 \leq \left(\sqrt{8c_0}+3\right)\dim_E(\mathcal{F}_h,\lambda/K)\log(K/\lambda)\ln K,
\tag{132}
$$

which indicates that

$$
\sum_{h=1}^{H} \sup_{Z_{K,h}} \sum_{k=1}^{K} \left(D_{\lambda,\sigma_h,\mathcal{F}_{k,h}}(Z_{k,h})\right)^2 \leq \left(\sqrt{8c_0}+3\right)\dim_E(\mathcal{F},\lambda/K)\log(K/\lambda)\ln K.
\tag{133}
$$

## D    EXISTING IDEA: IMPORTANCE SAMPLING

For completeness, we repeat the discussion in existing importance sampling Wang et al. (2020).

**Assumption 1.** *For any $\varepsilon > 0$, the following holds:*

*1. there exists an $\varepsilon$-cover $\mathcal{C}(\mathcal{F},\varepsilon) \subseteq \mathcal{F}$ with size $|\mathcal{C}(\mathcal{F},\varepsilon)| \leq \mathcal{N}(\mathcal{F},\varepsilon)$, such that for any $f \in \mathcal{F}$, there exists $f' \in \mathcal{C}(\mathcal{F},\varepsilon)$ with $\|f - f'\|_\infty \leq \varepsilon$;*

*2. there exists an $\varepsilon$-cover $\mathcal{C}(\mathcal{S} \times \mathcal{A},\varepsilon)$ with size $|\mathcal{C}(\mathcal{S} \times \mathcal{A},\varepsilon)| \leq \mathcal{N}(\mathcal{S} \times \mathcal{A},\varepsilon)$, such that for any $(s,a) \in \mathcal{S} \times \mathcal{A}$, there exists $(s',a') \in \mathcal{C}(\mathcal{S} \times \mathcal{A},\varepsilon)$ with $\max_{f \in \mathcal{F}} |f(s,a) - f(s',a')| \leq \varepsilon$.*

---

**Algorithm 2** $\mathcal{F} - \text{LSVI}(\delta)$

---

1: **Input:** failure probability $\delta \in (0, 1)$ and number of episodes $K$
2: **for** episode $k = 1 : K$ **do**
3:     Receive initial state $s_{k,1} \sim \mu$
4:     $Q_{k,H+1}(\cdot, \cdot) \leftarrow 0$ and $V_{k,H+1}(\cdot) \leftarrow 0$
5:     $\mathcal{Z}_k \leftarrow \{(s_{t,h'}, a_{t,h'})\}_{(t,h') \in [k-1] \times [H]}$
6:     **for** $h = H : 1$ **do**
7:         $\mathcal{D}_{k,h} \leftarrow \{(s_{t,h'}, a_{t,h'}, r_{t,h'} + V_{k,H+1}(s_{t,h'+1}, a))\}_{(t,h') \in [k-1] \times [H]}$
8:         $f_{k,h} \leftarrow \arg\min_{f \in \mathcal{F}} \|f\|^2_{\mathcal{D}_{k,h}}$
9:         $b_{k,h}(\cdot, \cdot) \leftarrow \text{Bonus}\,(\mathcal{F}, f_{k,h}, \mathcal{Z}_k, \delta)$ (Algorithm 3)
10:        $Q_{k,h}(\cdot, \cdot) \leftarrow \min\{f_{k,h}(\cdot, \cdot) + b_{k,h}(\cdot, \cdot), H\}$ and $V_{k,h}(\cdot) = \max_{a \in \mathcal{A}} Q_{k,h}(\cdot, a)$
11:        $\pi_{k,h}(\cdot) \leftarrow \arg\max_{a \in \mathcal{A}} Q_{k,h}(\cdot, a)$
12:     **for** h=1:H **do**
13:         Take action $a_{k,h} \leftarrow \pi_{k,h}(s_{k,h})$ and observe $s_{k,h+1} \sim P(\cdot \mid s_{k,h}, a_{k,h})$ and $r_{k,h} = r(s_{k,h}, a_{k,h})$
14:     **end for**
15:     **end for**
16: **end for**

---

Assumption 1 requires both the function class $\mathcal{F}$ and the state-action pairs $\mathcal{S} \times \mathcal{A}$ have bounded covering numbers. Since our regret bound depends logarithmically on $\mathcal{N}(\mathcal{F}, \cdot)$ and $\mathcal{N}(\mathcal{S} \times \mathcal{A}, \cdot)$, it is acceptable for the covers to have exponential size. In particular, when $\mathcal{S}$ and $\mathcal{A}$ are finite, it is clear that $\log \mathcal{N}(\mathcal{F}, \varepsilon) = \widetilde{O}(|\mathcal{S}||\mathcal{A}|)$ and $\log \mathcal{N}(\mathcal{S} \times \mathcal{A}, \varepsilon) = \log(|\mathcal{S}||\mathcal{A}|)$. For the case of $d$-dimensional linear functions and generalized linear functions, $\log \mathcal{N}(\mathcal{F}, \varepsilon) = \widetilde{O}(d)$ and $\log \mathcal{N}(\mathcal{S} \times \mathcal{A}, \varepsilon) = \widetilde{O}(d)$. For quadratic functions, $\log \mathcal{N}(\mathcal{F}, \varepsilon) = \widetilde{O}(d^2)$ and $\log \mathcal{N}(\mathcal{S} \times \mathcal{A}, \varepsilon) = \widetilde{O}(d)$.

### D.1 ALGORITHM OVERVIEW

*Stable Upper-Confidence Bonus Function.* With more collected data, the least squares predictor is expected to return a better approximate the true $Q$-function. To encourage exploration, we carefully design a bonus function $b_{k,h}(\cdot, \cdot)$ which guarantees that, with high probability, $Q_{k,h+1}(s, a)$ is an overestimate of the one-step backup. The bonus function $b_{k,h}(\cdot, \cdot)$ is guaranteed to tightly characterize the estimation error of the one-step backup

$$r(\cdot, \cdot) + \sum_{s' \in \mathcal{S}} P(s' \mid \cdot, \cdot) V_{k,h+1}(s'), \tag{134}$$

where

$$V_{k,h+1}(\cdot) = \max_{a \in \mathcal{A}} Q_{k,h+1}(\cdot, a) \tag{135}$$

is the value function of the next step. The bonus function $b_{k,h}(\cdot, \cdot)$ is designed by carefully prioritizing important data and hence is stable even when the replay buffer has large cardinality.

### D.1.1 STABLE UCB VIA IMPORTANCE SAMPLING

To define the confidence region $\mathcal{F}_{k,h}$, a natural definition would be

$$\mathcal{F}_{k,h} = \left\{ f \in \mathcal{F} \mid \|f - f_{k,h}\|^2_{\mathcal{Z}_k} \leq \beta \right\}, \tag{136}$$

where $\beta$ is defined so that

$$r(\cdot, \cdot) + \sum_{s' \in \mathcal{S}} P(s' \mid \cdot, \cdot) V_{k,H+1}(s') \in \mathcal{F}_{k,h}. \tag{137}$$

with high probability, and recall that $\mathcal{Z}_k = \{(s_{t,h'}, a_{t,h'})\}_{(t,h') \in [k-1] \times [H]}$ is the set of state-action pairs defined in Line 5. However, as one can observe, the complexity of such a bonus function

---

**Algorithm 3** Sensitivity-Sampling $(\mathcal{F}, \mathcal{Z}, \lambda, \varepsilon, \delta)$

---

1: **Input:** function class $\mathcal{F}$, set of state-action pairs $\mathcal{Z} \subseteq \mathcal{S} \times \mathcal{A}$, accuracy parameters $\lambda, \varepsilon > 0$ and failure probability $\delta \in (0, 1)$
2: Initialize $\mathcal{Z}' \leftarrow \{\}$
3: For each $z \in \mathcal{Z}$, let $p_z$ to be smallest real number such that $1/p_z$ is an integer and

$$p_z \geq \min \left\{1, sensitivity_{\mathcal{Z}, \mathcal{F}, \lambda}(z) \cdot 72 \ln(4\mathcal{N}(\mathcal{F}, \varepsilon/72 \cdot \sqrt{\lambda\delta/(|\mathcal{Z}|)})/\delta)/\varepsilon^2 \right\}. \tag{138}$$

4: For each $z \in \mathcal{Z}$, independently add $1/p_z$ copies of $z$ into $\mathcal{Z}'$ with probability $p_z$
5: return $Z'$

---

**Algorithm 4** Bonus$(\mathcal{F}, \bar{f}, \mathcal{Z}, \delta)$

---

1: **Input:** function class $\mathcal{F}$, reference function $\bar{f} \in \mathcal{F}$, state-action pairs $\mathcal{Z} \subseteq \mathcal{S} \times \mathcal{A}$ and failure probability $\delta \in (0, 1)$
2: $\mathcal{Z}' \leftarrow$ Sensitivity-Sampling$(\mathcal{F}, \mathcal{Z}, \delta/(16T), 1/2, \delta)$ $\triangleright$
3: $\mathcal{Z}' \leftarrow \{\}$ if $|\mathcal{Z}'| \geq 4T/\delta$ or the number of distinct elements in $\mathcal{Z}'$ exceeds

$$6912 \dim_E \left(\mathcal{F}, \delta/\left(16T^2\right)\right) \log \left(64H^2T^2/\delta\right) \ln T \ln(4\mathcal{N}(\mathcal{F}, \delta/(566T))/\delta). \tag{140}$$

4: Let $\widehat{f} \in \mathcal{C}(\mathcal{F}, 1/(8\sqrt{4T/\delta}))$ be such that $\|\bar{f} - \widehat{f}\|_\infty \leq 1/(8\sqrt{4T/\delta})$
5: $\widehat{\mathcal{Z}} \leftarrow \{\}$
6: **for** $z \in \mathcal{Z}'$ **do**
7:     Let $\widehat{z} \in \mathcal{C}(\mathcal{S} \times \mathcal{A}, 1/(8\sqrt{4T/\delta}))$ be such that $\sup_{f, f' \in \mathcal{F}} |f(z) - f'(z)| \leq 1/(8\sqrt{4T/\delta})$
8:     $\widehat{\mathcal{Z}} \leftarrow \widehat{\mathcal{Z}} \cup \{\widehat{z}\}$
9:     return $\widehat{w}(\cdot, \cdot) := w(\widehat{\mathcal{F}}, \cdot, \cdot)$, where $\widehat{\mathcal{F}} = \left\{f \in \mathcal{F} \mid \|f - \widehat{f}\|_{\widehat{\mathcal{Z}}}^2 \leq 3\beta(\mathcal{F}, \delta) + 2\right\}$ and

$$\beta(\mathcal{F}, \delta) = c'H^2 \cdot \log^2(T/\delta) \cdot \dim_E \left(\mathcal{F}, \delta/T^3\right)$$
$$\cdot \ln \left(\mathcal{N} \left(\mathcal{F}, \delta/T^2\right)/\delta\right) \cdot \log(\mathcal{N}(\mathcal{S} \times \mathcal{A}, \delta/T)) \cdot T/\delta \tag{141}$$

    for some absolute constants $c' > 0$.
10: **end for**

---

could be extremely high as it is defined by a dataset $\mathcal{Z}_k$ whose size can be as large as $T = KH$. A high-complexity bonus function could potentially introduce instability issues in the algorithm. Technically, we require a stable bonus function to allow for highly concentrated estimate of the one-step backup so that the confidence region $\mathcal{F}_{k,h}$ is accurate even for bounded $\beta$. Our strategy to "stabilize" the bonus function is to reduce the size of the dataset by importance sampling, so that only important state-action pairs are kept and those unimportant ones (which potentially induce instability) are ignored. Another benefit of reducing the size of the dataset is that it leads to superior computational complexity when evaluating the bonus function in practice. In later part of this section, we introduce an approach to estimate the importance of each state-action pair and a corresponding sampling method based on that.

**Definition 4.** *For a given set of state-action pairs $\mathcal{Z} \subseteq \mathcal{S} \times \mathcal{A}$ and a function class $\mathcal{F}$, for each $z \in \mathcal{Z}$, define the $\lambda$-sensitivity of $(s, a)$ with respect to $\mathcal{Z}$ and $\mathcal{F}$ to be*

$$\text{sensitivity}_{\mathcal{Z}, \mathcal{F}, \lambda}(s, a) = \max_{\substack{f, f' \in \mathcal{F} \\ \|f - f'\|_{\mathcal{Z}}^2 \geq \lambda}} \frac{(f(s, a) - f'(s, a))^2}{\|f - f'\|_{\mathcal{Z}}^2}. \tag{139}$$

Sensitivity measures the importance of each data point $z$ in $\mathcal{Z}$ by considering the pair of functions $f, f' \in \mathcal{F}$ such that $z$ contributes the most to $\|f - f'\|_{\mathcal{Z}}^2$.

### D.2 Computational Efficiency

To implement importance sampling, one needs to evaluate the width function $w(\widehat{\mathcal{F}}, \cdot, \cdot)$ for a confidence region $\widehat{\mathcal{F}}$ of the form

$$\widehat{\mathcal{F}} = \left\{ f \in \mathcal{F} \mid \|f - \widehat{f}\|_{\mathcal{Z}}^2 \leq \beta \right\}, \tag{142}$$

which is a constrained optimization problem. When $\mathcal{F}$ is the class of linear functions, there is a closed-form formula for the width function and thus the width function can be efficiently evaluated in this case. Simple complexity upper bound is no longer available for the class of general functions considered in this paper. Instead, we bound the complexity of the bonus function by relying on the fact that the subsampled dataset has bounded size. Scrutinizing the sampling algorithm, it can be seen that the size of the subsampled dataset is upper bounded by the sum of the sensitivity of the data points in the given dataset times the log-convering number of the function class $\mathcal{F}$. To upper bound the sum of the sensitivity of the data points in the given dataset, we rely on a novel combinatorial argument which establishes a surprising connection between the sum of the sensitivity and the eluder dimension of the function class $\mathcal{F}$. We show that the sum of the sensitivity of data points is upper bounded by the eluder dimension of the dataset up to logarithm factors. Hence, the complexity of the subsampled dataset, and therefore, the complexity of the bonus function, is upper bound by the log-covering number of $\mathcal{S} \times \mathcal{A}$ (the complexity of each state-action pair) times the product of the eluder dimension of the function class and the log-covering number of the function class (the number of data points in the subsampled dataset).

In order to show that the confidence region is approximately preserved when using the subsampled dataset $\mathcal{Z}'$, we show that for any $f, f' \in \mathcal{F}, \|f - f'\|_{\mathcal{Z}'}^2$ is a good approximation to $\|f - f'\|_{\mathcal{Z}}^2$. To show this, we apply a union bound over all pairs of functions on the cover of $\mathcal{F}$ which allows us to consider fixed $f, f' \in \mathcal{F}$. For fixed $f, f' \in \mathcal{F}$, note that $\|f - f'\|_{\mathcal{Z}'}^2$ is an unbiased estimate of $\|f - f'\|_{\mathcal{Z}}^2$, and importance sampling proportional to the sensitivity implies an upper bound on the variance of the estimator which allows us to apply concentration bounds to prove the desired result. We note that the sensitivity sampling framework used here is very crucial to the theoreical guarantee of the algorithm. If one replaces sensitivity sampling with more naïve sampling approaches (e.g. uniform sampling), then the required sampling size would be much larger, which does not give any meaningful reduction on the size of the dataset and also leads to a high complexity bonus function.

Our algorithm applies the principle of optimism in the face of uncertainty (OFU) to balance exploration and exploitation. Note that $V_{k,h+1}$ is the value function estimated at step $h + 1$. In our analysis, we require the $Q$-function $Q_{k,h}$ estimated at level $h$ to satisfy

$$Q_{k,h}(\cdot, \cdot) \geq r(\cdot, \cdot) + \sum_{s' \in \mathcal{S}} P(s' \mid \cdot, \cdot) V_{k,h+1}(s') \tag{143}$$

with high probability. To achieve this, we optimize the least squares objective to find a solution $f_{k,h} \in \mathcal{F}$ using collected data. We then show that $f_{k,h}$ is close to $r(\cdot, \cdot) + \sum_{s' \in \mathcal{S}} P(s' \mid \cdot, \cdot) V_{k,h+1}(s')$. This would follow from standard analysis if the collected samples were independent of $V_{k,h+1}$. However, $V_{k,h+1}$ is calculated using the collected samples and thus they are subtly dependent on each other. To tackle this issue, we notice that $V_{k,h+1}$ is computed by using $f_{k,h+1}$ and the bonus function $b_{k,h+1}$, and both $f_{k,h+1}$ and the bonus function $b_{k,h+1}$ have bounded complexity, thanks to the design of bonus function. Hence, we can construct a $1/T$-cover to approximate $V_{k,h+1}$. By doing so, we can now bound the fitting error of $f_{k,h}$ by replacing $V_{k,h+1}$ with its closest neighbor in the $1/T$-cover which is independent of the dataset. By a union bound over all functions in the $1/T$-cover, it follows that with high probability,

$$r(\cdot, \cdot) + \sum_{s' \in \mathcal{S}} P(s' \mid \cdot, \cdot) V_{k,h+1}(s') \in \left\{ f \in \mathcal{F} \mid \|f - f_{k,h}\|_{\mathcal{Z}_k}^2 \leq \beta \right\} \tag{144}$$

for some $\beta$ that depends only on the complexity of the bonus function and the function class $\mathcal{F}$.

### D.3 Analysis of the Stable Bonus Function

Our first lemma gives an upper bound on the sum of the sensitivity in terms of the eluder dimension of the function class $\mathcal{F}$.

**Lemma 13.** *For a given set of state-action pairs $\mathcal{Z}$,*

$$\sum_{z \in \mathcal{Z}} \text{sensitivity}_{\mathcal{Z}, \mathcal{F}, \lambda}(z) \le 4 \dim_E(\mathcal{F}, \lambda/|\mathcal{Z}|) \log\left((H+1)^2 |\mathcal{Z}|/\lambda\right) \ln |\mathcal{Z}|. \tag{145}$$

*Proof.* For each $z \in \mathcal{Z}$, let $f, f' \in F$ be an arbitrary pair of functions such that $\|f - f'\|_{\mathcal{Z}}^2 \ge \lambda$ and

$$\frac{(f(z) - f'(z))^2}{\|f - f'\|_{\mathcal{Z}}^2} \tag{146}$$

is maximized, and we define $L(z) = (f(z) - f'(z))^2$ for such $f$ and $f'$. Note that $0 \le L(z) \le (H+1)^2$. Let $\mathcal{Z} = \bigcup_{\alpha=0}^{\log\left((H+1)^2 |\mathcal{Z}|/\lambda\right)-1} \mathcal{Z}^\alpha \cup \mathcal{Z}^\infty$ be a dyadic decomposition with respect to $L(\cdot)$, where for each $0 \le \alpha < \log\left((H+1)^2 |\mathcal{Z}|/\lambda\right)$, define

$$\mathcal{Z}^\alpha = \left\{ z \in \mathcal{Z} \mid L(z) \in \left((H+1)^2 \cdot 2^{-\alpha-1}, (H+1)^2 \cdot 2^{-\alpha}\right] \right\} \tag{147}$$

and

$$\mathcal{Z}^\infty = \{ z \in \mathcal{Z} \mid L(z) \le \lambda/|\mathcal{Z}| \} \tag{148}$$

Clearly, for any $z \in \mathcal{Z}^\infty$, $\text{sensitivity}_{\mathcal{Z}, \mathcal{F}, \lambda}(z) \le 1/|\mathcal{Z}|$ and thus

$$\sum_{z \in \mathcal{Z}^\infty} \text{sensitivity}_{\mathcal{Z}, \mathcal{F}, \lambda}(z) \le 1. \tag{149}$$

Now we bound $\sum_{z \in \mathcal{Z}^\alpha} \text{sensitivity}_{\mathcal{Z}, \mathcal{F}, \lambda}(z)$ for each $0 \le \alpha < \log\left((H+1)^2 |\mathcal{Z}|/\lambda\right)$ separately. For each $\alpha$, let

$$N_\alpha = |\mathcal{Z}^\alpha| / \dim_E\left(\mathcal{F}, (H+1)^2 \cdot 2^{-\alpha-1}\right), \tag{150}$$

and we decompose $\mathcal{Z}^\alpha$ into $N_\alpha + 1$ disjoint subsets, i.e., $\mathcal{Z}^\alpha = \bigcup_{j=1}^{N_\alpha+1} \mathcal{Z}_j^\alpha$, by using the following procedure. Let $\mathcal{Z}^\alpha = \left\{ z_1, z_2, \ldots, z_{|\mathcal{Z}^\alpha|} \right\}$ and we consider each $z_i$ sequentially. Initially $\mathcal{Z}_j^\alpha = \{\}$ for all $j$. Then, for each $z_i$, we find the largest $1 \le j \le N_\alpha$ such that $z_i$ is $(H+1)^2 \cdot 2^{-\alpha-1}$-independent of $\mathcal{Z}_j^\alpha$ with respect to $\mathcal{F}$. We set $j = N_\alpha + 1$ if such $j$ does not exist, and use $j(z_i) \in [N_\alpha + 1]$ to denote the choice of $j$ for $z_i$. By the design of the algorithm, for each $z_i$, it is clear that $z_i$ is dependent on each of $\mathcal{Z}_1^\alpha, \mathcal{Z}_2^\alpha, \ldots, \mathcal{Z}_{j(z_i)-1}^\alpha$

Now we show that for each $z_i \in \mathcal{Z}^\alpha$,

$$\text{sensitivity}_{\mathcal{Z}, \mathcal{F}, \lambda}(z_i) \le 2/j(z_i). \tag{151}$$

For any $z_i \in \mathcal{Z}^\alpha$, we use $f, f' \in F$ to denote the pair of functions in $\mathcal{F}$ such that $\|f - f'\|_{\mathcal{Z}}^2 \ge \lambda$ and

$$\frac{(f(z_i) - f'(z_i))^2}{\|f - f'\|_{\mathcal{Z}}^2} \tag{152}$$

is maximized. Since $z_i \in \mathcal{Z}^\alpha$, we must have $(f(z_i) - f'(z_i))^2 > (H+1)^2 \cdot 2^{-\alpha-1}$. Since $z_i$ is dependent on each of $\mathcal{Z}_1^\alpha, \mathcal{Z}_2^\alpha, \ldots, \mathcal{Z}_{j(z_i)-1}^\alpha$, for each $1 \le k < j(z_i)$, we have

$$\|f - f'\|_{\mathcal{Z}_k^\alpha} \ge (H+1)^2 \cdot 2^{-\alpha-1} \tag{153}$$

which implies

$$\text{sensitivity}_{\mathcal{Z}, \mathcal{F}, \lambda}(z_i) = \frac{(f(z_i) - f'(z_i))^2}{\|f - f'\|_{\mathcal{Z}}^2} \le \frac{(H+1)^2 \cdot 2^{-\alpha}}{\|f - f'\|_{\mathcal{Z}}^2}$$

$$\le \frac{(H+1)^2 \cdot 2^{-\alpha}}{\sum_{k=1}^{j(z_i)-1} \|f - f'\|_{\mathcal{Z}_k^\alpha} + (f(z_i) - f'(z_i))^2} \le 2/j(z_i). \tag{154}$$

Moreover, by the definition of $(H + 1)^2 \cdot 2^{-\alpha-1}$-independence, we have $\left| \mathcal{Z}_j^\alpha \right| \leq \dim_E \left( \mathcal{F}, (H+1)^2 \cdot 2^{-\alpha-1} \right)$ for all $1 \leq j \leq N_\alpha$. Therefore,

$$\sum_{z \in \mathcal{Z}^\alpha} \text{sensitivity}_{\mathcal{Z},\mathcal{F},\lambda}(z) \leq \sum_{1 \leq j \leq N_\alpha} \left| \mathcal{Z}_j^\alpha \right| \cdot 2/j + \sum_{z \in \mathcal{Z}_{N_\alpha+1}^\alpha} 2/N_\alpha$$

$$\leq 2 \dim_E \left( \mathcal{F}, (H+1)^2 \cdot 2^{-\alpha-1} \right) \ln(N_\alpha) + |\mathcal{Z}^\alpha| \cdot \frac{2 \dim_E \left( \mathcal{F}, (H+1)^2 \cdot 2^{-\alpha-1} \right)}{|\mathcal{Z}^\alpha|}$$

$$\leq \dim_E \left( \mathcal{F}, (H+1)^2 \cdot 2^{-\alpha-1} \right) \ln(|\mathcal{Z}|). \tag{155}$$

By the monotonicity of eluder dimension, it follows that

$$\sum_{z \in \mathcal{Z}} \text{sensitivity}_{\mathcal{Z},\mathcal{F},\lambda}(z)$$

$$\leq \sum_{\alpha=0}^{\log\left((H+1)^2|\mathcal{Z}|/\lambda\right)-1} \sum_{z \in \mathcal{Z}^\alpha} \text{sensitivity}_{\mathcal{Z},\mathcal{F},\lambda}(z) + \sum_{z \in \mathcal{Z}^\infty} \text{sensitivity}_{\mathcal{Z},\mathcal{F},\lambda}(z)$$

$$\leq 3 \log \left( (H+1)^2|\mathcal{Z}|/\lambda \right) \dim_E(\mathcal{F}, \lambda/|\mathcal{Z}|) \ln(|\mathcal{Z}|) + 1$$

$$\leq 4 \log \left( (H+1)^2|\mathcal{Z}|/\lambda \right) \dim_E(\mathcal{F}, \lambda/|\mathcal{Z}|) \ln(|\mathcal{Z}|). \tag{156}$$

$\square$

Using Lemma 13, we can prove an upper bound on the number of distinct elements in $\mathcal{Z}'$ returned by the sampling algorithm (Algorithm 23).

**Lemma 14.** *With probability at least $1 - \delta/4$, the number of distinct elements in $\mathcal{Z}'$ returned by Algorithm 2 is at most*

$$1728 \dim_E(\mathcal{F}, \lambda/|\mathcal{Z}|) \log \left( (H+1)^2|\mathcal{Z}|/\lambda \right) \ln(|\mathcal{Z}|) \ln(4\mathcal{N}(\mathcal{F}, \varepsilon/72 \cdot \sqrt{\lambda\delta/(|\mathcal{Z}|)})/\delta)/\varepsilon^2. \tag{157}$$

*Proof.* Note that

$$p_z \leq \min \left\{ 1, 2 \cdot \text{sensitivity}_{\mathcal{Z},\mathcal{F},\lambda}(z) \cdot 72 \ln(4\mathcal{N}(\mathcal{F}, \varepsilon/72 \cdot \sqrt{\lambda\delta/(|\mathcal{Z}|)})/\delta)/\varepsilon^2 \right\}, \tag{158}$$

since for any real number $x < 1$, there always exists $\widehat{x} \in [x, 2x]$ such that $1/\widehat{x}$ is an integer. Let $X_z$ be a random variable defined as

$$X_z = \begin{cases} 1 & z \in Z' \\ 0 & z \notin Z' \end{cases}. \tag{159}$$

Clearly, the number of distinct elements in $\mathcal{Z}'$ is upper bounded by $\sum_{z \in \mathcal{Z}} X_z$ and $\mathbb{E}[X_z] = p_z$. By Lemma 13,

$$\sum_{z \in \mathcal{Z}} \mathbb{E}[X_z]$$

$$\leq 576 \dim_E(\mathcal{F}, \lambda/|\mathcal{Z}|) \log \left( (H+1)^2|\mathcal{Z}|/\lambda \right) \ln(|\mathcal{Z}|) \ln(4\mathcal{N}(\mathcal{F}, \varepsilon/72 \cdot \sqrt{\lambda\delta/(|\mathcal{Z}|)})/\delta)/\varepsilon^2. \tag{160}$$

By Chernoff bound, with probability at least $1 - \delta/4$, we have

$$\sum_{z \in \mathcal{Z}} X_z$$

$$\geq 1728 \dim_E(\mathcal{F}, \lambda/|\mathcal{Z}|) \log \left( (H+1)^2|\mathcal{Z}|/\lambda \right) \ln(|\mathcal{Z}|) \ln(4\mathcal{N}(\mathcal{F}, \varepsilon/72 \cdot \sqrt{\lambda\delta/(|\mathcal{Z}|)})/\delta)/\varepsilon^2. \tag{161}$$

$\square$

Our second lemma upper bounds the number of elements in $\mathcal{Z}'$ returned by Algorithm 2.

**Lemma 15.** *With probability at least* $1 - \delta/4$, $|\mathcal{Z}'| \le 4|\mathcal{Z}|/\delta$.

*Proof.* Let $X_z$ be the random variable which is defined as

$$X_z = \begin{cases} 1/p_z & z \text{ is added into } \mathcal{Z}' \\ 0 & \text{otherwise} \end{cases}. \tag{162}$$

Note that $|\mathcal{Z}'| = \sum_{z \in \mathcal{Z}} X_z$ and $\mathbb{E}[X_z] = 1$. By Markov inequality, with probability $1 - \delta/4$, $|\mathcal{Z}'| \le 4|\mathcal{Z}|/\delta$.

$\square$

Our third lemma shows that for the given set of state-action pairs $\mathcal{Z}$ and function class $\mathcal{F}$, Algorithm 2 returns a set of state-action pairs $\mathcal{Z}'$ so that $\|f - f'\|_{\mathcal{Z}}^2$ is approximately preserved for all $f, f' \in \mathcal{F}$.

**Lemma 16.** *With probability at least* $1 - \delta/2$, *for any* $f, f' \in \mathcal{F}$,

$$(1 - \varepsilon)\|f - f'\|_{\mathcal{Z}}^2 - 2\lambda \le \|f - f'\|_{\mathcal{Z}'}^2 \le (1 + \varepsilon)\|f - f'\|_{\mathcal{Z}}^2 + 8|\mathcal{Z}|\lambda/\delta. \tag{163}$$

*Proof.* In our proof, we separately consider two cases: $\|f - f'\|_{\mathcal{Z}}^2 < 2\lambda$ and $\|f - f'\|_{\mathcal{Z}}^2 \ge 2\lambda$.

**Case I:** $\|f - f'\|_{\mathcal{Z}}^2 < 2\lambda$. Consider $f, f' \in \mathcal{F}$ with $\|f - f'\|_{\mathcal{Z}}^2 < 2\lambda$. Conditioned on the event defined in Lemma 15 which holds with probability at least $1 - \delta/4$, we have $\|f - f'\|_{\mathcal{Z}'}^2 \le |\mathcal{Z}'| \cdot \|f - f'\|_{\mathcal{Z}}^2 \le 8|\mathcal{Z}|\lambda/\delta$. Moreover, we always have $\|f - f'\|_{\mathcal{Z}'} \ge 0$. In summary, we have

$$\|f - f'\|_{\mathcal{Z}}^2 - 2\lambda \le \|f - f'\|_{\mathcal{Z}'}^2 \le \|f - f'\|_{\mathcal{Z}}^2 + 8|\mathcal{Z}|\lambda/\delta. \tag{164}$$

**Case II:** $\|f - f'\|_{\mathcal{Z}}^2 \ge 2\lambda$. We first show that for any fixed $f, f' \in \mathcal{F}$ with $\|f - f'\|_{\mathcal{Z}}^2 \ge \lambda$, with probability at least $1 - \delta/(4\mathcal{N}(\mathcal{F}, \varepsilon/72 \cdot \sqrt{\lambda\delta/(|\mathcal{Z}|)}))$, we have

$$(1 - \varepsilon/4)\|f - f'\|_{\mathcal{Z}}^2 \le \|f - f'\|_{\mathcal{Z}'}^2 \le (1 + \varepsilon/4)\|f - f'\|_{\mathcal{Z}}^2. \tag{165}$$

To prove this, for each $z \in \mathcal{Z}$, define

$$X_z = \begin{cases} \frac{1}{p_z}(f(z) - f'(z))^2 & z \text{ is added into } \mathcal{Z}' \text{ for } 1/p_z \text{ times} \\ 0 & \text{otherwise} \end{cases}. \tag{166}$$

Clearly, $\|f - f'\|_{\mathcal{Z}'} = \sum_{z \in \mathcal{Z}} X_z$ and $\mathbb{E}[X_z] = (f(z) - f'(z))^2$. Moreover, since $\|f - f'\|_{\mathcal{Z}}^2 \ge \lambda$, by (3) and Definition 3, we have

$$\max_{z \in \mathcal{Z}} X_z \le \|f - f'\|_{\mathcal{Z}}^2 \cdot \varepsilon^2/(72 \ln(4\mathcal{N}(\mathcal{F}, \varepsilon/72 \cdot \sqrt{\lambda\delta/(|\mathcal{Z}|)})/\delta). \tag{167}$$

Moreover, $\mathbb{E}[X_z^2] \le (f(z) - f'(z))^4/p_z$. Therefore, by Hölder's inequality,

$$\sum_{z \in \mathcal{Z}} \text{Var}[X_z] \le \sum_{z \in \mathcal{Z}} \mathbb{E}[X_z^2] \le \sum_{z \in \mathcal{Z}} (f(z) - f'(z))^2 \cdot \max_{z \in \mathcal{Z}} (f(z) - f'(z))^2/p_z$$

$$\le \|f - f'\|_{\mathcal{Z}}^4 \cdot \varepsilon^2/(72 \ln(4\mathcal{N}(\mathcal{F}, \varepsilon/72 \cdot \sqrt{\lambda\delta/(|\mathcal{Z}|)})/\delta). \tag{168}$$

Therefore, by Bernstein inequality,

$$\Pr\left[\left|\|f - f'\|_{\mathcal{Z}}^2 - \|f - f'\|_{\mathcal{Z}'}^2\right| \ge \varepsilon/4 \cdot \|f - f'\|_{\mathcal{Z}}^2\right]$$

$$= \Pr\left[\left|\sum_{z \in \mathcal{Z}} \mathbb{E}[X_z] - \sum_{z \in \mathcal{Z}} X_z\right| \ge \varepsilon/4 \cdot \|f - f'\|_{\mathcal{Z}}^2\right]$$

$$\le 2\exp\left(-\frac{\varepsilon^2/16 \cdot \|f - f'\|_{\mathcal{Z}}^4}{2\sum_{z \in \mathcal{Z}} \text{Var}[X_z] + 2\max_{z \in \mathcal{Z}} X_z \cdot \varepsilon/4 \cdot \|f - f'\|_{\mathcal{Z}}^2/3}\right)$$

$$\le (\delta/4)/(\mathcal{N}(\mathcal{F}, \varepsilon/72 \cdot \sqrt{\lambda\delta/(|\mathcal{Z}|)}))^2. \tag{169}$$

By union bound, the above inequality implies that with probability at least $1 - \delta/4$, for any $(f, f') \in \mathcal{C}(F, \varepsilon/72 \cdot \sqrt{\lambda\delta/(|\mathcal{Z}|)}) \times \mathcal{C}(F, \varepsilon/72 \cdot \sqrt{\lambda\delta/(|\mathcal{Z}|)})$ with $\|f - f'\|_{\mathcal{Z}}^2 \geq \lambda$

$$(1 - \varepsilon/4) \|f - f'\|_{\mathcal{Z}}^2 \leq \|f - f'\|_{\mathcal{Z}'}^2 \leq (1 + \varepsilon/4) \|f - f'\|_{\mathcal{Z}'}^2. \tag{170}$$

Now we condition on the event defined above and the event defined in Lemma 15. Consider $f, f' \in \mathcal{F}$ with $\|f - f'\|_{\mathcal{Z}}^2 \geq 2\lambda$. Recall that there exists

$$\left(\widehat{f}, \widehat{f'}\right) \in \mathcal{C}(F, \varepsilon/72 \cdot \sqrt{\lambda\delta/(|\mathcal{Z}|)}) \times \mathcal{C}(F, \varepsilon/72 \cdot \sqrt{\lambda\delta/(|\mathcal{Z}|)}). \tag{171}$$

such that $\|f - \widehat{f}\|_\infty \leq \sqrt{\lambda/(25|\mathcal{Z}|)}$ and $\left\|f' - \widehat{f'}\right\|_\infty \leq \sqrt{\lambda/(25|\mathcal{Z}|)}$. Therefore,

$$\left\|\widehat{f} - \widehat{f'}\right\|_{\mathcal{Z}}^2 = \sum_{z \in \mathcal{Z}} \left(\widehat{f}(z) - \widehat{f'}(z)\right)^2$$

$$= \sum_{z \in \mathcal{Z}} \left(f(z) - f'(z) + (\widehat{f}(z) - f(z)) + \left(f'(z) - \widehat{f'}(z)\right)\right)^2$$

$$\geq \left(\|f - f'\|_{\mathcal{Z}} - \|\widehat{f} - f\|_{\mathcal{Z}} - \left\|f' - \widehat{f'}\right\|_{\mathcal{Z}}\right)^2$$

$$\geq (\sqrt{2\lambda} - 2\sqrt{\lambda/25})^2 \geq \lambda. \tag{172}$$

Therefore, conditioned on the event defined above, we have

$$(1 - \varepsilon/4) \left\|\widehat{f} - \widehat{f'}\right\|_{\mathcal{Z}}^2 \leq \left\|\widehat{f} - \widehat{f'}\right\|_{\mathcal{Z}'}^2 \leq (1 + \varepsilon/4) \left\|\widehat{f} - \widehat{f'}\right\|_{\mathcal{Z}'}^2. \tag{173}$$

Conditioned on the event defined in Lemma 15 which holds with probability at least $1 - \delta/4$, we have

$$\|f - f'\|_{\mathcal{Z}'}^2 \leq \left(\left\|\widehat{f} - \widehat{f'}\right\|_{\mathcal{Z}'} + \|f - \widehat{f}\|_{\mathcal{Z}'} + \left\|f' - \widehat{f'}\right\|_{\mathcal{Z}'}\right)^2$$

$$\leq \left(\left\|\widehat{f} - \widehat{f'}\right\|_{\mathcal{Z}'} + 2\sqrt{|\mathcal{Z}'|} \cdot \varepsilon/72 \cdot \sqrt{\lambda\delta/(|\mathcal{Z}|)}\right)^2$$

$$\leq \left((1 + \varepsilon/6) \left\|\widehat{f} - \widehat{f'}\right\|_{\mathcal{Z}} + 2\sqrt{|\mathcal{Z}'|} \cdot \varepsilon/72 \cdot \sqrt{\lambda\delta/(|\mathcal{Z}|)}\right)^2$$

$$\leq \left((1 + \varepsilon/6)\|f - f\|_{\mathcal{Z}} + 2\sqrt{|\mathcal{Z}'|} \cdot \varepsilon/72 \cdot \sqrt{\lambda\delta/(|\mathcal{Z}|)} + 4\sqrt{|\mathcal{Z}|} \cdot \varepsilon/72 \cdot \sqrt{\lambda\delta/(|\mathcal{Z}|)}\right)^2$$

$$\leq (1 + \varepsilon)\|f - f\|_{\mathcal{Z}}^2, \tag{174}$$

where the last inequality holds since $\|f - f\|_{\mathcal{Z}} \geq \sqrt{\lambda}$. Similarly,

$$\|f - f'\|_{\mathcal{Z}'}^2 \geq \left(\left\|\widehat{f} - \widehat{f'}\right\|_{\mathcal{Z}'} - \|f - \widehat{f}\|_{\mathcal{Z}'} - \left\|f' - \widehat{f'}\right\|_{\mathcal{Z}'}\right)^2$$

$$\geq \left(\left\|\widehat{f} - \widehat{f'}\right\|_{\mathcal{Z}'} - 2\sqrt{|\mathcal{Z}'|} \cdot \varepsilon/72 \cdot \sqrt{\lambda\delta/(|\mathcal{Z}|)}\right)^2$$

$$\geq \left((1 - \varepsilon/6) \left\|\widehat{f} - \widehat{f'}\right\|_{\mathcal{Z}} - 2\sqrt{|\mathcal{Z}'|} \cdot \varepsilon/72 \cdot \sqrt{\lambda\delta/(|\mathcal{Z}|)}\right)^2$$

$$\geq \left((1 - \varepsilon/6)\|f - f\|_{\mathcal{Z}} - 2\sqrt{|\mathcal{Z}'|} \cdot \varepsilon/72 \cdot \sqrt{\lambda\delta/(|\mathcal{Z}|)} - 2\sqrt{|\mathcal{Z}|} \cdot \varepsilon/72 \cdot \sqrt{\lambda\delta/(|\mathcal{Z}|)}\right)^2$$

$$\geq (1 - \varepsilon)\|f - f\|_{\mathcal{Z}}^2. \tag{175}$$

$\square$

Combining Lemma 14, Lemma 15, and Lemma 16 with a union bound, we have the following proposition.

**Proposition 4.** *With probability at least $1 - \delta$, the size of $\mathcal{Z}'$ returned by Algorithm 2 satisfies $|\mathcal{Z}'| \le 4|\mathcal{Z}|/\delta$, the number of distinct elements in $\mathcal{Z}$ is at most*

$$1728 \dim_E(\mathcal{F}, \lambda/|\mathcal{Z}|) \log\left((H+1)^2|\mathcal{Z}|/\lambda\right) \ln(|\mathcal{Z}|) \ln(4\mathcal{N}(\mathcal{F}, \varepsilon/72 \cdot \sqrt{\lambda\delta/(|\mathcal{Z}|)})/\delta)/\varepsilon^2. \quad (176)$$

*and for any $f, f' \in \mathcal{F}$,*

$$(1-\varepsilon)\|f - f'\|_{\mathcal{Z}}^2 - 2\lambda \le \|f - f'\|_{\mathcal{Z}'}^2 \le (1+\varepsilon)\|f - f'\|_{\mathcal{Z}}^2 + 8|\mathcal{Z}|\lambda/\delta \quad (177)$$

**Proposition 5.** *For Algorithm 3, suppose $|\mathcal{Z}| \le KH = T$, the following holds.*

*1. With probability at least $1 - \delta/(16T)$,*

$$w(\underline{\mathcal{F}}, s, a) \le \widehat{w}(s, a) \le w(\overline{\mathcal{F}}, s, a), \quad (178)$$

*where $\underline{\mathcal{F}} = \left\{ f \in \mathcal{F} \mid \|f - \bar{f}\|_{\mathcal{Z}}^2 \le \beta(\mathcal{F}, \delta) \right\}$, and $\overline{\mathcal{F}} = \left\{ f \in \mathcal{F} \mid \|f - \bar{f}\|_{\mathcal{Z}}^2 \le 9\beta(\mathcal{F}, \delta) + 12 \right\}$.*

*2. $\widehat{w}(\cdot, \cdot) \in \mathcal{W}$ for a function set $\mathcal{W}$ with*

$$\log|\mathcal{W}| \le 6912 \dim_E\left(\mathcal{F}, \delta/\left(16T^2\right)\right) \log\left(16(H+1)^2T^2/\delta\right) \ln T \ln(4\mathcal{N}(\mathcal{F}, \delta/(566T))/\delta)$$

$$\cdot \log(\mathcal{N}(\mathcal{S} \times \mathcal{A}, 1/(8\sqrt{4T/\delta})) \cdot 4T/\delta) + \log(\mathcal{N}(\mathcal{F}, 1/(8\sqrt{4T/\delta})))$$

$$\le C \cdot \dim_E\left(\mathcal{F}, \delta/T^3\right) \cdot \log\left(H^2T^2/\delta\right) \cdot \ln T \cdot \ln\left(\mathcal{N}\left(\mathcal{F}, \delta/T^2\right)/\delta\right)$$

$$\cdot \log(\mathcal{N}(\mathcal{S} \times \mathcal{A}, \delta/T)) \cdot T/\delta), \quad (179)$$

*for some absolute constant $C > 0$ if $T$ is sufficiently large.*

*Proof.* For the first part, conditioned on the event defined in Proposition 4, for any $f \in \mathcal{F}$, we have

$$\|f - \bar{f}\|_{\widehat{\mathcal{Z}}}^2/2 - 1/2 \le \|f - \bar{f}\|_{\mathcal{Z}}^2 \le 3\|f - \bar{f}\|_{\widehat{\mathcal{Z}}}^2/2 + 1/2. \quad (180)$$

Therefore, we have

$$\|f - \widehat{f}\|_{\widehat{\mathcal{Z}}}^2 \le \left(\|f - \widehat{f}\|_{\overline{\mathcal{Z}}} + \sqrt{4T/\delta}/(8\sqrt{4T/\delta})\right)^2$$

$$\le \left(\|f - \bar{f}\|_{\overline{\mathcal{Z}}} + \sqrt{4T/\delta}/(8\sqrt{4T/\delta}) + \sqrt{4T/\delta}/(8\sqrt{4T/\delta})\right)^2$$

$$\le 2\|f - \bar{f}\|_{\frac{\mathcal{Z}}{2}}^2 + 2(\sqrt{4T/\delta}/(8\sqrt{4T/\delta}) + \sqrt{4T/\delta}/(8\sqrt{4T/\delta}))^2 \le 3\|f - \bar{f}\|_{\mathcal{Z}}^2 + 2, \quad (181)$$

and

$$\|f - \widehat{f}\|_{\widehat{\mathcal{Z}}}^2 \ge \left(\|f - \widehat{f}\|_{\overline{\mathcal{Z}}} - \sqrt{4T/\delta}/(8\sqrt{4T/\delta})\right)^2$$

$$\ge \left(\|f - \bar{f}\|_{\overline{\mathcal{Z}}} - \sqrt{4T/\delta}/(8\sqrt{4T/\delta}) - \sqrt{4T/\delta}/(8\sqrt{4T/\delta})\right)^2$$

$$\ge \|f - \bar{f}\|_{\frac{\mathcal{Z}}{2}}/2 - (\sqrt{4T/\delta}/(8\sqrt{4T/\delta}) + \sqrt{4T/\delta}/(8\sqrt{4T/\delta}))^2 \ge \|f - \bar{f}\|_{\mathcal{Z}}^2/3 - 2. \quad (182)$$

Therefore, for any $f \in \underline{\mathcal{F}}$, we have $\|f - \bar{f}\|_{\mathcal{Z}}^2 \le \beta(\mathcal{F}, \delta)$, which implies $\|f - \widehat{f}\|_{\widehat{\mathcal{Z}}}^2 \le 3\beta(\mathcal{F}, \delta) + 2$ and thus $f \in \widehat{\mathcal{F}}$. Moreover, for any $f \in \widehat{\mathcal{F}}$, we have $\|f - \widehat{f}\|_{\widehat{\mathcal{Z}}}^2 \le 3\beta(\mathcal{F}, \delta) + 2$, which implies $\|f - \bar{f}\|_{\mathcal{Z}}^2 \le 9\beta(\mathcal{F}, \delta) + 12$.

For the second part, note that $\widehat{w}(\cdot, \cdot)$ is uniquely defined by $\widehat{\mathcal{F}}$. When $|\overline{\mathcal{Z}}| \ge 4T/\delta$ or the number of distinct elements in $\overline{\mathcal{Z}}$ exceeds

$$6912 \dim_E\left(\mathcal{F}, \delta/\left(16T^2\right)\right) \log\left(16(H+1)^2T^2/\delta\right) \ln T \ln(4\mathcal{N}(\mathcal{F}, \delta/(566T))/\delta), \quad (183)$$

we have $|\widehat{\mathcal{Z}}| = 0$ and thus $\widehat{\mathcal{F}} = \mathcal{F}$. Otherwise, $\widehat{\mathcal{F}}$ is defined by $\widehat{f}$ and $\widehat{\mathcal{Z}}$. Since $\widehat{f} \in \mathcal{C}(\mathcal{F}, 1/(8\sqrt{4T/\delta}))$, the total number of distinct $\widehat{f}$ is upper bounded by $\mathcal{N}(\mathcal{F}, 1/(8\sqrt{4T/\delta}))$. Since there are at most

$$6912 \dim_E\left(\mathcal{F}, \delta/\left(16T^2\right)\right) \log\left(16(H+1)^2T^2/\delta\right) \ln T \ln(4\mathcal{N}(\mathcal{F}, \delta/(566T))/\delta) \quad (184)$$

distinct elements in $\widehat{\mathcal{Z}}$, while each of them belongs to $\mathcal{C}(\mathcal{S} \times \mathcal{A}, 1/(8\sqrt{4T/\delta}))$ and $|\widehat{\mathcal{Z}}| \le 4T/\delta$, the total number of distinct $\widehat{\mathcal{Z}}$ is upper bounded by

$$(\mathcal{N}(\mathcal{S} \times \mathcal{A}, 1/(8\sqrt{4T/\delta})) \cdot 4T/\delta)^{6912 \dim_E\left(\mathcal{F}, \delta/\left(16T^2\right)\right) \log\left(16(H+1)^2T^2/\delta\right) \ln T \ln(4\mathcal{N}(\mathcal{F}, \delta/(566T))/\delta)}. \quad (185)$$

$\square$

D.4 ANALYSIS OF THE ALGORITHM

We are now ready to prove the regret bound of Algorithm 1. The next lemma establishes a bound on the estimate of a single backup.

**Lemma 17.** *(Single Step Optimization Error). Consider a fixed $k \in [K]$. Let*

$$\mathcal{Z}_k = \{(s_{t,h'}, a_{t,h'})\}_{(t,h') \in [k-1] \times [H]}, \tag{186}$$

*as defined in Line 5 in Algorithm 1. For any $V : \mathcal{S} \to [0, H]$, define*

$$\mathcal{D}_k^V := \{(s_{t,h'}, a_{t,h'}, r_{t,h'} + V(s_{t,h'+1}))\}_{(t,h') \in [k-1] \times [H]}, \tag{187}$$

*and*

$$\widehat{f}^V := \arg\min_{f \in \mathcal{F}} \|f\|^2_{\mathcal{D}_k^V}. \tag{188}$$

*For any $V : \mathcal{S} \to [0, H]$ and $\delta \in (0, 1)$, there is an event $\mathcal{E}^{V,\delta}$ which holds with probability at least $1 - \delta$, such that conditioned on $\mathcal{E}^{V,\delta}$, for any $V' : \mathcal{S} \to [0, H]$ with $\|V' - V\|_\infty \leq 1/T$, we have*

$$\left\| \widehat{f}^{V'}(\cdot, \cdot) - r(\cdot, \cdot) - \sum_{s' \in \mathcal{S}} P(s' \mid \cdot, \cdot) V'(s') \right\|_{\mathcal{Z}_k} \leq c' \cdot (H\sqrt{\log(2/\delta) + \log \mathcal{N}(\mathcal{F}, 1/T)}), \tag{189}$$

*for some absolute constant $c' > 0$.*

*Proof.* In our proof, we consider a fixed $V : \mathcal{S} \to [0, H]$, and define

$$f^V(\cdot, \cdot) := r(\cdot, \cdot) + \sum_{s' \in \mathcal{S}} P(s' \mid \cdot, \cdot) V(s'). \tag{190}$$

For any $f \in \mathcal{F}$, we consider $\sum_{(t,h) \in [k-1] \times [H]} \xi_{t,h}(f)$ where

$$\xi_{t,h}(f) := 2\left(f(s_{t,h}, a_{t,h}) - f^V(s_{t,h}, a_{t,h})\right) \cdot \left(f^V(s_{t,h}, a_{t,h}) - r_{t,h} - V(s_{h+1}^\tau)\right). \tag{191}$$

For any $(t, h) \in [k-1] \times [H]$, define $\mathbb{F}_{t,h}$ as the filtration induced by the sequence

$$\{(s_{t,h'}, a_{t,h'})\}_{(t,h') \in [\tau-1] \times [H]} \cup \{(s_1^\tau, a_1^\tau), (s_2^\tau, a_2^\tau), \ldots, (s_{h-1}^\tau, a_{h-1}^\tau)\}. \tag{192}$$

Then $\mathbb{E}[\xi_{t,h}(f) \mid \mathbb{F}_{t,h}] = 0$ and

$$|\xi_{t,h}(f)| \leq 2(H+1)\left|f(s_{t,h}, a_{t,h}) - f^V(s_{t,h}, a_{t,h})\right|. \tag{193}$$

By Azuma-Hoeffding inequality, we have

$$\Pr\left[\left|\sum_{(t,h) \in [k-1] \times [H]} \xi_{t,h}(f)\right| \geq \varepsilon\right] \leq 2\exp\left(-\frac{\varepsilon^2}{8(H+1)^2 \|f - f^V\|^2_{\mathcal{Z}_k}}\right). \tag{194}$$

Let

$$\varepsilon = \left(8(H+1)^2 \log\left(\frac{2\mathcal{N}(\mathcal{F}, 1/T)}{\delta}\right) \cdot \|f - f^V\|^2_{\mathcal{Z}_k}\right)^{1/2}$$

$$\leq 4(H+1)\|f - f^V\|_{\mathcal{Z}_k} \cdot \sqrt{\log(2/\delta) + \log \mathcal{N}(\mathcal{F}, 1/T)} \tag{195}$$

We have, with probability at least $1 - \delta$, for all $f \in \mathcal{C}(\mathcal{F}, 1/T)$,

$$\left|\sum_{(t,h) \in [k-1] \times [H]} \xi_{t,h}(f)\right| \leq 4(H+1)\|f - f^V\|_{\mathcal{Z}_k} \cdot \sqrt{\log(2/\delta) + \log \mathcal{N}(\mathcal{F}, 1/T)}. \tag{196}$$

We define the above event to be $\mathcal{E}^{V,\delta}$, and we condition on this event for the rest of the proof. For all $f \in \mathcal{F}$, there exists $g \in \mathcal{C}(\mathcal{F}, 1/T)$, such that $\|f - g\|_\infty \le 1/T$, and we have

$$
\sum_{(t,h) \in [k-1] \times [H]} \xi_{t,h}(f) \le \left| \sum_{(t,h) \in [k-1] \times [H]} \xi_{t,h}(g) \right| + 2(H+1)
$$

$$
\le 4(H+1) \left\| g - f^V \right\|_{\mathcal{Z}_k} \cdot \sqrt{\log(2/\delta) + \log \mathcal{N}(\mathcal{F}, 1/T)} + 2(H+1)
$$

$$
\le 4(H+1) \left( \left\| f - f^V \right\|_{\mathcal{Z}_k} + 1 \right) \cdot \sqrt{\log(2/\delta) + \log \mathcal{N}(\mathcal{F}, 1/T)} + 2(H+1). \tag{197}
$$

Consider $V' : \mathcal{S} \to [0, H]$ with $\|V' - V\|_\infty \le 1/T$. We have

$$
\left\| f^{V'} - f^V \right\|_\infty \le \|V' - V\|_\infty \le 1/T. \tag{198}
$$

For any $f \in \mathcal{F}$,

$$
\|f\|_{\mathcal{D}_k^{V'}}^2 - \left\| f^{V'} \right\|_{\mathcal{D}_k^{V'}}^2 = \left\| f - f^{V'} \right\|_{\mathcal{Z}_k}^2
$$

$$
+ 2 \sum_{\left( s_{t,h'}, a_{t,h'} \right) \in \mathcal{Z}_k} \left( f\left( s_{t,h'}, a_{t,h'} \right) - f^{V'}\left( s_{t,h'}, a_{t,h'} \right) \right) \cdot \left( f^{V'}\left( s_{t,h'}, a_{t,h'} \right) - r_{t,h'} - V'\left( s_{t,h'+1} \right) \right). \tag{199}
$$

For the second term, we have,

$$
2 \sum_{\left( s_{t,h'}, a_{t,h'} \right) \in \mathcal{Z}_k} \left( f\left( s_{t,h'}, a_{t,h'} \right) - f^{V'}\left( s_{t,h'}, a_{t,h'} \right) \right) \cdot \left( f^{V'}\left( s_{t,h'}, a_{t,h'} \right) - r_{t,h'} - V'\left( s_{t,h'+1} \right) \right)
$$

$$
\ge 2 \sum_{\left( s_{t,h'}, a_{t,h'} \right) \in \mathcal{Z}_k} \left( f\left( s_{t,h'}, a_{t,h'} \right) - f^V\left( s_{t,h'}, a_{t,h'} \right) \right) \cdot \left( f^V\left( s_{t,h'}, a_{t,h'} \right) - r_{t,h'} - V\left( s_{t,h'+1} \right) \right)
$$

$$
- 4(H+1) \cdot \|V' - V\|_\infty \cdot |\mathcal{Z}_k|
$$

$$
= \sum_{(t,h) \in [k-1] \times [H]} \xi_{t,h}(f) - 4(H+1) \cdot \|V' - V\|_\infty \cdot |\mathcal{Z}_k|
$$

$$
\ge -4(H+1) \left( \left\| f - f^V \right\|_{\mathcal{Z}_k} + 1 \right) \cdot \sqrt{\log(2/\delta) + \log \mathcal{N}(\mathcal{F}, 1/T)}
$$

$$
- 2(H+1) - 4(H+1) \cdot \|V' - V\|_\infty \cdot |\mathcal{Z}_k|
$$

$$
\ge -4(H+1) \left( \left\| f - f^{V'} \right\|_{\mathcal{Z}_k} + 2 \right) \cdot \sqrt{\log(2/\delta) + \log \mathcal{N}(\mathcal{F}, 1/T)} - 6(H+1). \tag{200}
$$

Recall that $\widehat{f}^{V'} = \arg\min_{f \in \mathcal{F}} \|f\|_{\mathcal{D}_k^{V'}}^2$. We have $\left\| \widehat{f}^{V'} \right\|_{\mathcal{D}_k^{V'}}^2 - \left\| f^{V'} \right\|_{\mathcal{D}_k^{V'}}^2 \le 0$, which implies,

$$
0 \ge \left\| \widehat{f}^{V'} \right\|_{\mathcal{D}_k^{V'}}^2 - \left\| f^{V'} \right\|_{\mathcal{D}_k^{V'}}^2
$$

$$
= \left\| \widehat{f}^{V'} - f^{V'} \right\|_{\mathcal{Z}_k}^2
$$

$$
+ 2 \sum_{\left( s_{h'}^\tau, a_{h'}^\tau \right) \in \mathcal{Z}_k} \left( \widehat{f}\left( s_{h'}^\tau, a_{h'}^\tau \right) - f^{V'}\left( s_{h'}^\tau, a_{h'}^\tau \right) \right) \cdot \left( f^{V'}\left( s_{h'}^\tau, a_{h'}^\tau \right) - r_{h'}^\tau - V'\left( s_{h'+1}^\tau \right) \right)
$$

$$
\ge \left\| \widehat{f}^{V'} - f^{V'} \right\|_{\mathcal{Z}_k}^2
$$

$$
- 4(H+1) \left( \left\| \widehat{f}^{V'} - f^{V'} \right\|_{\mathcal{Z}_k} + 2 \right) \cdot \sqrt{\log(2/\delta) + \log \mathcal{N}(\mathcal{F}, 1/T)} - 6(H+1). \tag{201}
$$

Solving the above inequality, we have,

$$
\left\| \widehat{f}^{V'} - f^{V'} \right\|_{\mathcal{Z}_k} \le c' \cdot \left( H \cdot \sqrt{\log \delta^{-1} + \log \mathcal{N}(\mathcal{F}, 1/T)} \right), \tag{202}
$$

for an absolute constant $c' > 0$.

$\square$

**Lemma 18.** *(Confidence Region). In Algorithm 1, let $\mathcal{F}_{k,h}$ be a confidence region defined as*

$$\mathcal{F}_{k,h} = \left\{ f \in \mathcal{F} \mid \|f - f_{k,h}\|_{\mathcal{Z}_k}^2 \leq \beta(\mathcal{F}, \delta) \right\}. \tag{203}$$

*Then with probability at least $1 - \delta/8$, for all $k, h \in [K] \times [H]$,*

$$r(\cdot, \cdot) + \sum_{s' \in \mathcal{S}} P\left(s' \mid \cdot, \cdot\right) V_{k,h+1}\left(s'\right) \in \mathcal{F}_{k,h}, \tag{204}$$

*provided*

$$\beta(\mathcal{F}, \delta) \geq c' \cdot (H\sqrt{\log(T/\delta) + \log(|\mathcal{W}|) + \log \mathcal{N}(\mathcal{F}, 1/T)})^2, \tag{205}$$

*for some absolute constant $c' > 0$. Here $\mathcal{W}$ is given as in Proposition 5.*

*Proof.* For all $(k, h) \in [K] \times [H]$, the bonus function $b_{k,h}(\cdot, \cdot) \in \mathcal{W}$. Note that

$$\mathcal{Q} := \{\min\{f(\cdot, \cdot) + w(\cdot, \cdot), H\} \mid w \in \mathcal{W}, f \in \mathcal{C}(\mathcal{F}, 1/T)\} \cup \{0\} \tag{206}$$

is a $(1/T)$-cover of

$$Q_{k,h+1}(\cdot, \cdot) = \begin{cases} \min\{f_{k,h+1}(\cdot, \cdot) + b_{k,h+1}(\cdot, \cdot), H\} & h < H \\ 0 & h = H \end{cases}. \tag{207}$$

I.e., there exists $q \in \mathcal{Q}$ such that $\|q - Q_{k,h+1}\|_\infty \leq 1/T$. This implies

$$\mathcal{V} := \left\{ \max_{a \in \mathcal{A}} q(\cdot, a) \mid q \in \mathcal{Q} \right\} \tag{208}$$

is a $(1/T)$-cover of $V_{k,h+1}$ with $\log(|\mathcal{V}|) \leq \log |\mathcal{W}| + \log \mathcal{N}(\mathcal{F}, 1/T) + 1$. For each $V \in \mathcal{V}$, let $\mathcal{E}^{V,\delta/(8|\mathcal{V}|T)}$ be the event defined in Lemma 17. By Lemma 17, we have $\Pr\left[\bigcap_{V \in \mathcal{V}} \mathcal{E}^{V,\delta/(8|\mathcal{V}|T)}\right] \geq 1 - \delta/(8T)$. We condition on $\bigcap_{V \in \mathcal{V}} \mathcal{E}^{V,\delta/(8|\mathcal{V}|T)}$ in the rest part of the proof.

Recall that $f_{k,h}$ is the solution of the optimization problem in Line 8 of Algorithm 1, i.e., $f_{k,h} = \arg\min_{f \in \mathcal{F}} \|f\|_{\mathcal{D}_{k,h}}^2$. Let $V \in \mathcal{V}$ such that $\|V - V_{k,h+1}\|_\infty \leq 1/T$. Thus, by Lemma 5, we have

$$\left\| f_{k,h}(\cdot, \cdot) - \left( r(\cdot, \cdot) + \sum_{s' \in \mathcal{S}} P\left(s' \mid \cdot, \cdot\right) V_{k,h+1}\left(s'\right) \right) \right\|_{\mathcal{Z}_k}$$
$$\leq c' \cdot (H\sqrt{\log(T/\delta) + \log \mathcal{N}(\mathcal{F}, 1/T) + \log |\mathcal{W}|}) \tag{209}$$

for some absolute constant $c'$. Therefore, by a union bound, for all $(k, h) \in [K] \times [H]$, we have $f_{k,h}(\cdot, \cdot) - \left( r(\cdot, \cdot) + \sum_{s' \in \mathcal{S}} P\left(s' \mid \cdot, \cdot\right) V_{k,h+1}\left(s'\right) \right) \in \mathcal{F}_{k,h}$ with probability at least $1 - \delta/8$.

$\square$

The above lemma guarantees that, with high probability, $r(\cdot, \cdot) + \sum_{s' \in \mathcal{S}} P\left(s' \mid \cdot, \cdot\right) V_{k,h+1}(\cdot, \cdot)$ lies in the confidence region. With this, it is guaranteed that $\{Q_{k,h}\}_{(h,k) \in [H] \times [K]}$ are all optimistic, with high probability. This is formally presented in the next lemma.

**Lemma 19.** *With probability at least $1 - \delta/4$, for all $(k, h) \in [K] \times [H]$, for all $(s, a) \in \mathcal{S} \times \mathcal{A}$,*

$$Q_h^*(s, a) \leq Q_{k,h}(s, a) \leq r(s, a) + \sum_{s' \in \mathcal{S}} P\left(s' \mid s, a\right) V_{k,h+1}\left(s'\right) + 2b_{k,h}(s, a). \tag{210}$$

*Proof.* For each $(k, h) \in [K] \times [H]$, define

$$\mathcal{F}_{k,h} = \left\{ f \in \mathcal{F} \mid \|f - f_{k,h}\|_{\mathcal{Z}_k}^2 \leq \beta(\mathcal{F}, \delta) \right\}. \tag{211}$$

Let $\mathcal{E}$ be the event that for all $(k,h) \in [K] \times [H], r(\cdot,\cdot) + \sum_{s' \in \mathcal{S}} P(s' \mid \cdot,\cdot) V_{k,h+1}(s') \in \mathcal{F}_{k,h}$. By Lemma 18, $\Pr[\mathcal{E}] \geq 1 - \delta/8$. Let $\mathcal{E}'$ be the event that for all $(k,h) \in [K] \times [H]$ and $(s,a) \in \mathcal{S} \times \mathcal{A}$, $b_{k,h}(s,a) \geq w(\mathcal{F}_{k,h}, s, a)$. By Proposition 5 and union bound, $\mathcal{E}'$ holds failure probability at most $\delta/8$. In the rest part of the proof we condition on $\mathcal{E}$ and $\mathcal{E}'$.

Note that

$$\max_{f \in \mathcal{F}_{k,h}} |f(s,a) - f_{k,h}(s,a)| \leq w(\mathcal{F}_{k,h}, s, a) \leq b_{k,h}(s,a). \tag{212}$$

Since

$$r(\cdot,\cdot) + \sum_{s' \in \mathcal{S}} P(s' \mid \cdot,\cdot) V_{k,h+1}(s') \in \mathcal{F}_{k,h}, \tag{213}$$

for any $(s,a) \in \mathcal{S} \times \mathcal{A}$, we have

$$\left| r(s,a) + \sum_{s' \in \mathcal{S}} P(s' \mid s,a) V_{k,h+1}(s') - f_{k,h}(s,a) \right| \leq b_{k,h}(s,a). \tag{214}$$

Hence,

$$Q_{k,h}(s,a) \leq f_{k,h}(s,a) + b_{k,h}(s,a) \leq r(s,a) + \sum_{s' \in \mathcal{S}} P(s' \mid s,a) V_{k,h+1}(s') + 2b_{k,h}(s,a). \tag{215}$$

Now we prove $Q_h^*(s,a) \leq Q_{k,h}(s,a)$ by induction on $h$. When $h = H+1$, the desired inequality clearly holds. Now we assume $Q_{h+1}^*(\cdot,\cdot) \leq Q_{k,h+1}(\cdot,\cdot)$ for some $h \in [H]$. Clearly we have $V_{h+1}^*(\cdot) \leq V_{k,h+1}(\cdot)$. Therefore, for all $(s,a) \in \mathcal{S} \times \mathcal{A}$

$$Q_h^*(s,a) = r(s,a) + \sum_{s' \in \mathcal{S}} P(s' \mid s,a) V_{h+1}^*(s')$$

$$\leq \min \left\{ H, r(s,a) + \sum_{s' \in \mathcal{S}} P(s' \mid s,a) V_{k,h+1}(s') \right\}$$

$$\leq \min \{H, f_{k,h}(s,a) + b_{k,h}(s,a)\}$$
$$= Q_{k,h}(s,a). \tag{216}$$

$\square$

The next lemma upper bounds the regret of the algorithm by the sum of $b_{k,h}(\cdot,\cdot)$.

**Lemma 20.** *With probability at least* $1 - \delta/2$,

$$\mathrm{Reg}(K) \leq 2 \sum_{k=1}^{K} \sum_{h=1}^{H} b_{k,h}(s_{k,h}, a_{k,h}) + 4H\sqrt{KH \cdot \log(8/\delta)}. \tag{217}$$

*Proof.* In our proof, for any $(k,h) \in [K] \times [H-1]$ define

$$\xi_{k,h} = \sum_{s' \in \mathcal{S}} P(s' \mid s_{k,h}, a_{k,h}) \left( V_{k,h+1}(s') - V_{h+1}^{\pi_k}(s') \right) - \left( V_{k,h+1}(s_{k,h+1}) - V_{h+1}^{\pi_k}(s_{k,h+1}) \right), \tag{218}$$

and define $\mathbb{F}_{k,h}$ as the filtration induced by the sequence

$$\{(s_{h'}^\tau, a_{h'}^\tau)\}_{(\tau,h') \in [k-1] \times [H]} \cup \{(s_{k,1}, a_{k,1}), (s_{k,2}, a_{k,2}), \ldots, (s_{k,h}, a_{k,h})\}. \tag{219}$$

Then

$$\mathbb{E}[\xi_{k,h} \mid \mathbb{F}_{k,h}] = 0 \text{ and } |\xi_{k,h}| \leq 2H. \tag{220}$$

By Azuma-Hoeffding inequality, with probability at least $1 - \delta/4$,

$$\sum_{k=1}^{K} \sum_{h=1}^{H-1} \xi_{k,h} \leq 4H \sqrt{KH \cdot \log(8/\delta)}. \tag{221}$$

We condition on the above event in the rest of the proof. We also condition on the event defined in Lemma 19 which holds with probability $1 - \delta/4$.

Recall that

$$\text{Reg}(K) = \sum_{k=1}^{K} \left( V_1^* \left( s_{k,1} \right) - V_1^{\pi_k} \left( s_{k,1} \right) \right) \leq \sum_{k=1}^{K} V_{k,1} \left( s_{k,1} \right) - V_1^{\pi_k} \left( s_{k,1} \right). \tag{222}$$

We have

$$\text{Reg}(K)$$

$$\leq \sum_{k=1}^{K} \left( r\left( s_{k,1}, a_{k,1} \right) + \sum_{s' \in \mathcal{S}} P\left( s' \mid s_{k,1}, a_{k,1} \right) V_{k,2} \left( s' \right) + 2b_{k,1} \left( s_{k,1}, a_{k,1} \right) \right.$$

$$\left. - r\left( s_{k,1}, a_{k,1} \right) - \sum_{s' \in \mathcal{S}} P\left( s' \mid s_{k,1}, a_{k,1} \right) V_2^{\pi_k} \left( s' \right) \right)$$

$$= \sum_{k=1}^{K} \sum_{s' \in \mathcal{S}} P\left( s' \mid s_{k,1}, a_{k,1} \right) \left( V_{k,2} \left( s' \right) - V_2^{\pi_k} \left( s' \right) \right) + 2b_{k,1} \left( s_{k,1}, a_{k,1} \right)$$

$$= \sum_{k=1}^{K} V_{k,2} \left( s_{k,2} \right) - V_2^{\pi_k} \left( s_{k,2} \right) + \xi_{k,1} + 2b_{k,1} \left( s_{k,1}, a_{k,1} \right)$$

$$\leq \sum_{k=1}^{K} V_3^k \left( s_3^k \right) - V_3^{\pi_k} \left( s_3^k \right) + \xi_{k,1} + \xi_{k,2} + 2b_{k,1} \left( s_{k,1}, a_{k,1} \right) + 2b_{k,2} \left( s_{k,2}, a_{k,2} \right)$$

$$\leq \sum_{k=1}^{K} \sum_{h=1}^{H-1} \xi_{k,h} + \sum_{k=1}^{K} \sum_{h=1}^{H} 2b_{k,h} \left( s_{k,h}, a_{k,h} \right). \tag{223}$$

Therefore,

$$\text{Reg}(K) \leq 2 \sum_{k=1}^{K} \sum_{h=1}^{H} b_{k,h} \left( s_{k,h}, a_{k,h} \right) + 4H \sqrt{KH \cdot \log(8/\delta)}. \tag{224}$$

$\square$

It remains to bound $\sum_{k=1}^{K} \sum_{h=1}^{H} b_{k,h} \left( s_{k,h}, a_{k,h} \right)$, for which we will exploit fact that $\mathcal{F}$ has bounded eluder dimension.

**Lemma 21.** *With probability at least $1 - \delta/4$, for any $\varepsilon > 0$,*

$$\sum_{k=1}^{K} \sum_{h=1}^{H} \mathbb{I} \left( b_{k,h} \left( s_{k,h}, a_{k,h} \right) > \varepsilon \right) \leq \left( \frac{c\beta(\mathcal{F}, \delta)}{\varepsilon^2} + H \right) \cdot \dim_E(\mathcal{F}, \varepsilon), \tag{225}$$

*for some absolute constant $c > 0$. Here $\beta(\mathcal{F}, \delta)$ is as defined in (4).*

*Proof.* Let $\mathcal{E}$ be the event that or all $(k, h) \in [K] \times [H]$,

$$b_{k,h}(\cdot, \cdot) \leq w\left( \overline{\mathcal{F}}_{k,h}, \cdot, \cdot \right), \tag{226}$$

where

$$\overline{\mathcal{F}}_{k,h} = \left\{ f \in \mathcal{F} : \| f - f_{k,h} \|_{\mathcal{Z}_k}^2 \leq 9\beta + 12 \right\}. \tag{227}$$

By Proposition 5, $\mathcal{E}$ holds with probability at least $1 - \delta/4$. In the rest of the proof, we condition on $\mathcal{E}$.

Let $\mathcal{L} = \{(s_{k,h}, a_{k,h}) \mid b_{k,h}(s_{k,h}, a_{k,h}) > \varepsilon\}$ with $|\mathcal{L}| = L$. We show that there exists $(s_{k,h}, a_{k,h}) \in \mathcal{L}$ such that $(s_{k,h}, a_{k,h})$ is $\varepsilon$-dependent on at least $L/\dim_E(\mathcal{F}, \varepsilon) - H$ disjoint subsequences in $\mathcal{Z}_k \cap \mathcal{L}$. We demonstrate this by using the following procedure. Let $\mathcal{L}_1, \mathcal{L}_2, \dots, \mathcal{L}_{L/\dim_E(\mathcal{F}, \varepsilon) - 1}$ be $L/\dim_E(\mathcal{F}, \varepsilon) - 1$ disjoint subsequences of $\mathcal{L}$ which are initially empty. We consider

$$\{(s_{k,1}, a_{k,1}), (s_{k,2}, a_{k,2}), \dots, (s_{k,H}, a_{k,H})\} \cap \mathcal{L}, \tag{228}$$

for each $k \in [K]$ sequentially. For each $k \in [K]$, for each $z \in \{(s_{k,1}, a_{k,1}), (s_{k,2}, a_{k,2}), \dots, (s_{k,H}, a_{k,H})\} \cap \mathcal{L}$, we find $j \in [L/\dim_E(\mathcal{F}, \varepsilon) - 1]$ such that $z$ is $\varepsilon$-independent of $\mathcal{L}_j$ and then add $z$ into $\mathcal{L}_j$. By the definition of $\varepsilon$-independence, $|\mathcal{L}_j| \leq \dim_E(\mathcal{F}, \varepsilon)$ for all $j$ and thus we will eventually find some $(s_{k,h}, a_{k,h}) \in \mathcal{L}$ such that $(s_{k,h}, a_{k,h})$ is $\varepsilon$-dependent on each of $\mathcal{L}_1, \mathcal{L}_2, \dots, \mathcal{L}_{L/\dim_E(\mathcal{F}, \varepsilon) - 1}$. Among $\mathcal{L}_1, \mathcal{L}_2, \dots, \mathcal{L}_{L/\dim_E(\mathcal{F}, \varepsilon) - 1}$, there are at most $H - 1$ of them that contain an element in

$$\{(s_{k,1}, a_{k,1}), (s_{k,2}, a_{k,2}), \dots, (s_{k,H}, a_{k,H})\} \cap \mathcal{L}, \tag{229}$$

and all other subsequences only contain elements in $\mathcal{Z}_k \cap \mathcal{L}$. Therefore, $(s_{k,h}, a_{k,h})$ is $\varepsilon$-dependent on at least $L/\dim_E(\mathcal{F}, \varepsilon) - H$ disjoint subsequences in $\mathcal{Z}_k \cap \mathcal{L}$.

On the other hand, since $(s_{k,h}, a_{k,h}) \in \mathcal{L}$, we have $b_{k,h}(s_{k,h}, a_{k,h}) > \varepsilon$, which implies there exists $f, f' \in \mathcal{F}$ with $\|f - f_{k,h}\|_{\mathcal{Z}_k}^2 \leq 9\beta + 12$ and $\|f' - f_{k,h}\|_{\mathcal{Z}_k}^2 \leq 9\beta + 12$ such that $f(z) - f'(z) > \varepsilon$. By triangle inequality, we have $\|f - f'\|_{\mathcal{Z}_k}^2 \leq 36\beta + 48$. On the other hand, since $(s_{k,h}, a_{k,h})$ is $\varepsilon$-dependent on at least $L/\dim_E(\mathcal{F}, \varepsilon) - H$ disjoint subsequences in $\mathcal{Z}_k \cap \mathcal{L}$, we have

$$(L/\dim_E(\mathcal{F}, \varepsilon) - H)\varepsilon^2 \leq \|f - f\|_{\mathcal{Z}_k}^2 \leq 36\beta + 48, \tag{230}$$

which implies

$$L \leq \left(\frac{36\beta + 48}{\varepsilon^2} + H\right) \dim_E(\mathcal{F}, \varepsilon). \tag{231}$$

$\square$

Lastly, we apply the above lemma to bound the overall regret.

**Lemma 22.** *With probability at least* $1 - \delta/4$,

$$\sum_{k=1}^{K} \sum_{1}^{H} b_{k,h}(s_{k,h}, a_{k,h}) \leq 1 + 4H^2 \dim_E(\mathcal{F}, 1/T) + \sqrt{c \cdot \dim_E(\mathcal{F}, 1/T) \cdot T \cdot \beta(\mathcal{F}, \delta)}, \tag{232}$$

*for some absolute constant* $c > 0$. *Here* $\beta(\mathcal{F}, \delta)$ *is as defined in (4).*

*Proof.* In the proof we condition on the event defined in Lemma 21. We define $w_{k,h} := b_{k,h}(s_{k,h}, a_{k,h})$. Let $w_1 \geq w_2 \geq \dots \geq w_T$ be a permutation of $\{w_{k,h}\}_{(k,h) \in [K] \times [H]}$. By the event defined in Lemma 21, for any $w_t \geq 1/T$, we have

$$t \leq \left(\frac{c\beta(\mathcal{F}, \delta)}{w_t^2} + H\right) \dim_E(\mathcal{F}, w_t) \leq \left(\frac{c\beta(\mathcal{F}, \delta)}{w_t^2} + H\right) \dim_E(\mathcal{F}, 1/T), \tag{233}$$

which implies

$$w_t \leq \left(\frac{t}{\dim_E(\mathcal{F}, 1/T)} - H\right)^{-1/2} \cdot \sqrt{c\beta(\mathcal{F}, \delta)}. \tag{234}$$

Moreover, we have $w_t \leq 4H$. Therefore,

$$\sum_{t=1}^{T} w_t \leq 1 + 4H^2 \dim_E(\mathcal{F}, 1/T) + \sum_{H \dim_E(\mathcal{F}, 1/T) < t \leq T} \left(\frac{t}{\dim_E(\mathcal{F}, 1/T)} - H\right)^{-1/2} \cdot \sqrt{c\beta(\mathcal{F}, \delta)}$$

$$\leq 1 + 4H^2 \dim_E(\mathcal{F}, 1/T) + 2\sqrt{c \cdot \dim_E(\mathcal{F}, 1/T) \cdot T \cdot \beta(\mathcal{F}, \delta)}. \tag{235}$$

$\square$

We are now ready to prove our main theorem.

Proof of Theorem 1. By Lemma 20 and Lemma 22, with probability at least $1 - \delta$,

$\text{Reg}(K)$

$$\leq \min \left\{ KH, \sum_{k=1}^{K} \sum_{h=1}^{H} 2b_{k,h} \left( s_{k,h}, a_{k,h} \right) + 4H\sqrt{KH \cdot \log(8/\delta)} \right\} \tag{236}$$

$$\leq c \cdot \min \left\{ KH, \left( \dim_E(\mathcal{F}, 1/T) \cdot H^2 + \sqrt{\dim_E(\mathcal{F}, 1/T) \cdot T \cdot \beta(\mathcal{F}, \delta)} + H\sqrt{KH \cdot \log \delta^{-1}} \right) \right\}, \tag{237}$$

for some absolute constants $c > 0$. Substituting the value of $\beta(\mathcal{F}, \delta)$ completes the proof.

## E  IDEA: WEIGHT

In this section, we repeat the key results in He et al. (2022) that are useful for our derivation.

**Lemma 23.** *For any $0 < \delta < 1$ and corruption budget $C \geq 0$, set the confidence radius $\beta = R\sqrt{d \log \left( (1 + KL^2/\lambda)/\delta \right)} + \sqrt{\lambda}S + \alpha C$ in Algorithm 1, then with probability at least $1 - \delta$, for every round $k$, the estimator $\boldsymbol{\theta}_k$ satisfies that $\|\boldsymbol{\theta}_k - \boldsymbol{\theta}^*\|_{\boldsymbol{\Sigma}_k} \leq \beta$.*

**Lemma 24.** *For any $0 < \delta < 1$ and corruption budget $C \geq 0$, set the confidence radius $\beta$ in Algorithm 1 as follows:*

$$\beta = R\sqrt{d \log \left( (1 + KL^2/\lambda)/\delta \right)} + \alpha C + \sqrt{\lambda}S. \tag{238}$$

*Then with probability at least $1 - \delta$, its regret in the first $K$ rounds is upper bounded by*

$$\text{Regret}(K) = O\left( dR\sqrt{K \log^2 \left( (1 + KL^2/\lambda)/\delta \right)} + \alpha C\sqrt{dK \log^2 \left( (1 + KL^2/\lambda)/\delta \right)} \right. \tag{239}$$

$$+ S\sqrt{d\lambda K \log \left( 1 + KL^2/\lambda \right)} + \frac{Rd^{1.5}}{\alpha} \times \sqrt{\log^3 \left( (1 + KL^2/\lambda)/\delta \right)} \tag{240}$$

$$+ \frac{dS\sqrt{\lambda}}{\alpha} \times \sqrt{\log^2 \left( (1 + KL^2/\lambda)/\delta \right)} + dC\sqrt{\log^2 \left( (1 + KL^2/\lambda)/\delta \right)} \right). \tag{241}$$

*In addition, if choosing $\alpha = (R\sqrt{d} + \sqrt{\lambda}S)/C$ and $\lambda = R^2/S^2$, its regret can be upper bounded by*

$$\text{Regret}(K) = \widetilde{O}(d\sqrt{K} + dC). \tag{242}$$

### E.1  PROOF OF LEMMA 23

*Proof.* According to the definition of estimated vector $\boldsymbol{\theta}_k$ in Algorithm 1 (Line 3), we have

$$\boldsymbol{\theta}_k = \boldsymbol{\Sigma}_k^{-1}\mathbf{b}_k = \boldsymbol{\Sigma}_k^{-1} \sum_{i=1}^{k-1} w_i \mathbf{x}_i r_i = \boldsymbol{\Sigma}_k^{-1} \sum_{i=1}^{k-1} w_i \mathbf{x}_i \left( \mathbf{x}_i^\top \boldsymbol{\theta} + \eta_i + c_i \right). \tag{243}$$

This equation further implies that the difference between estimated vector $\boldsymbol{\theta}_k$ and the unknown vector $\boldsymbol{\theta}^*$ can be decomposed as:

$$\|\boldsymbol{\theta}_k - \boldsymbol{\theta}^*\|_{\boldsymbol{\Sigma}_k} = \left\| \boldsymbol{\Sigma}_k^{-1} \sum_{i=1}^{k-1} w_i \mathbf{x}_i \left( \mathbf{x}_i^\top \boldsymbol{\theta}^* + \eta_i + c_i \right) - \boldsymbol{\theta}^* \right\|_{\boldsymbol{\Sigma}_k}$$

$$= \left\| \boldsymbol{\Sigma}_k^{-1} \sum_{i=1}^{k-1} w_i \mathbf{x}_i \left( \mathbf{x}_i^\top \boldsymbol{\theta}^* + \eta_i + c_i \right) - \boldsymbol{\Sigma}_k^{-1} \left( \sum_{i=1}^{k-1} w_i \mathbf{x}_i \mathbf{x}_i^\top + \lambda\mathbf{I} \right) \boldsymbol{\theta}^* \right\|_{\boldsymbol{\Sigma}_k}$$

$$= \left\| \boldsymbol{\Sigma}_k^{-1} \sum_{i=1}^{k-1} w_i \mathbf{x}_i \eta_i + \boldsymbol{\Sigma}_k^{-1} \sum_{i=1}^{k-1} w_i \mathbf{x}_i c_i - \lambda\boldsymbol{\Sigma}_k^{-1}\boldsymbol{\theta}^* \right\|_{\boldsymbol{\Sigma}_k}$$

$$\leq \underbrace{\left\| \boldsymbol{\Sigma}_k^{-1} \sum_{i=1}^{k-1} w_i \mathbf{x}_i \eta_i \right\|_{\boldsymbol{\Sigma}_k}}_{\text{Stochastic error: } I_1} + \underbrace{\left\| \boldsymbol{\Sigma}_k^{-1} \sum_{i=1}^{k-1} w_i \mathbf{x}_i c_i \right\|_{\boldsymbol{\Sigma}_k}}_{\text{Corruption error: } I_2} + \underbrace{\left\| \lambda\boldsymbol{\Sigma}_k^{-1}\boldsymbol{\theta}^* \right\|_{\boldsymbol{\Sigma}_k}}_{\text{Regularization error: } I_3}, \tag{244}$$

where the inequality holds due to the fact that $\|\mathbf{a} + \mathbf{b} + \boldsymbol{c}\|_{\boldsymbol{\Sigma}_k} \leq \|\mathbf{a}\|_{\boldsymbol{\Sigma}_k} + \|\mathbf{b}\|{\boldsymbol{\Sigma}_k} + \|\boldsymbol{c}\|{\boldsymbol{\Sigma}_k}$.

For the stochastic error term $I_1$, it can be bounded by the concentration Lemma H. 2 in AbbasiYadkori et al. (2011). More specifically, we introduce the auxiliary vector $\mathbf{x}'_i$ and noise $\eta'_i$ such that $\mathbf{x}'_i = \sqrt{w_i}\mathbf{x}_i$ and $\eta'_i = \sqrt{w_i}\eta_i$. According to the definition of weight $\boldsymbol{\theta}_i$, both of these two situations satisfies that the weight $\boldsymbol{\theta}_i$ is bounded by $w_i \leq 1$. Since the original vector $\mathbf{x}_i$ satisfies that $\|\mathbf{x}_i\|_2 \leq L$ and the original stochastic noise $\eta_i$ is $R$-sub Gaussian, these results further imply that

$$\|\mathbf{x}'_i\|_2 = \|\sqrt{w_i}\mathbf{x}_i\|_2 \leq L, \eta'_i = \sqrt{w_i}\eta_i \, is R - subGaussian. \tag{245}$$

With this notation, the covariance matrix $\boldsymbol{\Sigma}_k$ and the stochastic error term $I_1$ can be rewritten and bounded as:

$$\boldsymbol{\Sigma}_k = \lambda\mathbf{I} + \sum_{i=1}^{k-1} w_i\mathbf{x}_i\mathbf{x}_i^\top = \lambda\mathbf{I} + \sum_{i=1}^{k-1} \mathbf{x}'_i (\mathbf{x}'_i)^\top \tag{246}$$

$$I_1 = \left\|\boldsymbol{\Sigma}_k^{-1} \sum_{i=1}^{k-1} w_i\mathbf{x}_i\eta_i\right\|_{\boldsymbol{\Sigma}_k} \tag{247}$$

$$= \left\|\sum_{i=1}^{k-1} w_i\mathbf{x}_i\eta_i\right\|_{\boldsymbol{\Sigma}_k^{-1}} \tag{248}$$

$$= \left\|\sum_{i=1}^{k-1} \mathbf{x}'_i\eta'_i\right\|_{\boldsymbol{\Sigma}_k^{-1}} \tag{249}$$

$$\leq \sqrt{2R^2 \log\left(\frac{\det(\boldsymbol{\Sigma}_k)^{1/2} \det(\boldsymbol{\Sigma}_1)^{-1/2}}{\delta}\right)} \tag{250}$$

$$\leq R\sqrt{d\log\left((1 + KL^2/\lambda)/\delta\right)}, \tag{251}$$

where the first inequality holds due to Lemma H. 2 and the second inequality holds due to the facts that $\boldsymbol{\Sigma}_k = \lambda\mathbf{I} + \sum_{i=1}^{k-1} \mathbf{x}'_i (\mathbf{x}'_i)^\top$ and $\|\mathbf{x}'\|_2 \leq L$.

For the corruption error term $I_2$, it can be bounded by

$$I_2 = \left\|\boldsymbol{\Sigma}_k^{-1} \sum_{i=1}^{k-1} w_i\mathbf{x}_i c_i\right\|_{\boldsymbol{\Sigma}_k}$$

$$= \left\|\boldsymbol{\Sigma}_k^{-1/2} \sum_{i=1}^{k-1} w_i\mathbf{x}_i c_i\right\|_2$$

$$\leq \sum_{i=1}^{k-1} \left\|\boldsymbol{\Sigma}_k^{-1/2} w_i\mathbf{x}_i c_i\right\|_2$$

$$= \sum_{i=1}^{k-1} |c_i| \times w_i \left\|\boldsymbol{\Sigma}_k^{-1/2}\mathbf{x}_i\right\|$$

$$\leq \sum_{i=1}^{k-1} |c_i| \alpha$$

$$\leq \alpha C, \tag{252}$$

where the first inequality holds due to the fact that $\|\mathbf{a} + \mathbf{b}\|_2 \leq \|\mathbf{a}\|_2 + \|\mathbf{b}\|_2$, the second inequality holds due to the definition of weight $w_i$ in Algorithm (Line 6) with the fact that $\boldsymbol{\Sigma}_k \succeq \boldsymbol{\Sigma}_i$ and the last inequality holds due to the definition of corruption level $C$.

For the regularization error term $I_3$, we have

$$I_3 = \left\|\lambda\boldsymbol{\Sigma}_k^{-1}\boldsymbol{\theta}^*\right\|_{\boldsymbol{\Sigma}_k} = \lambda\|\boldsymbol{\theta}^*\|_{\boldsymbol{\Sigma}_k^{-1}} \leq \sqrt{\lambda}\|\boldsymbol{\theta}^*\|_2 \leq \sqrt{\lambda}S, \tag{253}$$

where the first inequality holds due to $\|\boldsymbol{\theta}^*\|_{\boldsymbol{\Sigma}_k} \leq \|\boldsymbol{\theta}^*\|_2 / \sqrt{\lambda_{\min}(\boldsymbol{\Sigma}_k)}$ with the fact that $\boldsymbol{\Sigma}_k = \lambda \mathbf{I} + \sum_{i=1}^{k-1} w_i \mathbf{x}_i \mathbf{x}_i^\top \succeq \lambda \mathbf{I}$ and the last inequality holds due to the assumption that $\|\boldsymbol{\theta}^*\|_2 \leq S$.

Finally, we have

$$\|\boldsymbol{\theta}_k - \boldsymbol{\theta}^*\|_{\boldsymbol{\Sigma}_k} \leq I_1 + I_2 + I_3 \leq R\sqrt{d \log\left(\left(1 + KL^2/\lambda\right)/\delta\right)} + \alpha C + \sqrt{\lambda} S. \tag{254}$$

Therefore, we finish the proof of Lemma 23.

$\square$

