# OpenReview forum: "RLHF with Inconsistent Multi-Agent Feedback Under General Function Approximation: A Theoretical Perspective"
_ICLR.cc/2025/Conference — Submitted to ICLR 2025_

### Official Review · Reviewer_PsxU · 2024-10-23

**Soundness:** 2
**Presentation:** 2
**Contribution:** 2
**Rating:** 5
**Confidence:** 2

**Summary:**

The authors provide a theoretical study on how feedback heterogeneity impacts the learning outcomes in RLHF. Specifically, it formulates an inconsistency model where the difference between feedback from each of the m different sources and the ground truth reward function is bounded above. The authors propose an algorithm with favorable regret guarantees under this model.

**Strengths:**

1. The problem set up is novel and important.

2. Theoretical work on RLHF is important.

**Weaknesses:**

1. Conceptually, I believe addressing the heterogeneity of feedback is an important direction for RLHF. However, I'm struggling with the notion of a "ground truth" reward function in this context. Based on your inconsistency model, it seems that the ground truth reward function can lie anywhere within a region, with the distance between it and each agent's feedback bounded.

However, considering your Example 1, if two agents have fundamentally different preferences, why should we even assume that a single ground truth reward function exists? It seems more reasonable to assume a ground truth reward function in situations where all agents share similar preferences, but their feedback is noisy or inconsistent. Unfortunately, both Example 1 and Figure 1 didn’t clearly convey a message to me.

2.  There are few claims I don’t really follow. Possibly due to my lack of knowledge. I will ask them in the question section.
3.   Writing can be more coherent. See minor suggestions below:

Line 182, maybe $P_m(\tau_k \succ \tau_0)$ since it is the probability of agent m’s choice.

Line 219,$V_1^{\pi_k}(\tau_o)$ is not clearly defined, my guess is that it is $E[\sigma(R^*(\tau^{\pi^k})-\sigma(R^*(\tau_0))]$. And why we have $V_1$ instead of just $V$?

Line 286, $\bar{R_\alpha}$ shows up without definition (after reading I realise it has shown up on line 280)

Line 369, $R_k$ is not defined until next page (line 382). And I am assuming $R_0=R$?

Line 454 what is $\Pi$?

4. In the conclusion part, maybe some suggestions of further work and limitations of your own approach will be great.

**Questions:**

1. Could you explain figure 1 to me please? And what does example 1 trying to illustrate there? (c.f. the first comment in the weakness session)

2. Why mutual information between two agent’s feedback helps learning? (c.f. your line 331 to 341)

3. How is your approach bringing benefits compared to the original RLHF approach? I.e. if I just directly learn the reward model based on all the data, why the regret will be worse?

---

> ### Author Response · Authors · 2024-11-24
> **Official Comment by Authors (Part 1)**
>
> We thank the reviewer for providing the valuable review. We have addressed the reviewer’s comments and modified the paper accordingly. Please note that in the revised paper, we highlighted our changes in blue.
>
> > **Q1**: Conceptually, I believe addressing the heterogeneity of feedback is an important direction for RLHF. However, I'm struggling with the notion of a "ground truth" reward function in this context. Based on your inconsistency model, it seems that the ground truth reward function can lie anywhere within a region, with the distance between it and each agent's feedback bounded.
> >
> > However, considering your Example 1, if two agents have fundamentally different preferences, why should we even assume that a single ground truth reward function exists? It seems more reasonable to assume a ground truth reward function in situations where all agents share similar preferences, but their feedback is noisy or inconsistent. Unfortunately, both Example 1 and Figure 1 didn’t clearly convey a message to me.
> >
> > Could you explain figure 1 to me please? And what does example 1 trying to illustrate there?
>
> **A1**: We clarify that our paper focuses on studying the impact of inconsistency when human reference models could be different from the (single) ground truth and human reference models could be different from each other. We acknowledge that studying the case with multiple ground-truths could be an interesting and important future direction, but it is out of the scope of this paper. We have listed it as an important future work in Section 5.
>
> Moreover, our Example 1 is for illustrating a scenario where different agents make biased choices based on subjective impressions, while a single ground-truth choice exists. Although extending this to consider multiple ground truths tailored to individual agents is an intriguing future work, such an extension requires careful consideration and can be quite challenging. This is because adversarial or malicious feedback from agents could lead to unsafe solutions in practice.
>
> Finally, to address the confusion caused by Fig. 1, we have revised it in the updated version, together with **newly-added Fig. 2**. The **new Fig. 1** highlights the differences between our RLHF setting and the traditional setting. Specifically:
>
>    * **Fig. 1a** illustrates the traditional RLHF setting, where the feedback (depicted in the shaded area) is generated based on the ground-truth reward function $R^*$.
>
>   * **Fig. 1b** illustrates our RLHF setting, where inconsistent multi-agent feedback (depicted in the shaded area) is generated based on the agent-specific reward function $R^m$, which may differ from the ground-truth reward function $R^*$.
>
> > **Q2**: Line 182, maybe $P_m(\tau_k \succ \tau_0)$ since it is the probability of agent $m$’s choice.
>
> **A2**: Thank you. Yes, we have added the index $m$.
>
> > **Q3**: Line 219, $V_{1}^{\pi_k}(\tau_0)$ is not clearly defined, my guess is that it is $E[ \sigma( R^*( \tau^{\pi^k} )) - \sigma( R^*( \tau_0 )) ]$. And why do we have $V_1$ instead of just $V$?
>
> **A3**: Yes, $V_{1}^{\pi_k}(\tau_0) = \mathbb{E} [ \sigma(R^*(\tau^{\pi_k})-R^*(\tau_0)) ]$. We have added it right after Eq. (3). We use the index $1$ to emphasize this $V$-value is for the whole episode starting from step $1$.
>
> > **Q4**: Line 286, $\bar{R}_{\alpha}$ shows up without definition (after reading I realize it has shown up on line 280)
>
> **A4**: Yes, $\bar{R}_{\alpha}$ is defined in line 280, as well as in line 370 (line 373 in the revision).
>
> > **Q5**: Line 369, $R_k$ is not defined until the next page (line 382). And I am assuming $R_0=R$?
>
> **A5**: We clarify that $R_k$ (in line 421 of the revision) has to be defined after defining $\bar{R}_k$ (in line 368 of the revision), because $R_k$ depends on $\bar{R}_k$. We have added explanations for this after defining $\bar{R}_k$ in the revision. Yes, $R_0=R$.
>
> > **Q6**: Line 454 what is $\Pi$?
>
> **A6**: $\Pi$ is a set containing all history-dependent policies. Please see the sentence added in line 465 of the revision.
>
> > **Q7**: In the conclusion part, maybe some suggestions of further work and limitations of your own approach will be great.
>
> **A7**: Thank you. We have added the following discussions in the conclusion part:
>
>    * "Since this work only studies the case with one single ground-truth reward function, it would be interesting to extend our results to the case with multiple (personalized) ground-truth to handle the preference of users. It would also be important to consider a more general form of inconsistency."

---

> ### Author Response · Authors · 2024-11-24
> **Official Comment by Authors (Part 2)**
>
> > **Q8**: Why mutual information between two agents' feedback helps learning? (c.f. your line 331 to 341)
>
> **A8**: This is intuitively because mutual information contributes to stronger statistical dependency, which results in a better concentration property of the historical samples, e.g., multiple agent feedback. We can remove this part if the review feels confusing, since it is only an intuition and a thought experiment, and it does not affect any results in the paper.
>
> > **Q9**: How is your approach bringing benefits compared to the original RLHF approach? I.e. If I just directly learn the reward model based on all the data, why will the regret be worse?
>
> **A9**: The regret will be worse, i.e., $\tilde{O}(\sqrt{\xi K})$ rather than $\tilde{O}(\sqrt{K}+\xi)$, if one directly learns the reward model based on all the data.

---

> > ### Comment · Reviewer_PsxU · 2024-11-25
> >
> > Thank you for your response. Your newly added figures do clarify some of my doubts.
> >
> > "This is intuitively because mutual information contributes to stronger statistical dependency, which results in a better concentration property of the historical samples, e.g., multiple agent feedback. " I see. But I am not sure the concerntration of historical samples is always beneficial? Especially when both sources are of low-quality?

---

> > > ### Author Response · Authors · 2024-11-26
> > >
> > > > "This is intuitively because mutual information contributes to stronger statistical dependency, which results in a better concentration property of the historical samples, e.g., multiple agent feedback." I see. But I am not sure the concentration of historical samples is always beneficial? Especially when both sources are of low-quality?
> > >
> > > **A10**: You made a good point that aligns with the conclusion in our regret results. That is, the concentration of historical samples is **NOT** always beneficial when there exists inconsistency, such as when "sources are of low-quality."
> > >
> > >    * This observation explains why our regret decreases with the number of agents $M$ (mainly due to faster convergence in concentration achieved with more feedback copies) but increases with the level of inconsistency $\xi$ (due to unavoidable gaps caused by agent/source quality discrepancies).
> > >
> > > In particular:
> > >
> > >    1.  When there is no or low inconsistency, e.g., all agents/sources are perfect and $\xi = 0$, our regret will be dominated by the first term, which decreases as the number of agents $M$ increases.
> > >
> > >    2.  When the inconsistency $\xi$ is large, our regret will be dominated by the second term, which increases as the level of inconsistency $\xi$ increases.

---

> > > > ### Comment · Reviewer_PsxU · 2024-11-26
> > > >
> > > > Thanks. But it seems like the claim you made about mutual information isn't precise enough/backed up properly---I would suggest removing claims like this. And I would like to retain the score.

---

> > > > > ### Author Response · Authors · 2024-12-02
> > > > >
> > > > > **A11**: We respectfully disagree that our claim about mutual information isn't precise enough. We clarify that mutual information quantifies the *statistical relationship*, but NOT necessarily *the amount of high-quality information*. Given this, let's consider our claim regarding the impact of both the number of agents $M$ and inconsistency $\xi$ on regret. Specifically:
> > > > >
> > > > > 1. **The Impact of Number of Agents $M$ on Regret**
> > > > >
> > > > >    * **In lines 313-322 of the paper**, our claim about mutual information aims to explain the intuition why (compared to individually considering each agent in Eq. (6)) by jointly considering multiple agents in Eq. (7),
> > > > >
> > > > >          "... the performance would improve with the number of agents $M$ ..."
> > > > >
> > > > >      Importantly, we did NOT claim that the effectiveness of mutual information is unaffected by inconsistency $\xi$. See Bullet 2 below for the impact of $\xi$ on regret.
> > > > >
> > > > >    * **In our answer A8**, we stated that
> > > > >
> > > > >          "... mutual information contributes to stronger statistical dependency, which results in a better concentration property of the historical samples ..."
> > > > >
> > > > >      This reinforces the intuition that combining feedback from multiple agents enhances learning.
> > > > >
> > > > >    * **In our answer A10**, we emphasized that
> > > > >
> > > > >          "... our regret decreases with the number of agents $M$ (mainly due to faster convergence in concentration achieved with more feedback copies) ..."
> > > > >
> > > > >      Again, this supports the intuition and conclusion that increasing $M$ improves performance.
> > > > >
> > > > > 2. **The Impact of Inconsistency $\xi$ on Regret**:
> > > > >
> > > > >    * **In lines 341-line 346 of the paper**, we *explicitly acknowledged* that
> > > > >
> > > > >          "the effectiveness of mutual information is affected by $\xi$."
> > > > >
> > > > >      This is precisely why Eq. (7) is still NOT good enough, and why we introduced modifications leading to Eq. (8) to address inconsistency.
> > > > >
> > > > >    * **In our answer A8**, we aim to answer the reviewer's question
> > > > >
> > > > >          "Why mutual information between two agents' feedback helps learning"
> > > > >
> > > > >      Thus, we claimed that
> > > > >
> > > > >          "... mutual information contributes to stronger statistical dependency ..."
> > > > >
> > > > >      Still, we did NOT claim that the effectiveness of mutual information is unaffected by inconsistency $\xi$.
> > > > >
> > > > >    * **In our answer A10**, we claimed that
> > > > >
> > > > >          "... our regret decreases with the number of agents $M$ ... but increases with the level of inconsistency $\xi$ (due to unavoidable gaps caused by agent/source quality discrepancies) ..."
> > > > >
> > > > >      This reaffirms our acknowledgement that the effectiveness of mutual information is affected by $\xi$.
> > > > >
> > > > > We hope these clarifications resolve the remaining confusion. If our responses satisfactorily address the reviewer's concerns, we kindly request reconsideration of the rating. Thank you!!

---

### Official Review · Reviewer_2ec7 · 2024-10-24

**Soundness:** 3
**Presentation:** 1
**Contribution:** 3
**Rating:** 5
**Confidence:** 1

**Summary:**

This paper studies the problem of reinforcement learning from human feedback (RLHF), where the feedback may come from $M$ different sources each with different reward functions. All these reward functions could be inconsistent with the ground truth reward, where the inconsistency level is denoted by $\xi$. This work focuses on regret minimization under the general function approximation setting. The authors propose four new ideas for the algorithm design, namely "Steiner-Point-Based Confidence Center", "Sub-Importance Sampling for Reducing Complexity", "Scaled Policy Weights for Reducing Biases", and "Optimism in the Face of Policy Uncertainty". The regret scales with $\sqrt{K/M}$ (here $K$ is the number of episodes) with a constant term depending linearly with $\xi$. This suggests that a low inconsistent level and abundant feedback sources are beneficial to RLHF.

**Strengths:**

1. The problem of RLHF with inconsistent feedback sources is novel, interesting and grounded.
2. The algorithm design techniques are highly technical.

**Weaknesses:**

1. The presentation of the paper makes it hard to understand. For example, the authors do not give results in special cases such as tabular MDP (or even bandits) and linear MDP.

**Questions:**

Since I'm having trouble understanding this work, I do not have technical questions.

1. If the algorithm is adapted to tabular multi-armed bandit setting, will the algorithm admit time-efficient implementation (without enumerating the reward transition model in the confidence set)?
2. Can the algorithm be adapted from comparing $(\tau_i, \tau_0)$ to $(\tau_i, \tau_j)$?
3. What will the algorithm and its regret upper bound be for tabular multi-armed bandits?

---

> ### Author Response · Authors · 2024-11-24
>
> We thank the reviewer for providing the valuable review. We have addressed the reviewer’s comments and modified the paper accordingly. Please note that in the revised paper, we highlighted our changes in blue.
>
> > **Q1**: The presentation of the paper makes it hard to understand. For example, the authors do not give results in special cases such as tabular MDP (or even bandits) and linear MDP.
>
> **A1**: We have carefully revised the paper accordingly to improve the clarity and focus, ensuring that the main message of the paper is more accessible. Specifically,
>
> 1. **Alignment of Equations**: Instead of putting all equations in Algorithm 1, in the revision we introduced the important equations directly when introducing each new idea, and then refer to each of these equations in the corresponding step of Algorithm 1. For example:
>
>     * We introduced Eq. (8) and Eq. (12) about using the Steiner point method directly under Idea I and Idea II, and then refer to them in line 6 and line 7 in Algorithm 1, respectively.
>
>     * We introduced Eq. (14) and Eq. (15) about using scaling weights directly under Idea III, and then refer to them in line 10 and line 11 in Algorithm 1, respectively.
>
>     * We introduced Eq. (18) about using OFPU directly under Idea III, and then refer to it in line 14 in Algorithm 1.
>
> 2. **Notation-Heavy Nature**: While the complexity of the studied problem necessitates a certain level of notation, we have added more accessible explanations to simplify both ideas descriptions and Algorithm 1. For example:
>
>    * We connected the steps and lines in Algorithm 1 to the new ideas (see Algorithm 1 on page 6): "New Idea II" for line 4; "New Ideas I and II" for lines 6-8; "New Ideas I and III" for lines 10-12; "New Idea III" for line 14. **Please see the new format of Algorithm 1 on page 6.**
>
>     * We added new illustrative figures, i.e., **Fig. 1a and Fig. 1b**, on page 3 to better explain the differences and challenges in our setting.
>
>    * We added new illustrative figures, i.e., **Fig. 2a, Fig. 2b, and Fig. 2c**, on page 7 to better explain our new ideas I, II, and III.
>
> 3. **Enhancement of Figure 1**: We have changed Fig. 1 to provide a more comprehensive and detailed visualization that better communicates the key concepts of our new setting and considerations. For example:
>
>     * The new **Fig. 1a** on page 3 illustrates that the feedback in traditional RLHF is generated based on the ground truth reward function $R^*$, while the new **Fig. 1b** on page 3 illustrates that the feedback in our case (RLHF with inconsistent multi-agent feedback) is generated based on the human/agent reference function $R^m$ of each agent $m$.
>
> 4. **Additional Visualizations**: In response to your suggestion, we have included new iterative figures, i.e., Fig. 2a, Fig. 2b, and Fig. 2c, on page 7 to illustrate the progression of the proposed method. These figures aim to provide a step-by-step depiction of the theoretical framework, making the ideas more accessible and intuitive. Specifically:
>
>     * **Fig. 2a** illustrates our new idea II on designing sub-importance sampling, where solid and dash arrows illustrate the choice of historical trajectories $\hat{\Gamma}_k$ from $\hat{\Gamma}_k$.
>
>     * **Fig. 2b** illustrates our new idea I on designing the Steiner point method, where the mapping from simple average $\hat{R}_k'$ to the Fermat point $\hat{R}_k$ illustrates Steinerization.
>
>    * **Fig. 2c** illustrates our new idea III about scaling weights and OFPU, where the mapping from the lower function space in the figure to upper policy illustrates the weighted transformation in this idea.
>
> The regret in the tabular MAB case would be $\tilde{O}(\sqrt{\frac{K}{M} }+\xi K)$, the regret in the linear MDP case would be $\tilde{O}(\sqrt{\frac{dK}{M}\mathcal{N} }+\xi d)$, by applying the arguments in vanilla RL for these special cases.
>
> > **Q2**: If the algorithm is adapted to a tabular multi-armed bandit setting, will the algorithm admit time-efficient implementation (without enumerating the reward transition model in the confidence set)?
>
> **A2**:  Yes, our algorithm admits time-efficient implementation in the tabular MAB case.
>
> > **Q3**: Can the algorithm be adapted from comparing $(\tau_i,\tau_0)$ to $(\tau_i,\tau_j)$?
>
> **A3**: Yes, this extension will be straightforward but lengthy. We are happy to add this in the final version.
>
> > **Q4**: What will the algorithm and its regret upper bound be for tabular multi-armed bandits?
>
> **A4**: The regret in the tabular MAB case would be $\tilde{O}(\sqrt{\frac{K}{M} }+\xi K)$.

---

> > ### Comment · Reviewer_2ec7 · 2024-11-25
> >
> > I appreciate the authors for the response. However, even though all of my questions are replied to, the answers of simply "yes, it can" do not address my concerns. For example, the authors should elaborate on the high level idea of how to extend to $(i, j)$ comparison. Given the issues raised by other reviewers, I decide to keep my rating unchanged.

---

> ### Author Response · Authors · 2024-11-26
>
> > However, even though all of my questions are replied to, the answers of simply "yes, it can" do not address my concerns. For example, the authors should elaborate on the high level idea of how to extend to $(i,j)$ comparison.
>
> **A5**: We would like to clarify that such an extension is straightforward, and it has already been demonstrated in traditional RLHF (without inconsistency) literature. We are happy to add this in the paper.
>
> Specifically, mapping from a solution for comparing $(\tau_i,\tau_0)$ to a solution for comparing $(\tau_i,\tau_j)$, the only two minor changes would be as follows:
>
>    1. The idea for estimating the reward function and transition kernel will remain the same, with the adjustment of replacing $\tau_0$ with $\tau_2$​.
>
>    2. The idea for policy update will also remain the same, with the addition of a bonus term for $\tau_2$​ in the same form as that of $\tau_1$​.
>
> > Given the issues raised by other reviewers, I decide to keep my rating unchanged.
>
> **A6**: We would appreciate it if you could clarify any specific issues raised by other reviewers that you feel remain unaddressed. We would be happy to provide further clarification accordingly.

---

> > ### Author Response · Authors · 2024-12-02
> >
> > As the rebuttal period is nearing its end, we wanted to follow up to see if the reviewer has any further questions. We are happy to provide additional clarification or address any remaining points.
> >
> > If our responses have satisfactorily addressed the reviewer's concerns, we would greatly appreciate your consideration in raising the rate of our work. Thank you!!

---

### Official Review · Reviewer_KMHh · 2024-11-02

**Soundness:** 2
**Presentation:** 1
**Contribution:** 2
**Rating:** 3
**Confidence:** 3

**Summary:**

The paper investigates reinforcement learning from human feedback (RLHF) in a setting where feedback is generated by multiple agents whose internal reward functions may differ from an objective ground-truth reward function. This introduces inconsistency, as each agent's feedback reflects their own subjective understanding, influenced by biases or varying priorities.

**Strengths:**

The authors propose a novel algorithm designed to handle such inconsistencies, featuring the development of a Steiner-point-based confidence set that leverages feedback from multiple agents and a weighted importance sampling method to manage complexity. The paper also establishes theoretical guarantees demonstrating that multi-agent feedback can still be beneficial even under inconsistencies, though with limitations based on feedback quality.

**Weaknesses:**

I. The paper's readability is notably poor, presenting significant challenges for comprehension. Specifically:
1. From pages four to ten, the text is densely packed with high-complexity mathematical formulas, making it difficult for readers to follow the theoretical flow without substantial effort. The sheer volume and intricacy of the equations hinder accessibility, even for those familiar with reinforcement learning theory.
2. The pseudocode on page six is highly complex, introducing elaborate steps and technical constructs that are challenging to parse. The dense presentation of the algorithm complicates understanding, as it requires careful and repeated examination to fully grasp its structure and function.

I suggest breaking up dense sections of equations with more explanatory text, providing intuitive explanations alongside technical details, and restructuring the algorithm presentation to be more step-by-step.

II. Furthermore, the definition of Equation 2 seems wrong, as it defines the concept of inconsistency as an epsilon neighborhood, which, in contrast, means consistency.

**Questions:**

1. Why is inconsistency defined as an epsilon neighborhood in Equation 2? What is the rationale behind using this specific formulation, and how does it capture the nuances of multi-agent feedback inconsistency?
2. Why does Equation 3 include a maximization over the inconsistency defined in Equation 2? Could you explain the necessity and impact of this choice on the regret analysis and how it influences the theoretical guarantees?
3. Does your theoretical work translate into a feasible, implementable algorithm with corresponding experimental results? If so, can you provide details on the practical application and any empirical evidence supporting the algorithm's effectiveness?
4. It seems $\sigma$ is misused in Equation 21.

---

> ### Author Response · Authors · 2024-11-24
> **Official Comment by Authors (Part 1)**
>
> We thank the reviewer for providing the valuable review. We have addressed the reviewer’s comments and modified the paper accordingly. Please note that in the revised paper, we highlighted our changes in blue.
>
> > **Q1**: The paper's readability is notably poor, presenting significant challenges for comprehension. Specifically:
> >
> > 1. From pages four to ten, the text is densely packed with high-complexity mathematical formulas, making it difficult for readers to follow the theoretical flow without substantial effort. The sheer volume and intricacy of the equations hinder accessibility, even for those familiar with reinforcement learning theory.
> >
> > 2. The pseudocode on page six is highly complex, introducing elaborate steps and technical constructs that are challenging to parse. The dense presentation of the algorithm complicates understanding, as it requires careful and repeated examination to fully grasp its structure and function.
> >
> > I suggest breaking up dense sections of equations with more explanatory text, providing intuitive explanations alongside technical details, and restructuring the algorithm presentation to be more step-by-step.
>
> **A1**: We have carefully polished this part to make it clearer and more accessible for easier understanding. Specifically:
>
> 1. **For pages four to ten**: While the complexity of the studied problem necessitates a certain level of notation, we have added more accessible explanations to simplify both ideas descriptions and Algorithm 1. For example:
>
>    * We added new illustrative figures, i.e., **Fig. 1a and Fig. 1b**, on page 3 to better explain the differences and challenges in our setting.
>
>    * We added new illustrative figures, i.e., **Fig. 2a, Fig. 2b, and Fig. 2c**, on page 7 to better explain our new ideas I, II, and III.
>
>    * We connected the steps and lines in Algorithm 1 to the new ideas (see Algorithm 1 on page 6): "New Idea II" for line 4; "New Ideas I and II" for lines 6-8; "New Ideas I and III" for lines 10-12; "New Idea III" for line 14. **Please see the new format of Algorithm 1 on page 6.**
>
> 2. **For the pseudocode of Algorithm 1**: Instead of putting all equations in Algorithm 1, in the revision we introduced the important equations directly when introducing each new idea, and then refer to each of these equations in the corresponding step of Algorithm 1. For example:
>
>     * We introduced Eq. (8) and Eq. (12) about using the Steiner point method directly under Idea I and Idea II, and then refer to them in line 6 and line 7 in Algorithm 1, respectively.
>
>     * We introduced Eq. (14) and Eq. (15) about using scaling weights directly under Idea III, and then refer to them in line 10 and line 11 in Algorithm 1, respectively.
>
>     * We introduced Eq. (18) about using OFPU directly under Idea III, and then refer to it in line 14 in Algorithm 1.
>
> We hope these revisions, together with intuitions provided in Section 3, address your concerns and make the theoretical sections more accessible while maintaining the rigor of the analysis. Thank you again for your valuable suggestions.
>
> > **Q2**: The definition of Equation 2 seems wrong, as it defines the concept of inconsistency as an epsilon neighborhood, which, in contrast, means consistency. Why is inconsistency defined as an epsilon neighborhood in Equation 2? What is the rationale behind using this specific formulation, and how does it capture the nuances of multi-agent feedback inconsistency?
>
> **A2**: We clarify that the definition of Eq. (2) is correct, and it defines the *inconsistency* because of the following reasons:
>
> 1.  Eq. (2) quantifies the discrepancy between the human/agent reward model and the ground-truth reward model.
>
> 2. The discrepancy $\xi$ could be arbitrarily large, e.g., scale with time horizon $K$.
>
> Thus, it means the agent reference model could be arbitrarily different from the ground truth, and hence means the level of inconsistency.

---

> ### Author Response · Authors · 2024-11-24
> **Official Comment by Authors (Part 2)**
>
> > **Q3**: (a) Why does Equation 3 include a maximization over the inconsistency defined in Equation 2? (b) Could you explain the necessity and impact of this choice on the regret analysis and how it influences the theoretical guarantees?
>
> **A3**: For question (a), the reasons are as follows:
>
>    1. The maximization in Eq. (3) is included because we consider a new type of regret for optimizing the policy against the worst-case inconsistency.
>
>    2. **Without such maximization**, the problem reduces back to the standard RLHF setting.
>
> For question (b), the impacts of this maximization are listed below (from a high-level perspective). Our  whole paper (including both the algorithm design and regret analysis) is for addressing such impacts/challenges.
>
> 1. **Impact on estimating the confidence set and proving the high-probability event**: Since the agent feedback is inconsistent, for leveraging such feedback, a natural idea would be to use the feedback of each agent to estimate their reward models, and then search for the optimal policy jointly. However, this will lose the fundamental power of multi-feedback, i.e., the resulting performance does not improve with the number of agents. Thus, we should estimate the confidence center by utilizing multi-feedback simultaneously. However, the traditional complexity analysis in RL does not apply, since the confidence center, as well as the ground truth, may be outside of the agent reward function space and arbitrarily dynamic due to inconsistency (see Fig. 1 and Fig. 2). To address this new difficulty, we non-trivially modify Steiner-Point Approximation from theoretical physics and combinatorial geometry, which requires fundamentally new analytical methods in RL for a sub-linear regret. Due to this new construction, it is clear the high-probability event will be fundamentally affected, as shown in Appendix B.1.
>
> 2. **Impact on the functional complexity and important samples**: Due to the nature of multi-agent feedback and general function approximation, the traditional sample-based complexity would result in a final regret increasing linearly in time horizon $K$. To address this new difficulty, one needs to design a new approximation method to select the sampled trajectories, e.g., sub-importance sampling under Fermat analysis as we mentioned in Appendix B.1 and Appendix B.2, such that the functional complexity is reduced as it is based on only a subset of sensitive samples. It is clear that the new layer of complexity also needs to be captured in the analysis.
>
> 3. **Impact on the biases in optimizing policies and the traditional optimism-in-the-Face-of-Uncertainty**: There have been solutions on addressing the biases in the feedback, e.g., add weights to the action selection step. Directly applying this does not work due to the heterogeneous feedback in our case. To resolve this, one needs to design a different scaling weight directly on the policy, such that a greedy decision under policy uncertainty in our case still guarantees optimality.
>
> > **Q4**: Does your theoretical work translate into a feasible, implementable algorithm with corresponding experimental results? If so, can you provide details on the practical application and any empirical evidence supporting the algorithm's effectiveness?
>
> **A4**: Thank you for the suggestion. Our theoretical work translates into a feasible, implementable algorithm. One practical application would be the motivating example introduced in Section 2, and we will add numerical results for more practical applications.
>
> > **Q5**: It seems $\sigma$ is misused in Equation 21.
>
> **A5**: We clarify that $\sigma$ in Eq. (21) is correct, since $\sigma(\tau\mid R) \triangleq \sigma(R(\tau)-R(\tau_0))$ as defined at the beginning of Section 3. In view of this comment, we have repeated this definition right before Eq. (21) in the revised paper.

---

> ### Comment · Reviewer_KMHh · 2024-11-25
>
> Thanks for replying. Some concerns have not been addressed, especially the definition of inconsistency.

---

> > ### Author Response · Authors · 2024-11-26
> >
> > > Some concerns have not been addressed, especially the definition of inconsistency.
> >
> > **A6**: We would appreciate it if you could clarify any specific concern you still have regarding the inconsistency definition or any other parts. In addition to our responses A2 and A3, we would like to note that similar definitions have been widely adopted in various problems, such as robust bandits/RL [He et al., 2022; Wei et al., 2022, etc.] and robust optimization [Li et al., 2019, etc.], with minor variations tailored to the nature of each specific problem.
> >
> > J. He, D. Zhou, T. Zhang, and Q. Gu. Nearly optimal algorithms for linear contextual bandits with adversarial corruptions. Advances in neural information processing systems,
> > 35:34614–34625, 2022.
> >
> > C. Wei, C. Dann, and J. Zimmert. A model selection approach for corruption robust reinforcement learning. International Conference on Algorithmic Learning Theory, pp. 1043–1096, 2022.
> >
> > Y. Li, E. Y. Lou, and L. Shan. Stochastic linear optimization with adversarial corruption. arXiv preprint arXiv:1909.02109, 2019.

---

> > > ### Comment · Reviewer_KMHh · 2024-11-26
> > >
> > > Could you point out where the similar definition exactly are in your references?

---

> ### Author Response · Authors · 2024-11-26
>
> No equation numbers in the references, but they are as follows:
>
> [He et al., 2022]: In the preliminaries section (i.e., Section 2), they have $\sum_k |r_k - r_k'| \leq C$.
>
> [Wei et al., 2022]: In the problem setting section (i.e., Section 3), they have $\sum_t \max_a |\mu^a - \mu_t^a| \leq C$.
>
> [Li et al., 2019]: In the preliminaries section (i.e., Section 2), they have $\sum_t \max_x |r_t(x) - r_t'(x)| \leq C$.

---

> > ### Author Response · Authors · 2024-12-02
> >
> > As the rebuttal period is nearing its end, we wanted to follow up to see if the reviewer has any further questions. We are happy to provide additional clarification or address any remaining points.
> >
> > If our responses have satisfactorily addressed the reviewer's concerns, we would greatly appreciate your consideration in raising the rate of our work. Thank you!!

---

### Official Review · Reviewer_9VYV · 2024-11-04

**Soundness:** 3
**Presentation:** 1
**Contribution:** 3
**Rating:** 5
**Confidence:** 2

**Summary:**

The paper focuses on the problem of using inconsistent feedbacks for RLHF under general function approximation due to possible human biases and diverse experiences. The real-world human feedbacks are expected to be different than the ground-truth reward function and possible be inconsistent with them and other human feedbacks. Therefore, the investigation is on whether the inconsistency helps the learning or exacerbate it. The performance is evaluated by the regret under inconsistency definition. The paper presents their RLHF algorithm based on Steiner point to solve this problem and present 3 ideas to tackle the issue and provide a theoretical analysis on regret due to inconsistent multi-agent feedback.

**Strengths:**

* A novel algorithm RLHF-IMAF is proposed.
* Inconsistent feedback for RLHF is indeed a topic that is understudied and worth more attention due to the nature of human feedback.
* The problem is well motivated.
* The reasoning behind proposed ideas are well described and the necessary proofs are provided.

**Weaknesses:**

* The paper is not well written. There are alignment issues with the equations. Due to the nature of the paper, it is heavily notated; but this makes it harder to focus on the message of the paper.
* Providing some visualizations would be helpful to convey the proposed ideas. The visualization in Figure 1 is insufficient to get the message across. Since the paper is fully theoretical, there may not be any experiment to validate the approach. But a visualization consisting of iterative figures could be good.
* The explanation for the results in appendix and the references to the appendix are missing. The claims in the main paper and proofs in the appendix are disconnected from each other, and the information put in appendix is not explained. The proofs in the appendix and the theoretical analysis in the main paper all look disconnected.
* A toy experiment to validate the algorithm would be useful.

**Questions:**

* How do you evaluate that you achieved what you proposed? Given the theoretical analysis, you suggest that multi-agent feedback even under inconsistency is helpful; but then you assert that if the feedback has poor quality, even when M is large, the regret is still bad although the feedback is helpful. How can the feedback be helpful when the regret is bad if we evaluate the benefit of the method with the regret?

---

> ### Author Response · Authors · 2024-11-24
> **Official Comment by Authors (Part 1)**
>
> We thank the reviewer for providing the valuable review. We have addressed the reviewer’s comments and modified the paper accordingly. Please note that in the revised paper, we highlighted the changes in blue.
>
> > **Q1**: The paper is not well written. There are alignment issues with the equations. Due to the nature of the paper, it is heavily notated; but this makes it harder to focus on the message of the paper.
>
> **A1**: We have carefully revised the paper accordingly to improve the clarity and focus, ensuring that the main message of the paper is more accessible. Specifically:
>
> 1. **Alignment of Equations**: Instead of putting all equations in Algorithm 1, in the revision we introduced the important equations directly when introducing each new idea, and then refer to each of these equations in the corresponding step of Algorithm 1. For example:
>
>     * We introduced Eq. (8) and Eq. (12) about using the Steiner point method directly under Idea I and Idea II, and then refer to them in line 6 and line 7 in Algorithm 1, respectively.
>
>     * We introduced Eq. (14) and Eq. (15) about using scaling weights directly under Idea III, and then refer to them in line 10 and line 11 in Algorithm 1, respectively.
>
>     * We introduced Eq. (18) about using OFPU directly under Idea III, and then refer to it in line 14 in Algorithm 1.
>
> 2. **Notation-Heavy Nature**: While the complexity of the studied problem necessitates a certain level of notation, we have added more accessible explanations to simplify both ideas descriptions and Algorithm 1. For example:
>
>    * We connected the steps and lines in Algorithm 1 to the new ideas (see Algorithm 1 on page 6): "New Idea II" for line 4; "New Ideas I and II" for lines 6-8; "New Ideas I and III" for lines 10-12; "New Idea III" for line 14. **Please see the new format of Algorithm 1 on page 6.**
>
>     * We added new illustrative figures, i.e., **Fig. 1a and Fig. 1b**, on page 3 to better explain the differences and challenges in our setting.
>
>    * We added new illustrative figures, i.e., **Fig. 2a, Fig. 2b, and Fig. 2c**, on page 7 to better explain our new ideas I, II, and III.
>
> We hope these revisions, together with intuitions provided in Section 3, address your concerns and improve the overall writing quality of the paper. We sincerely thank you for bringing these issues to our attention.
>
> > **Q2**: Providing some visualizations would be helpful to convey the proposed ideas. The visualization in Figure 1 is insufficient to get the message across. Since the paper is fully theoretical, there may not be any experiment to validate the approach. But a visualization consisting of iterative figures could be good.
>
> **A2**: Thank you for your suggestion regarding the use of visualizations to better convey our proposed ideas. We have revised the paper accordingly. Specifically:
>
> 1. **Enhancement of Figure 1**: We have changed Fig. 1 to provide a more comprehensive and detailed visualization that better communicates the key concepts of our new setting and considerations. For example:
>
>     * The new **Fig. 1a** on page 3 illustrates that the feedback in traditional RLHF is generated based on the ground truth reward function $R^*$, while the new **Fig. 1b** on page 3 illustrates that the feedback in our case (RLHF with inconsistent multi-agent feedback) is generated based on the human/agent reference function $R^m$ of each agent $m$.
>
> 2. **Additional Visualizations**: In response to your suggestion, we have included new iterative figures, i.e., Fig. 2a, Fig. 2b, and Fig. 2c, on page 7 to illustrate the progression of the proposed method. These figures aim to provide a step-by-step depiction of the theoretical framework, making the ideas more accessible and intuitive. Specifically:
>
>     * **Fig. 2a** illustrates our new idea II on designing sub-importance sampling, where solid and dash arrows illustrate the choice of historical trajectories $\hat{\Gamma}_k$ from $\Gamma_k$.
>
>     * **Fig. 2b** illustrates our new idea I on designing the Steiner point method, where the mapping from simple average $\hat{R}_k'$ to the Fermat point $\hat{R}_k$ illustrates Steinerization.
>
>    * **Fig. 2c** illustrates our new idea III about scaling weights and OFPU, where the mapping from the lower function space in the figure to upper policy illustrates the weighted transformation in this idea.
>
> We hope these updates address your concerns and improve the overall clarity and presentation of the paper. Thank you again for your valuable feedback.

---

> ### Author Response · Authors · 2024-11-24
> **Official Comment by Authors (Part 2)**
>
> > **Q3**: The explanation for the results in appendix and the references to the appendix are missing. The claims in the main paper and proofs in the appendix are disconnected from each other, and the information put in the appendix is not explained. The proofs in the appendix and the theoretical analysis in the main paper all look disconnected.
>
> **A3**: Thank you. We have carefully polished the explanation in appendices and added references to the appendices. Specifically:
>
> 1. **References to the Appendix**: To strengthen the connection between the main paper and the appendices, we have added more references within the main text, guiding readers to relevant sections of the appendices when needed. For example:
>
>    * In the proof sketch of Section 4, we added the reference (in line 499 of the paper) to Appendix B.1 for main step 1 of our proof.
>
>    * In the proof sketch of Section 4, we added the reference (in line 504 of the paper) to Appendix B.2 for main step 2 of our proof. We also added the reference (in line 502) to Appendix D for necessary but standard proof steps that are not included in Appendix B.2.
>
>    * In the proof sketch of Section 4, we added reference (in line 524 of the paper) to Appendix B.3 for main step 3 of our proof. We also added reference (in line 508) to Appendix E for necessary but standard proof steps that are not included in Appendix B.3.
>
> 2. **Explanation of Results in the Appendices**: We have added detailed explanations and explicit references for the results presented in the appendices to ensure they are well-contextualized and easier to follow. For example,
>
>    * We have revised the high-level explanation of our proof for Theorem 1 at the beginning of Appendix B, where we added the references to Appendix B.1, Appendix B.2, and Appendix B.3 for each of the steps correspondingly.
>
>     * The supporting results for the proofs in Appendix B are provided in Appendix C, and the supporting lemmas have been referred to in Appendix B. For example, we referred to Lemma 4 to prove Eq. (34), Eq. (52), and Eq. (53). We referred to Lemma 5 to prove Eq. (49). We referred to Lemma 10 to prove Proposition 2.
>
> We hope these revisions address your concerns and improve the overall coherence and presentation of the manuscript. Thank you again for your valuable suggestions.
>
> > **Q4**: A toy experiment to validate the algorithm would be useful.
>
> **A4**: Thank you. We will add a toy experiment with numerical results in the revision.
>
> > **Q5**: How do you evaluate that you achieved what you proposed? Given the theoretical analysis, you suggest that multi-agent feedback even under inconsistency is helpful; but then you assert that if the feedback has poor quality, even when $M$ is large, the regret is still bad although the feedback is helpful. How can the feedback be helpful when the regret is bad if we evaluate the benefit of the method with the regret?
>
> **A5**: We clarify that the interaction between the feedback quality (represented by $\zeta$) and the number of feedback sources (represented by $M$) does not conflict but rather complements the regret result. This is also why this problem is so interesting and challenging. Specifically:
>
> 1. **Role of Feedback Quality $\zeta$**: When $\zeta$ is low, the feedback is of poor quality, which naturally impacts the overall regret negatively, as the feedback contributes less reliable information to the learning process.
>
> 2. **Role of Feedback Quantity $M$**: Increasing $M$ introduces a larger volume of feedback, which helps mitigate inconsistencies when the feedback is of reasonable quality. However, if the feedback quality $\zeta$ is uniformly poor, increasing $M$ alone cannot sufficiently counteract the negative effect on regret.

---

> ### Author Response · Authors · 2024-11-24
> **Official Comment by Authors (Part 3)**
>
> 3. **Impact of Combined Parameters**: The interaction between $\zeta$ and $M$ does not conflict but rather complements the theoretical analysis. Feedback remains "helpful" in the sense that it provides information contributing to learning, but when evaluated through regret, the benefit diminishes as $\zeta$ increases, regardless of $M$. This aligns with our analysis, which emphasizes that both quality and quantity are critical for achieving low regret.
>
> We have updated the manuscript to explicitly address this point and provide a clearer explanation of how $\zeta$ and $M$ interact.  Please see lines 485-490 in Section 4, and we quoted below for the reviewer's convenience.
>
>     "(i) The regret decreases with $M$, indicating that having more feedback sources is generally beneficial, even in the presence of inconsistency. This highlights the utility of multi-agent feedback in improving performance. (ii) However, the regret also includes a term dependent on $\xi$ that does not decrease with $M$. This indicates that while increasing $M$ can mitigate some effects of inconsistency, if the feedback quality is consistently poor (i.e., high $\xi$), part of the overall regret remains significant regardless of $M$. Thus, the benefit of additional feedback is *limited* by its quality."
>
> We hope this resolves the concern and enhances the understanding of our theoretical framework.

---

> ### Author Response · Authors · 2024-11-26
>
> As the rebuttal period is nearing its end, we wanted to follow up to see if the reviewer have any further questions. We are happy to provide additional clarification or address any remaining points.
>
> If our responses have sufficiently clarified your concerns, we would greatly appreciate your consideration in raising the rate of our work.

---

> ### Comment · Reviewer_9VYV · 2024-11-27
>
> I thank the authors for all the clarifications and revisions. The updated manuscript is indeed improved, thus I am increasing my score. Nevertheless, I still believe the work is below acceptance threshold and needs to be improved further.

---

> > ### Author Response · Authors · 2024-12-02
> >
> > We very much appreciate that you have increased your score and recognize the effort we have put into addressing your concerns. We would greatly appreciate it if the reviewer could explicitly point out major weaknesses that are the reasons for the score being below the acceptance threshold. We are fully committed to refining the paper based on your feedback.
> >
> > If our responses have satisfactorily addressed the reviewer's concerns, we kindly request reconsideration of the rating. Thank you!!

---

### Meta-Review · Area_Chair_yffW · 2024-12-22

**Metareview:**

This paper explores RLHF with heterogeneous multi-agent feedback. The main concerns raised by reviewers are as follows:

- Poor Presentation: All reviewers found the current version of the paper poorly presented and unsuitable for acceptance. Its highly technical and math-heavy nature makes it difficult for the general ML audience to follow. A major restructuring of the paper would significantly improve its clarity and accessibility for future submissions.

- Weak Justification for the New Preference Model: The motivation for introducing this new preference model is not well justified. It remains unclear how this model offers advantages over the standard BT model in handling heterogeneous human preferences.

Additional Note: The BT model might already be capable of addressing heterogeneous human preferences, as it can potentially capture the distribution of preferences at the population level, even if individual preferences differ.

**Additional Comments On Reviewer Discussion:**

All reviewers unanimously agree the paper is poorly written. This issue was not sufficiently addressed during the rebuttal phase. This is the main reason I recommended for rejection.

---

### Decision · Program_Chairs · 2025-01-22

Reject